# Learning Rate Scaling across LoRA Ranks and Transfer to Full Finetuning

Nan Chen [1]   Soledad Villar [1]   Soufiane Hayou [1]

## Abstract

Low-Rank Adaptation (LoRA) is a standard tool for parameter-efficient finetuning of large models. While it induces a small memory footprint, its training dynamics can be surprisingly complex as they depend on several hyperparameters such as initialization, adapter rank, and learning rate. In particular, it is unclear *how the optimal learning rate scales with adapter rank*, which forces practitioners to re-tune the learning rate whenever the rank is changed. In this paper, we introduce *Maximal-Update Adaptation* ($\mu$A), a theoretical framework that characterizes how the "optimal" learning rate should scale with model width and adapter rank to produce stable, non-vanishing feature updates under standard configurations. Our analysis leverages techniques from hyperparameter transfer and reveals that the optimal learning rate exhibits different scaling patterns depending on initialization and LoRA scaling factor. Specifically, we identify two regimes: one where the optimal learning rate remains roughly invariant across ranks, and another where it scales inversely with rank. We further identify a configuration that allows learning rate transfer from LoRA to full finetuning, drastically reducing the cost of learning rate tuning for full finetuning. Experiments across language, vision, vision–language, image generation, and reinforcement-learning tasks validate our scaling rules and show that learning rates tuned on LoRA transfer reliably to full finetuning.

## 1. Introduction

The widespread availability of large pretrained models has made finetuning the standard approach to adapt a general-purpose model to a specific downstream task (Radford et al.,

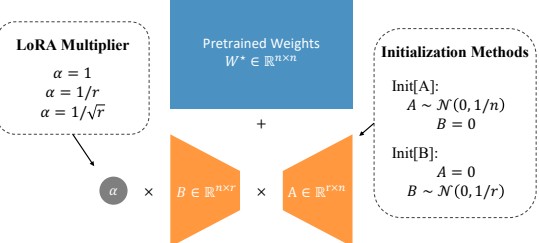

*Figure 1.* LoRA finetuning with standard design choices.

2018; Devlin et al., 2019). At the same time, model capability typically improves with scale, with state-of-the-art language and vision models now reaching hundreds of billions of parameters (Liu et al., 2024a; Yang et al., 2025; Team et al., 2025). The improvement in performance with scale makes larger models attractive to practitioners, but it also makes full finetuning (FFT) increasingly expensive. While FFT remains the standard for tasks such as instruction-tuning (Biderman et al., 2024), parameter-efficient finetuning (PEFT) has become a default approach in many settings (Han et al., 2024). Low-Rank Adaptation (LoRA) (Hu et al., 2022) is arguably the most popular parameter-efficient finetuning method. It adds a low-rank factorization $\alpha BA$ to the (frozen) pretrained weight matrix $W^\star \in \mathbb{R}^{n \times n}$, where $B \in \mathbb{R}^{n \times r}$, $A \in \mathbb{R}^{r \times n}$, $r$ is the LoRA rank, and $\alpha$ is a scaling multiplier. Only $B$ and $A$ are trainable in this setting.

Deploying LoRA in practice requires several design choices beyond the rank $r$. To start from a no-op adapter, practitioners typically initialize one factor to random values and the other to zero; we denote these two common choices as Init[A] (random $A$, $B = 0$) and Init[B] (random $B$, $A = 0$) (Hayou et al., 2024a). In addition, common implementations use different rank-dependent multipliers, typically $\alpha \in \{1, r^{-1}, r^{-1/2}\}$ (Kalajdzievski, 2023; Biderman et al., 2024); Figure 1 summarizes these standard choices. Because LoRA update is bilinear in $A$ and $B$, the initialization, multiplier $\alpha$, rank $r$, and learning rate $\eta$ are coupled and jointly determine the effective update magnitude. This coupling makes hyperparameter tuning brittle in practice. Changing the rank or switching between initialization and multiplier choices can render a previously optimal learning rate suboptimal, forcing costly re-tuning. Figure 2 provides a representative example on Llama-3.2-1B finetuned on the

[1]Department of Applied Mathematics and Statistics, Johns Hopkins University, Maryland, United States of America. Correspondence to: Soledad Villar <soledad.villar@jhu.edu>, Soufiane Hayou <hayou@jhu.edu>.

*Proceedings of the 43rd International Conference on Machine Learning*, Seoul, South Korea. PMLR 306, 2026. Copyright 2026 by the author(s).

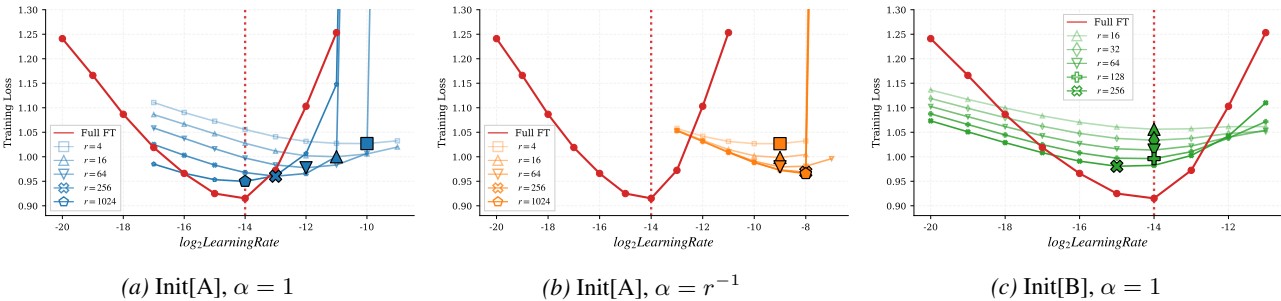

*Figure 2.* Learning rate sweeps for three standard LoRA configurations on Llama-3.2-1B (Tulu3). Each panel plots final training loss (EMA-smoothed) versus log-scale learning rate $\log_2(\eta)$ for multiple ranks (colored) and FFT (red). Large markers indicate the optimal learning rate for each rank; the dashed red line marks the optimal FFT learning rate. The y-axis is clipped for readability.

Tulu3 dataset, showing end-of-training loss as a function of learning rate for multiple ranks under three common configurations. Under Init[A] with constant $\alpha = 1$, the best learning rate decreases with rank $r$ (Figure 2a). Setting $\alpha = r^{-1}$ while keeping Init[A] yields a different pattern: the optimal learning rate becomes approximately rank-invariant, but the stable range is narrow—small increases above the optimum cause divergence (Figure 2b). With Init[B] and constant $\alpha = 1$, the optimal learning rate is again rank-invariant and now aligns with the FFT optimum (Figure 2c). Notably, this optimal rate is more than an order of magnitude smaller than under Init[A] with $\alpha = r^{-1}$. Taken together, these experiments show that the learning rate–rank relationship changes qualitatively across initialization and multiplier choices. This raises a natural question: *how does the optimal learning rate $\eta$ depend on rank $r$?*

To answer this question, we analyze LoRA finetuning in the joint limit $n, r \to \infty$. Building on prior works analyzing infinite-width networks (Yang & Hu, 2020; 2021; Hayou et al., 2024a;b), we study how initialization, multiplier $\alpha$, and learning rate $\eta$ jointly govern the magnitude of LoRA updates under a tractable theoretical setup. This allows us to characterize when LoRA updates remain bounded yet non-vanishing in the large-$n$, large-$r$ regime—avoiding instability while ensuring learning does not stall. We call this regime the Maximal-Update Adaptation ($\mu$A). From this analysis, we derive explicit scaling prescriptions for how the learning rate should depend on model width and LoRA rank under each configuration. These prescriptions explain why some configurations (e.g., Init[A] with constant $\alpha$) require the learning rate to decrease with rank, while others (e.g., Init[B] with constant $\alpha$) admit rank-invariant learning rates. Our analysis also connects LoRA to FFT. For Init[B] with constant $\alpha$, the learning rate scaling that ensures stable feature learning matches that of FFT under the same assumptions. This matching is a necessary condition for hyperparameter transfer between the two regimes.

We verify our theory through extensive experiments spanning diverse tasks, modalities, and training paradigms.

Across all settings, the observed learning rate–rank patterns consistently match our derived scaling rules, confirming that $\mu$A scaling rules generalize across architectures and domains. Our contributions are summarized as follows:

- We introduce $\mu$A (Maximal-Update Adaptation), a theoretical framework for LoRA finetuning dynamics in the joint limit $n, r \to \infty$ and characterize how the optimal learning rate scales with model width and adapter rank to ensure maximal (stable) adaptation.

- Our analysis reveals two distinct regimes: one where the optimal learning rate scales inversely with rank, and one in which it remains rank-invariant. We identify a LoRA configuration where the optimal learning rate matches that of FFT (transfer from LoRA to FFT).

- Extensive experiments demonstrate that $\mu$A scaling rules support learning rate transfer across LoRA ranks and from LoRA to FFT.

**Conflict of Interest Disclosure.** The authors declare no financial conflicts of interest related to this work.

## 2. Related Works

**Parameter-efficient finetuning.** PEFT techniques fall into three main categories (Han et al., 2024; Ji et al., 2025): (i) selectively updating a subset of parameters (Collobert & Weston, 2008; Guo et al., 2021; Liao et al., 2023); (ii) adding lightweight adapters (Rebuffi et al., 2017; Houlsby et al., 2019; Lin et al., 2020; He et al., 2022; Hu et al., 2022); or (iii) tuning continuous prompts (Li & Liang, 2021; Lester et al., 2021; Liu et al., 2023). LoRA (Hu et al., 2022) exemplifies the additive method and has inspired numerous variants targeting efficiency (Valipour et al., 2023; Zhang et al., 2023; Dettmers et al., 2023; Kopiczko et al., 2024; Ke et al., 2024; Li et al., 2024a; Liu et al., 2024b; Hayou et al., 2025) and stability (Hayou et al., 2024b;a; Yen et al., 2025). Recent studies compare LoRA and FFT across gradient alignment (Wang et al., 2024; 2025), expressive power (Zeng &

Lee, 2024), and spectral structure of weight updates (Shuttleworth et al., 2024). Complementing these studies, we identify a LoRA configuration where the optimal learning rate matches that of FFT, supporting learning rate transfer between the two methods.

**Design choices of LoRA.** Standard schemes randomly initialize either $A$ or $B$ using established techniques (Glorot & Bengio, 2010; He et al., 2015), setting the other to zero. Alternative methods initialize weights from pretrained weights (Büyükakyüz, 2024; Meng et al., 2024), deterministic orthogonal bases (Zhang et al., 2025), downstream data (Yang et al., 2024b; Wang et al., 2024; Paischer et al., 2025), or quantized models (Li et al., 2024b). Prior theoretical analyses of initialization (Hayou et al., 2024a; Li et al., 2025) focus on the infinite-width regime and treat the rank as constant; in contrast, we study the joint limit where both width $n$ and rank $r$ grow. For the LoRA multiplier $\alpha$, the original LoRA (Hu et al., 2022) uses $\alpha = r^{-1}$ to keep update magnitudes controlled, rsLoRA (Kalajdzievski, 2023) proposes $\alpha = r^{-1/2}$ for stability at larger ranks, and recent works increasingly adopt $\alpha = 1$ (Gabrielsson et al., 2024; Biderman et al., 2024; Charakorn et al., 2025). Schulman (2025) empirically studies LoRA design choices but provides only heuristic rules for learning rate transfer between LoRA and FFT, acknowledging this as an open problem that we address rigorously in this work.

**Asymptotic scaling analysis.** Our analysis builds on prior works analyzing the asymptotic behavior of neural networks as width approaches infinity (Yang & Hu, 2020; 2021; Yang et al., 2021; Hayou et al., 2024a;b). Related works apply similar frameworks to analyze depth scaling (Hayou et al., 2019; 2021; He et al., 2023; Hayou & Yang, 2023; Noci et al., 2023; Yang et al., 2024a). Most closely related to ours are Hayou et al. (2024a;b), which analyze LoRA finetuning in the limit $n \to \infty$ but treat the rank $r$ as constant. This analysis does not reveal how training dynamics depend on the rank $r$. Here, we analyze the joint limit $n, r \to \infty$, yielding scaling prescriptions that depend on both network width and LoRA rank. Empirically, the optimal learning rates closely match the derived scaling rules.

## 3. Setup and Definitions

We introduce definitions and notation for our theoretical analysis, following Hayou et al. (2024a).

### 3.1. LoRA Layer and Features

**Definition 3.1** (Low-Rank Adaptation (LoRA))**.** Let $W^\star \in \mathbb{R}^{n \times n}$ be a pretrained weight matrix that remains fixed during finetuning. LoRA parameterizes the weights as

$$W = W^\star + \alpha BA,$$

where $B \in \mathbb{R}^{n \times r}$ and $A \in \mathbb{R}^{r \times n}$ are trainable factors with rank $r \ll n$ and $\alpha \in \mathbb{R}$ is a scalar multiplier.

For a generic LoRA-augmented linear layer, we denote the input by $\underline{Z} \in \mathbb{R}^n$ and the output by $\bar{Z} \in \mathbb{R}^n$,

$$\bar{Z} = W^\star \underline{Z} + \alpha BA\underline{Z}.$$

LoRA factors are usually initialized with $BA = 0$ so that finetuning starts from the pretrained model. There are two straightforward schemes that satisfy this property.

**Definition 3.2** (Initialization schemes for LoRA)**.** To initialize the LoRA update as a no-op, we consider:[1]

- **Init[A]:** $A_0$ has i.i.d. entries $\mathcal{N}(0, 1/n)$ and $B_0 = 0$.

- **Init[B]:** $B_0$ has i.i.d. entries $\mathcal{N}(0, 1/r)$ and $A_0 = 0$.

We now define the intermediate and output LoRA features.

**Definition 3.3** (LoRA features)**.** For a layer input $\underline{Z}$, define the intermediate and output LoRA features

$$Z_A := A\underline{Z} \in \mathbb{R}^r, \qquad Z_B := \alpha B Z_A = \alpha BA\underline{Z} \in \mathbb{R}^n.$$

We consider a fixed number $T$ of finetuning steps, where $T$ is independent of $n$ and $r$. For $t \leq T$, we use superscripts to denote LoRA features at step $t$, i.e. $Z_A^t$, $Z_B^t$ and subscripts to denote weights at step $t$, i.e. $A_t$, $B_t$.

To isolate the contribution of a single LoRA module to feature learning, we assume that only one LoRA layer is trainable while all other network components remain frozen. Under this assumption, the layer input $\underline{Z}$ (corresponding to a given model input $x$) remains constant during finetuning.

At any step $t$, recall that $Z_A^t = A_t \underline{Z}$ and $Z_B^t = \alpha B_t Z_A^t$. Expanding the increment $\Delta Z_B^t := Z_B^t - Z_B^{t-1}$ yields:

$$\Delta Z_B^t = \underbrace{\alpha B_{t-1}\Delta Z_A^t}_{\delta_t^1} + \underbrace{\alpha \Delta B_t Z_A^{t-1}}_{\delta_t^2} + \underbrace{\alpha \Delta B_t \Delta Z_A^t}_{\delta_t^3} \quad (1)$$

We refer to the three contributions in Equation (1) as $\delta_t^1$, $\delta_t^2$, and $\delta_t^3$. As previously explained in Hayou et al. (2024b), the terms $\delta_t^1$ and $\delta_t^2$ correspond to *linear* updates in which one LoRA factor is trained while the other is held fixed, whereas $\delta_t^3$ corresponds to the *multiplicative updates* of $A$ and $B$. The next section discusses this decomposition and how it relates to stable feature learning.

### 3.2. Stable Feature Learning

We aim to understand how feature updates behave as a function of model width $n$ and LoRA rank $r$. For this purpose,

---

[1]The choice of the Gaussian distribution here is for simplification purpose only. It can be replaced with any centred distribution with finite second moment.

we consider the setting where both $n$ and $r$ grow, subject to $r \le n$. To formalize how features and updates scale with $(n, r)$, we adopt the following asymptotic notation.

**Definition 3.4** (Asymptotic notation). For sequences $c_{n,r} \in \mathbb{R}$ and $d_{n,r} \in \mathbb{R}_+$, we write $c_{n,r} = \mathcal{O}(d_{n,r})$ (resp. $c_{n,r} = \Omega(d_{n,r})$) if there exist constants $\kappa > 0$ and $N \in \mathbb{N}$ such that for all $n, r > N$ with $r \le n$, we have $|c_{n,r}| < \kappa d_{n,r}$ (resp. $|c_{n,r}| > \kappa d_{n,r}$). We write $c_{n,r} = \Theta(d_{n,r})$ if both bounds hold. This notation extends element-wise to vectors: for $c_{n,r} = (c_{n,r}^i)_{i=1}^k \in \mathbb{R}^k$ and $d_{n,r} = (d_{n,r}^i)_{i=1}^k \in \mathbb{R}_+^k$, we write $c_{n,r} = \mathcal{O}(d_{n,r})$ when $c_{n,r}^i = \mathcal{O}(d_{n,r}^i)$ for all $i \in [k]$, and similarly for $\Omega$ and $\Theta$. For random variables, the notation is understood in the second-moment sense: $c_{n,r} = \mathcal{O}(d_{n,r})$ means $\left(\mathbb{E}|c_{n,r}|^2\right)^{1/2} = \mathcal{O}(d_{n,r})$.

Our stability objective requires that the LoRA-induced feature contribution $Z_B^t$ remains bounded over $T$ finetuning steps (where $T$ is fixed), so that activations remain bounded.

**Definition 3.5** (Feature stability). Fix a finite number of finetuning steps $T$ independent of $(n, r)$. We say LoRA finetuning is *stable* if for all steps $t \le T$, we have $Z_B^t = \mathcal{O}(1)$ in the joint limit $n, r \to \infty$ with $r \le n$.

Since LoRA is typically initialized as a no-op ($Z_B^0 = 0$), Lemma 3.6 below implies that bounding the per-step increments $\Delta Z_B^t = \mathcal{O}(1)$ is sufficient for stability.

**Lemma 3.6** (Telescoping bound). *Fix $t \le T$. If $Z_B^0 = \mathcal{O}(1)$ and $\Delta Z_B^s = \mathcal{O}(1)$ for all $s \le t$, then $Z_B^t = \mathcal{O}(1)$.*

However, stability alone permits trivial solutions—for instance, initializing both $A$ and $B$ to zero yields vanishing updates. We therefore require that the LoRA feature increment remains order-one.

**Definition 3.7** (Stable feature learning with LoRA). We say LoRA induces *stable feature learning* if the dynamics are stable (Definition 3.5), and $\Delta Z_B^t = \Theta(1)$ for all $t \le T$.

Intuitively, having $\Delta Z_B^t = \Theta(1)$ is associated with "maximal" adaptation in a similar fashion to $\mu$P for pretraining, see (Yang & Hu, 2020). By enforcing this condition, we derived *Maximal-Update Adaptation* scaling rules in the next section. Beyond stability, it is useful to understand *which* components of the update drive learning. Ideally, each of $\delta_t^1$, $\delta_t^2$, and $\delta_t^3$ should be $\Theta(1)$, indicating that both factors $A$ and $B$ contribute meaningfully within each step. Accordingly, in our main results we track the term-wise scalings as a diagnostic: if a term vanishes in the large-$(n, r)$ limit, the corresponding factor becomes effectively inactive.

Recall from Equation (1) that the feature update $\Delta Z_B^t$ decomposes into three terms $(\delta_t^i)_{i \in \{1,2,3\}}$. A sufficient condition for stability is to bound each term individually:

$$\delta_t^1 = \mathcal{O}(1), \qquad \delta_t^2 = \mathcal{O}(1), \qquad \delta_t^3 = \mathcal{O}(1).$$

This term-wise control avoids relying on cancellations among the three contributions. Stable feature learning additionally requires $\Delta Z_B^t = \Theta(1)$, which implies that at least one term must be $\Omega(1)$ under the chosen scaling, while the remaining terms may be lower-order.

## 4. Maximal-Update Adaptation ($\mu$A)

We now state the learning rate scaling rules that guarantee stable feature learning under Init[A] and Init[B]. Full proofs and derivations (including the scaling of each $\delta_t^i$) appear in Appendix A. Our analysis relies on two assumptions: (i) bounded forward and backward signals at each step, and (ii) a SignSGD update rule that uses the sign of the gradient, serving as a tractable proxy for the Adam optimizer. More broadly, our analysis is carried out at a physics level of rigor, omitting technical conditions that would otherwise make the derivations fully rigorous but unnecessarily complicated.

**Assumption 4.1** (Bounded forward/backward signals). For any fixed step $t \le T$, the layer input and the backpropagated output gradient satisfy

$$\underline{Z} = \Theta(1), \qquad d\bar{Z} := \frac{\partial \mathcal{L}}{\partial \bar{Z}} = \Theta(1).$$

This ensures that features and gradients neither explode nor vanish at any step as $(n, r) \to \infty$.

**Assumption 4.2** (Optimizer abstraction). We use a simplified variant of Adam (Kingma & Ba, 2015) without momentum, reducing it to SignSGD (Bernstein et al., 2018), which gives element-wise sign updates:

$$g_A^t := \text{sign}\left(\frac{\partial \mathcal{L}_t}{\partial A_{t-1}}\right), \qquad g_B^t := \text{sign}\left(\frac{\partial \mathcal{L}_t}{\partial B_{t-1}}\right),$$

and parameter updates

$$A_t = A_{t-1} - \eta \, g_A^t, \qquad B_t = B_{t-1} - \eta \, g_B^t,$$

where $\text{sign}(\cdot)$ is applied element-wise (with $\text{sign}(0) = 0$) and $\eta$ denotes the learning rate.

Following (Hayou et al., 2024a), we consider the setting where the loss is computed on a single sample. This simplifies the theoretical analysis and yields closed-form expressions. We further discuss the motivations behind these approximation assumptions and the extent to which the resulting scaling rules generalize beyond them in Appendix A.2.2. In what follows, we characterize the asymptotic behavior of feature updates as a function of $(n, r)$ and show that it depends on initialization schemes. We present the results for Init[A] and Init[B] and compare the differences. The derived scaling rules are empirically validated across a range of data modalities and training paradigms in Section 5. Since the individual sub-terms $\delta_t^1$, $\delta_t^2$, and $\delta_t^3$ are difficult to track reliably at that scale, we further verify their asymptotic scaling in a controlled toy experiment in Appendix G.

*Table 1.* $\mu$A scaling rules from Corollary 4.4 for common choices of $\alpha$. Each cell shows the argument of $\Theta(\cdot)$.

| $\gamma$ | $\alpha$ | $\eta$ | $Z_A^t$ |
|---|---|---|---|
| $0$ | $1$ | $n^{-1/2}r^{-1/2}$ | $n^{1/2}r^{-1/2}$ |
| $\frac{1}{2}$ | $r^{-1/2}$ | $n^{-1/2}r^{-1/4}$ | $n^{1/2}r^{-1/4}$ |
| $1$ | $r^{-1}$ | $n^{-1/2}$ | $n^{1/2}$ |

### 4.1. Init[A] Analysis

We begin with Init[A], where $A_0$ is randomly initialized and $B_0 = 0$. Theorem 4.3 characterizes the scale of the LoRA feature increment $\Delta Z_B^t$. Corollary 4.4 then yields the learning rate scaling that achieves stable feature learning.

**Theorem 4.3** (Init[A], Informal). *Under Assumptions 4.1 and 4.2, for any fixed $t \leq T$ with Init[A],*

$$\Delta Z_B^t \;=\; \max\{\Theta(\eta \alpha r),\; \Theta(\eta^2 \alpha r n)\}.$$

Imposing the stable feature learning condition $\Delta Z_B^t = \Theta(1)$ yields the following scaling rule for learning rate $\eta$.

**Corollary 4.4** ($\mu$A scaling rules for Init[A], Informal). *Let $\alpha = r^{-\gamma}$ for $\gamma \in [0, 1]$. Under Init[A], achieving stable feature learning (i.e., $\Delta Z_B^t = \Theta(1)$ for fixed $t \leq T$) requires*

$$\eta = \Theta(n^{-1/2}r^{-(1-\gamma)/2}).$$

*With this choice,*

$$\delta_t^1 = \Theta(1),\; \delta_t^2 = \Theta(1),\; \delta_t^3 = \Theta(1),$$

*and the intermediate feature scales as*

$$Z_A^t = \Theta(n^{1/2}\, r^{-(1-\gamma)/2}).$$

To make the prescriptions concrete, we summarize the resulting scalings for common multiplier choices in Table 1.

**Interpretation of Corollary 4.4.** We highlight the following key insights drawn from the corollary.

*Learning rate scaling with rank.* For a fixed model width $n$, the relationship between the optimal learning rate and rank $r$ depends on the choice of $\alpha$. With $\alpha = 1$, the learning rate decreases as $\eta \propto r^{-1/2}$: increasing the rank by $4\times$ requires halving the learning rate, consistent with Figure 2a. In contrast, setting $\alpha = r^{-1}$ absorbs the rank dependence, yielding a rank-invariant learning rate that depends only on width, consistent with Figure 2b.

*Intermediate feature scaling.* The intermediate features $Z_A^t$ grow with width $n$ but shrink with rank when $\alpha \in \{1, r^{-1/2}\}$. For $\alpha = 1$, doubling the rank reduces the intermediate feature scale by $\sqrt{2}$, which can improve internal

numerical stability in wide models where large intermediate activations have been identified as a source of training instability (Hayou et al., 2024a). However, when $\alpha = r^{-1}$, increasing the rank no longer suppresses the intermediate scale, which may explain why Init[A] with $\alpha = r^{-1}$ exhibits a narrower stable learning rate range in practice.

*Learning efficiency.* Under the stable learning rate scaling, all three contributions are order-one. This balance holds uniformly across all standard choices of $\alpha$ (i.e., all $\gamma \in [0, 1]$), so updates to factors $A$ and $B$ both contribute meaningfully to the feature update at every step, with neither factor reducing to a fixed random projection during training.

### 4.2. Init[B] Analysis

We next consider Init[B], where $B_0$ is randomly initialized and $A_0 = 0$.

**Theorem 4.5** (Init[B], Informal). *Under Assumptions 4.1 and 4.2, for any fixed $t \leq T$ with Init[B],*

$$\Delta Z_B^t \;=\; \max\{\Theta(\eta \alpha n),\; \Theta(\eta^2 \alpha n r)\}.$$

Unlike Init[A], the resulting learning rate prescription depends more sensitively on how $\alpha$ scales with $r$. In the main text we focus on the most common configuration $\alpha = 1$ and defer additional cases to Appendix A.5.

**Corollary 4.6** ($\mu$A rules for Init[B] with $\alpha = 1$, Informal). *Let $\alpha = 1$. Under Init[B], achieving stable feature learning (i.e., $\Delta Z_B^t = \Theta(1)$ for fixed $t \leq T$) requires*

$$\eta = \Theta(n^{-1}).$$

*With this choice,*

$$\delta_t^1 = \Theta(1),\; \delta_t^2 = \Theta(rn^{-1}),\; \delta_t^3 = \Theta(r^{1/2}n^{-1}),$$

*and the intermediate feature scales as*

$$Z_A^t = \Theta(1).$$

**Interpretation of Corollary 4.6.** This corollary reveals a regime qualitatively different from Init[A].

*Rank-invariant learning rate.* With $\alpha = 1$, the required learning rate scales only with width ($\eta \propto n^{-1}$) and is independent of rank. This explains the roughly rank-invariant optimal learning rate observed under Init[B] in Figure 2c.

*Connection to FFT.* The same width scaling $\eta \propto n^{-1}$ arises for full finetuning of a linear layer under the same SignSGD abstraction (see Theorem A.22 in Appendix A.6). This matching shows a necessary condition for learning rate transfer between LoRA and FFT: a learning rate tuned for Init[B] with $\alpha = 1$ can serve as a reasonable starting point for FFT without re-tuning. We empirically verify this transferability in our experiments.

*Intermediate feature scaling.* The intermediate features $Z_A^t$ remain order-one throughout training, keeping internal activations controlled regardless of width.

*Learning efficiency.* The term-wise contributions are imbalanced: here $\delta_t^1 = \Theta(1)$ while $\delta_t^2$ and $\delta_t^3$ vanish when $r \ll n$. When the rank is much smaller than the width, updates to $B$ contribute negligibly, and the adapter behaves as though $B$ were fixed at its random initialization, with learning occurring primarily through $A$. This may also explain why we typically observe Init[A] outperforming Init[B] at small ranks, where the imbalance is most pronounced. Nonetheless, increasing the rank can strengthen $\delta_t^2$; in the extreme case where $r$ is on the order of $n$, both factors contribute meaningfully to feature updates.

## 5. Experiments

Our experiments verify $\mu$A scaling rules empirically and learning rate transfer from LoRA to FFT. We examine three regimes: supervised finetuning (SFT) across diverse architectures, reinforcement learning with verifiable rewards (RLVR), and text-to-image diffusion model finetuning [2].

### 5.1. Experimental Setup

**Unified finetuning recipe.** We use a unified finetuning recipe across all experiments. Specifically, we use AdamW with weight decay of $0.01$ and global-norm gradient clipping of $1.0$. The learning rate warms up linearly over the first $5\%$ of training steps to its peak ($\eta$), then follows a cosine decay to $0.1\times$ the peak over the remaining steps.

**SFT setting.** We experiment with five public base models:

*Decoder-only language models.* For instruction tuning, we tune Llama-3.2-1B (Grattafiori et al., 2024) on the Tulu-3 supervised finetuning mixture (Lambert et al., 2024). For long-context reasoning, we train Qwen2.5-3B-Instruct (Qwen et al., 2025) on the OpenThoughts-114k (Guha et al., 2025).

*Encoder-only models.* We finetune RoBERTa-large (Liu et al., 2019) on the ANLI benchmark (Nie et al., 2020) and adapt ViT-Huge/14 (Dosovitskiy et al., 2021) to ImageNet-1K (Deng et al., 2009). Following prior work (Guo et al., 2024), we fix the learning rate of the classification head at $10^{-3}$ and sweep only the LoRA learning rate.

*Vision–language model.* For multimodal instruction following, we finetune Qwen3-VL-2B-Instruct (Bai et al., 2025) on LLaVA-Instruct-Mix, containing image–instruction pairs.

**RLVR setting.** We adapt Llama-3.1-8B (Grattafiori et al., 2024) to GSM8k (Cobbe et al., 2021) with GRPO (Shao

et al., 2024). The reward is based on answer correctness and format compliance.

**Text-to-image diffusion model finetuning.** We finetune Stable-Diffusion-v1.5 (Rombach et al., 2022) on the Naruto-BLIP-Captions (Cervenka, 2022) dataset.

**Evaluation protocol.** We compare three LoRA configurations: Init[A] with $\alpha = 1$, Init[A] with $\alpha = r^{-1}$, and Init[B] with $\alpha = 1$. For each configuration, we sweep learning rate $\eta$ (peak learning rate) on a $\log_2$-scale grid with multiplicative steps of $2\times$ and select the best value independently for (i) final training loss (EMA-smoothed) and (ii) a task-specific validation metric. For RLVR, we select learning rates by final training reward; for diffusion models, by validation FID (Heusel et al., 2017). More details are in Appendix B. We report training-loss-based sweeps in the main text and defer the other results to Appendix C, D, E.

**Theory–experiment alignment.** Our scaling rules characterize the leading-order dependence of the stable learning rate on width $n$ and rank $r$ in the joint limit $(n, r) \to \infty$. In practice, however, finetuning at finite width and rank introduces two deviations from the asymptotic prediction.

*Finite-size effects.* When $(n, r)$ are finite, constant factors and lower-order terms can shift the optimal learning rate. These shifts are more pronounced for smaller ranks, where the large-$r$ asymptotics are less accurate.

*Discrete learning rate grid.* We sweep learning rates on a $\log_2$-spaced grid. The reported optimum is the best value among discrete points. When the loss landscape is flat near the true optimum, adjacent grid points perform similarly, and training noise may determine which one appears best.

Accordingly, we evaluate the scaling rules by qualitative trends rather than exact numerical agreement. For configurations predicted to be rank-invariant, we expect optimal learning rates to cluster within one grid step ($2\times$). For rank-dependent configurations, we expect the optimum to change monotonically with rank and to span multiple grid points.

### 5.2. SFT Results

**Init[A] with $\alpha = 1$ (Figures 3, 2a)** For a fixed base model (i.e., fixed width $n$), our theory in Table 1 predicts that the optimal learning rate scales as $\eta \propto r^{-1/2}$ with rank. This prediction is broadly supported by the experiments: as rank increases, the best-performing learning rate monotonically shifts toward smaller values. Since each figure plots $\log_2(\eta)$ on the X-axis, a $4\times$ increase in rank should shift the optimum leftward by one unit (i.e., a $2\times$ reduction in learning rate)—a pattern clearly visible across most models. We observe minor deviations on Qwen2.5-3B-Instruct, where both $r = 16$ and $r = 64$ attain their minimum loss at the

---

[2]Source code is publicly available in https://github.com/NanChanNN/muA-lora.

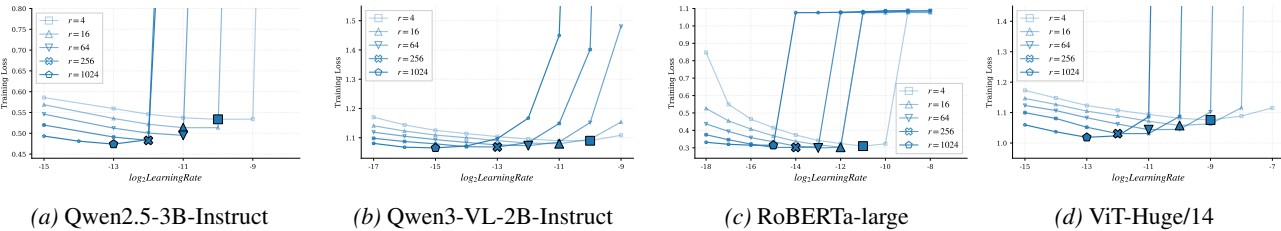

*(a)* Qwen2.5-3B-Instruct      *(b)* Qwen3-VL-2B-Instruct      *(c)* RoBERTa-large      *(d)* ViT-Huge/14

*Figure 3.* Learning rate sweeps for Init[A] with $\alpha = 1$ across four models. Curves show final training loss (EMA-smoothed) versus log-scale learning rate ($\log_2(\eta)$). Large markers denote per-rank optima. The Y-axis is clipped for readability.

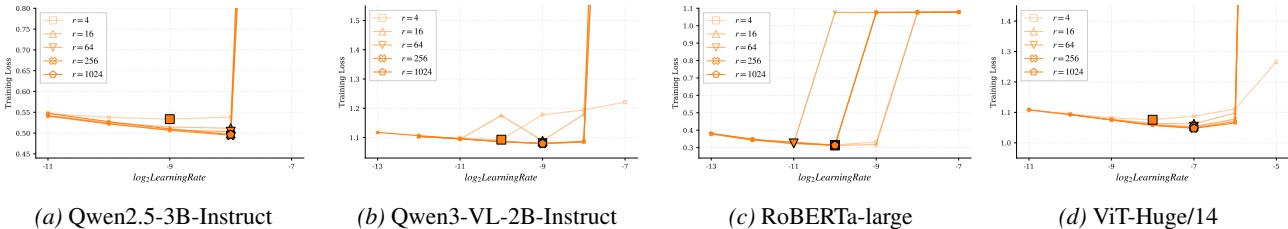

*(a)* Qwen2.5-3B-Instruct      *(b)* Qwen3-VL-2B-Instruct      *(c)* RoBERTa-large      *(d)* ViT-Huge/14

*Figure 4.* Learning rate sweeps for Init[A] with rank-dependent $\alpha = r^{-1}$ across four models. Curves show final training loss (EMA-smoothed) versus log-scale learning rate ($\log_2(\eta)$). Large markers denote per-rank optima. The Y-axis is clipped for readability.

same learning rate. We attribute this to the learning rate grid discretization: as shown in Figure 3a, the loss curves for $r = 4$ and $r = 16$ flatten near the optimum. Moreover, models finetuned with Init[A] and $\alpha = 1$ are sensitive to large learning rates: increasing the learning rate by $2\times$ beyond the optimum can substantially degrade training loss or cause divergence, particularly for Qwen2.5-3B-Instruct and RoBERTa-large. While we report results for $\alpha = 1$ here, our theory holds for any rank-independent constant $\alpha$. In Appendix C.1, we further verify that the same scaling behavior persists for other constant $\alpha \in \{2, 4\}$.

**Init[A] with $\alpha = r^{-1}$ (Figures 4, 2b)** According to our theory, for a fixed base model, the optimal learning rate should be independent of LoRA rank under this setting. The experimental results support this prediction: for all models, the optimal learning rates across ranks fall within a narrow range (typically within $2\times$), consistent with rank-invariance. As with Init[A] and $\alpha = 1$, this small variation is attributable to the discrete learning rate grid. Compared to the $\alpha = 1$ setting, the optimal learning rates under $\alpha = r^{-1}$ are substantially larger. We also observe that smaller ranks (e.g., $r = 4$) usually favor slightly smaller learning rates than larger ranks—opposite to the trend observed with $\alpha = 1$. This likely reflects the finite-size effects: at smaller ranks, lower-order terms can exert a non-negligible influence on feature learning and suppress the optimal learning rate.

**Init[B] with $\alpha = 1$ (Figures 5, 2c)** In this case, the optimal learning rate should be independent of LoRA rank. The empirical results confirm this prediction: across all models, the best learning rates for different ranks cluster

tightly. The small variation likely stems from learning rate grid resolution and finite-size effects. Although both Init[B] with $\alpha = 1$ and Init[A] with $\alpha = r^{-1}$ exhibit rank-invariant optimal learning rates, our theory shows they differ in width scaling: $\eta \propto n^{-1}$ for the former versus $\eta \propto n^{-1/2}$ for the latter. For large $n$, Init[B] thus requires substantially smaller learning rates, as our experiments confirm. We also observe that Init[B] with $\alpha = 1$ is more tolerant of larger learning rates. This stability is consistent with the theoretical finding that intermediate features remain order-one under Init[B], preventing internal activations from growing with width.

Crucially, our theory reveals that FFT shares the same learning rate scaling $\eta \propto n^{-1}$ with Init[B] with $\alpha = 1$, suggesting that their optimal learning rates may align and leading to learning rate transfer between the two. Experiments support this conjecture: for most models, the optimal learning rates for Init[B] ($\alpha = 1$) align closely with those for FFT. The exception is RoBERTa-large, where FFT favors a slightly smaller learning rate than Init[B]. We attribute this gap to the additional classification head: although its learning rate is fixed, jointly training it with LoRA adapters may still perturb the optimal learning rate. Nevertheless, our experiments show that learning rates tuned for Init[B] with $\alpha = 1$ transfer effectively to FFT in most model architectures, with minimal performance degradation. In Appendix C.3, we provide preliminary evidence that the learning rate transfer extends to downstream-task performance.

### 5.3. RLVR Results

RLVR results are reported in Figure 6. The learning rate sweeps reveal patterns consistent with our findings under

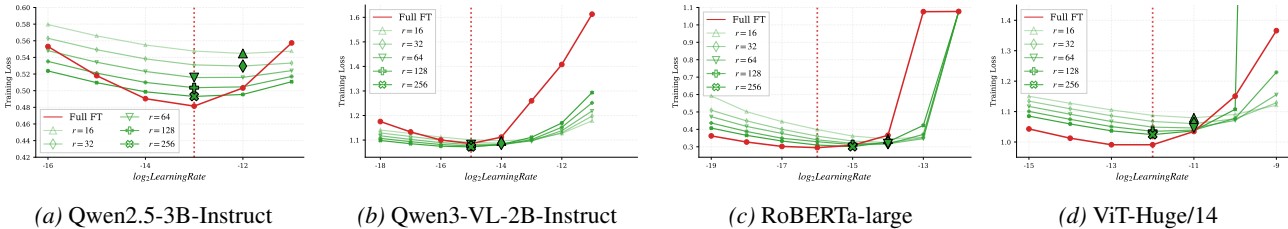

*(a)* Qwen2.5-3B-Instruct     *(b)* Qwen3-VL-2B-Instruct     *(c)* RoBERTa-large     *(d)* ViT-Huge/14

*Figure 5.* Learning rate sweeps for Init[B] with $\alpha = 1$ across four models. Curves show final training loss (EMA-smoothed) versus log-scale learning rate ($\log_2(\eta)$) for multiple ranks (green) and FFT (red). Large markers denote per-rank optima. The Y-axis is clipped for readability. The red vertical dashed line marks the optimal FFT learning rate.

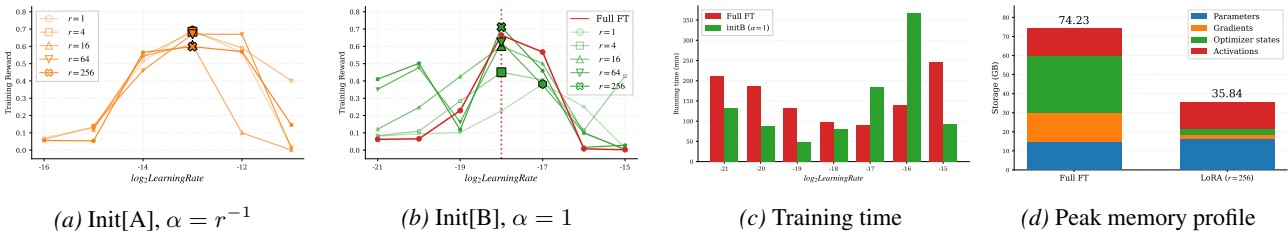

*(a)* Init[A], $\alpha = r^{-1}$     *(b)* Init[B], $\alpha = 1$     *(c)* Training time     *(d)* Peak memory profile

*Figure 6.* RLVR results on Llama-3.1-8B (GSM8k) with GRPO. (a)–(b) Learning rate sweeps plotting final training reward versus $\log_2(\eta)$. Large markers denote the best learning rate per rank; the red dashed line marks the FFT optimum. (c) Training time versus learning rate for FFT (red) and Init[B] with $\alpha=1$ and $r=256$ (green). (d) Peak GPU memory breakdown for FFT (left) and LoRA with $r=256$ (right).

**SFT.** Under Init[A] with $\alpha = r^{-1}$ (Figure 6a), the optimal learning rate is approximately rank-invariant: curves for different ranks peak at nearly the same learning rate. Similarly, under Init[B] with $\alpha = 1$ (Figure 6b), the optimal learning rate is rank-invariant and aligns with the FFT optimum, suggesting that learning rates tuned with LoRA can transfer seamlessly to FFT. The results of Init[A] with $\alpha = 1$ are deferred to Appendix D.1.

Compared with SFT, RL finetuning is more sensitive to learning rate selection. In SFT, a learning rate that is too small simply slows convergence; in RLVR, however, overly large or small learning rates substantially degrade training reward. Importantly, this sensitivity also manifests in training time. As shown in Figure 6c, the optimal learning rate incurs the shortest training time, while suboptimal rates can increase wall-clock time by 2–4×. The $\mu A$ scaling rules thus provide practical value beyond final performance: by identifying well-behaved learning rate regions in advance, practitioners can avoid time-consuming hyperparameter searches over suboptimal configurations. We provide a detailed investigation on this phenomenon in Appendix D.2.

Although LoRA and FFT have comparable per-step wall-clock times in our experiment, their memory footprints differ substantially. Figure 6d shows that FFT consumes over 2× the GPU memory of LoRA ($r=256$), with most overhead attributable to gradient buffers and optimizer states. This gap widens further at larger model sizes. Our theory thus enables a practical workflow that exploits this difference: one can sweep learning rates using Init[B] with $\alpha=1$

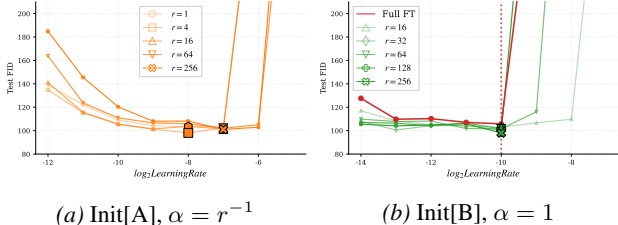

*(a)* Init[A], $\alpha = r^{-1}$     *(b)* Init[B], $\alpha = 1$

*Figure 7.* Results on Stable-Diffusion-1.5 (naruto-blip-captions). Subfigures plot validation FID score versus log-scale learning rate ($\log_2(\eta)$). Large markers denote the best learning rate for each rank; the red vertical dashed line marks the FFT optimum.

on mid-tier GPUs (e.g., A6000), which are more available and cost-effective, then transfer the optimal rate directly to FFT on higher-capacity hardware with minimal re-tuning.

### 5.4. Text-to-Image Diffusion Fine-Tuning Results

Figure 7 summarizes the results for finetuning a stable diffusion model. The learning rate sweeps show patterns consistent with our theory: under Init[A] with $\alpha = r^{-1}$, the optimal learning rate is rank-invariant, while under Init[B] with $\alpha = 1$, the optimal learning rate remains independent of rank and aligns with that of FFT. Notably, FFT is more sensitive to learning rate selection on diffusion models, exhibiting a sharp increase in validation FID at higher learning rates. A natural extension of our single-layer analysis is to interpret the $\mu A$ scaling rules locally, applying them per linear projection with its own width and rank. Under this view, the UNet's heterogeneous widths imply that no single global

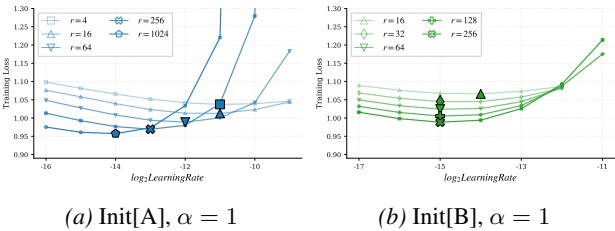

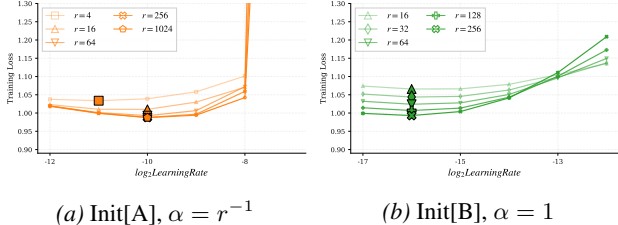

| *(a)* Init[A], $\alpha = 1$ | *(b)* Init[B], $\alpha = 1$ |
| --- | --- |

| *(a)* Init[A], $\alpha = r^{-1}$ | *(b)* Init[B], $\alpha = 1$ |
| --- | --- |

*Figure 8.* Learning rate sweeps under the SignSGD optimizer on Llama-3.2-1B (Tulu3) across two LoRA configurations. Each panel plots final training loss (EMA-smoothed) versus log-scale learning rate $\log_2(\eta)$ for multiple LoRA ranks. Large markers denote per-rank optima. The y-axis is clipped for readability.

*Figure 9.* Learning rate sweeps under a constant learning rate schedule on Llama-3.2-1B (Tulu3) across two LoRA configurations. Each panel plots final training loss (EMA-smoothed) versus log-scale learning rate $\log_2(\eta)$ for multiple LoRA ranks. Large markers denote per-rank optima. Y-axis clipped for readability.

learning rate is simultaneously optimal across all weight matrices, which may explain FFT's sensitivity here. Additional results for Init[A] with $\alpha = 1$ are in Appendix E.1.

### 5.5. Ablation study

**SignSGD versus AdamW.** Our theoretical derivations adopt SignSGD as a tractable abstraction of Adam (Assumption 4.2) because SignSGD preserves the property of Adam most relevant to our scaling analysis: *element-wise gradient normalization*, under which the effective update magnitude is $\Theta(1)$ with respect to width and rank, regardless of the raw gradient scale. We hypothesize that Adam's momentum and second-moment estimates primarily smooth the per-element updates without altering this $\Theta(1)$ scaling, and therefore expect the scaling rules derived under SignSGD to remain valid under Adam. To verify this hypothesis empirically, we replace AdamW with SignSGD and rerun the learning rate sweeps on Llama-3.2-1B (Tulu3). As shown in Figure 8, the scaling patterns under SignSGD match those under AdamW: under Init[A] with $\alpha = 1$, the optimal learning rate decreases with rank as $\eta \propto r^{-1/2}$, and under Init[B] with $\alpha = 1$, it remains rank-invariant. Additional results are provided in Appendix F.1. These observations confirm our hypothesis and support SignSGD as a tractable and empirically justified proxy for Adam in our derivations.

**Constant learning-rate schedule.** Our main experiments employ a linear warmup with cosine decay learning rate schedule, following common practice. To assess the robustness of our scaling rules to the choice of learning rate schedule, we replace this schedule with a constant learning rate and repeat the learning rate sweeps on Llama-3.2-1B (Tulu3). As shown in Figure 9, the scaling patterns under the constant schedule match those under the original schedule: the optimal learning rate remains rank-invariant under Init[A] with $\alpha = r^{-1}$ and Init[B] with $\alpha = 1$, though at different magnitudes. This consistency supports the robustness of our scaling rules across learning rate schedules.

## 6. Conclusion and Limitations

This work proposes $\mu$A (Maximal-Update Adaptation), a theoretical framework for LoRA finetuning that characterizes how the optimal learning rate scales with model width and adapter rank to achieve maximal feature updates. Using this framework, we identify a LoRA configuration that enables learning rate transfer between LoRA and FFT. Comprehensive experiments validate the $\mu$A scaling rules and demonstrate effective learning rate transfer in practice.

Our work has two main limitations. First, our theory is asymptotic and can be sensitive to noise for finite $(n, r)$. Second, our analysis establishes a necessary condition for learning rate transfer; future work is needed to identify and prove sufficient conditions for a more complete theory. Nevertheless, our experiments show strong agreement between theoretical predictions and empirical results.

## Acknowledgment

SV and NC were supported by the NSF-Simons Collaboration on the Mathematical Theory of Deep Learning (NSF DMS 2031985). SV is also funded by NSF CAREER 2339682 and NSF BSF 2430292. This work used GPU servers at Delta AI (NCSA) through allocation #CIS250883 from the Advanced Cyberinfrastructure Coordination Ecosystem: Services & Support (ACCESS) program (Boerner et al., 2023), which is supported by U.S. National Science Foundation grants #2138259, #2138286, #2138307, #2137603, and #2138296.

## Impact Statement

This work aims to advance machine learning research by improving our understanding of learning rate scaling and hyperparameter transfer in parameter-efficient finetuning methods. Its primary societal benefit is a reduction in the computational cost of hyperparameter tuning: by providing principled scaling rules, practitioners can avoid expensive learning rate sweeps when changing LoRA ranks or transitioning between LoRA and full finetuning. These efficiency gains might reduce energy consumption and the associated carbon footprint, while also lowering barriers to entry for researchers and practitioners with limited computational resources. We do not anticipate negative societal impacts beyond those generally associated with improved capabilities in large language and image generation models, and therefore do not discuss such impacts further here.

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

# A. Theoretical Proofs and Derivations for $\mu$A

**Purpose and scope.** This appendix provides the detailed derivations supporting the theories in the main text. We analyze the dynamics of LoRA finetuning in the limit where both the network width $n$ and the LoRA rank $r$ grow to infinity. Our first goal is to ensure that the output features remain bounded throughout finetuning—that is, they do not explode as $n$ and $r$ increase. Achieving this requires scaling the learning rate appropriately with $n$ and $r$. Our second goal is to ensure that the feature updates remain non-vanishing, so that learning proceeds efficiently and does not stall as the network grows.

## A.1. Preliminaries and Notation Recap

We begin by briefly restating the notation used in the main text to make the appendix self-contained.

### A.1.1. ASYMPTOTIC NOTATION AND SCALING CONVENTIONS

**Definition A.1** (Asymptotic notation). For sequences $c_{n,r} \in \mathbb{R}$ and $d_{n,r} \in \mathbb{R}_+$, we write $c_{n,r} = \mathcal{O}(d_{n,r})$ (resp. $c_{n,r} = \Omega(d_{n,r})$) if there exist constants $\kappa > 0$ and $N \in \mathbb{N}$ such that for all $n, r > N$ with $r \leq n$, we have $|c_{n,r}| < \kappa d_{n,r}$ (resp. $|c_{n,r}| > \kappa d_{n,r}$). We write $c_{n,r} = \Theta(d_{n,r})$ if both bounds hold. This notation extends element-wise to vectors: for $c_{n,r} = (c_{n,r}^i)_{i=1}^k \in \mathbb{R}^k$ and $d_{n,r} = (d_{n,r}^i)_{i=1}^k \in \mathbb{R}_+^k$, we write $c_{n,r} = \mathcal{O}(d_{n,r})$ when $c_{n,r}^i = \mathcal{O}(d_{n,r}^i)$ for all $i \in [k]$, and similarly for $\Omega$ and $\Theta$. For random variables, the notation is understood in the second-moment sense: $c_{n,r} = \mathcal{O}(d_{n,r})$ means $\left(\mathbb{E}|c_{n,r}|^2\right)^{1/2} = \mathcal{O}(d_{n,r})$.

**Fixed training regime.** All results are proved for a fixed training horizon $T$ that does not scale with $(n, r)$.

### A.1.2. LORA LAYER, FEATURES, AND THE PER-STEP DECOMPOSITION

**Definition A.2** (Low-Rank Adaptation (LoRA)). Let $W^\star \in \mathbb{R}^{n \times n}$ be a pretrained weight matrix that remains fixed during finetuning. LoRA parameterizes the finetuned weights as

$$W = W^\star + \alpha BA,$$

where $B \in \mathbb{R}^{n \times r}$ and $A \in \mathbb{R}^{r \times n}$ are trainable factors with rank $r \ll n$ and $\alpha \in \mathbb{R}$ is a scalar multiplier.

For a given layer augmented with a LoRA adaptor, we use $\underline{Z}$ and $\bar{Z}$ to denote the input and output of the layer, respectively. Specifically, we can write the layer operation as $\bar{Z} = W^*\underline{Z} + \alpha BA\underline{Z}$. In addition, we use $d\bar{Z}$ to denote the backpropagated gradient of the loss with respect to $\bar{Z}$, i.e., $d\bar{Z} = \partial\mathcal{L}/\partial\bar{Z}$. Next, we define the following intermediate and output features.

**Definition A.3** (LoRA features). For a layer input $\underline{Z}$, define the intermediate and output LoRA features

$$Z_A := A\underline{Z} \in \mathbb{R}^r, \qquad Z_B := \alpha BZ_A = \alpha BA\underline{Z} \in \mathbb{R}^n.$$

For $t \leq T$, we use superscripts to denote LoRA features at step $t$, i.e. $Z_A^t$, $Z_B^t$ and subscripts to denote weights at step $t$, i.e. $A_t$, $B_t$. To isolate the contribution of individual LoRA layers to feature learning, we assume that only a single LoRA layer is trainable while all other layers remain fixed. Under this assumption, the layer input $\underline{Z}$ remains unchanged across finetuning steps. After step $t$, the update to $Z_B$ decomposes as:

$$\Delta Z_B^t = \underbrace{\alpha B_{t-1}\Delta Z_A^t}_{\delta_t^1} + \underbrace{\alpha\Delta B_t Z_A^{t-1}}_{\delta_t^2} + \underbrace{\alpha\Delta B_t \Delta Z_A^t}_{\delta_t^3} \tag{2}$$

Next, we give the formal definition of the initialization schemes considered in this work.

**Definition A.4** (Initialization schemes for LoRA). To initialize the LoRA update as a no-op, we consider:

- **Init[A]:** $A_0$ has i.i.d. entries $\mathcal{N}(0, 1/n)$ and $B_0 = 0$.

- **Init[B]:** $B_0$ has i.i.d. entries $\mathcal{N}(0, 1/r)$ and $A_0 = 0$.

### A.1.3. STABILITY AND NON-VANISHING LEARNING CRITERIA

**Definition A.5** (Feature stability). Fix a finite number of finetuning steps $T$ independent of $(n, r)$. We say LoRA finetuning is *stable* if for all steps $t \leq T$, we have $Z_B^t = \mathcal{O}(1)$ in the joint limit $n, r \to \infty$ with $r \leq n$.

**Lemma A.6** (Telescoping bound). *Fix $t \leq T$. If $Z_B^0 = \mathcal{O}(1)$ and $\Delta Z_B^s = \mathcal{O}(1)$ for all $s \leq t$, then $Z_B^t = \mathcal{O}(1)$.*

*Proof.* By telescoping, $Z_B^t = Z_B^0 + \sum_{s=1}^t \Delta Z_B^s$. For a fixed $t$, this is a sum of $Z_B^0 = \mathcal{O}(1)$ and finitely many $\mathcal{O}(1)$ terms, hence $\mathcal{O}(1)$. $\square$

Since the LoRA adapter is typically initialized as a no-op (i.e., $Z_B^0 = 0$), Lemma A.6 implies that bounding the feature updates $\Delta Z_B^s = \mathcal{O}(1)$ is sufficient to ensure bounded features $Z_B^t = \mathcal{O}(1)$. Thus, verifying stability (Definition A.5) reduces to verifying that the updates remain bounded. However, to avoid trivial updates, we also require $\Delta Z_B^s$ to be of non-vanishing magnitude. This leads to the following definition:

**Definition A.7** (Stable feature learning with LoRA). We say LoRA induces *stable feature learning* if the dynamics are stable (Definition A.5), and $\Delta Z_B^t = \Theta(1)$ for all $t \leq T$.

As shown in Equation (2), the feature update $\Delta Z_B^t$ decomposes into three terms $(\delta_t^i)_{i \in \{1,2,3\}}$. A *sufficient* condition for feature stability is the term-wise bound $\delta_t^i = \mathcal{O}(1)$ for all $i \in \{1, 2, 3\}$, which avoids relying on cancellations between terms. Stable feature learning additionally requires $\Delta Z_B^t = \Theta(1)$; under the term-wise control above, this in particular implies that at least one term must satisfy $\delta_t^i = \Omega(1)$. Efficiency imposes a stronger requirement: each term must be exactly $\Theta(1)$, ensuring that both weight matrices $A$ and $B$ contribute meaningfully to the update. We formalize this as follows:

**Definition A.8** (Efficient Feature Learning with LoRA). We say that LoRA finetuning is efficient if it induces stable feature learning (Definition A.7), and for all finetuning steps $t \leq T$, we have:

$$\delta_t^i = \Theta(1), \quad i \in \{1, 2, 3\}.$$

## A.2. Technical Setup

### A.2.1. ASSUMPTIONS ON TRAINING DYNAMICS

**Assumption A.9** (Bounded forward/backward signals). For any fixed step $t \leq T$, the layer input and the backpropagated output gradient satisfy

$$\underline{Z} = \Theta(1), \qquad d\bar{Z} := \frac{\partial \mathcal{L}}{\partial \bar{Z}} = \Theta(1).$$

This ensures that features and gradients neither explode nor vanish at any step as $(n, r) \to \infty$.

**Assumption A.10** (Optimizer abstraction). We use a simplified variant of Adam (Kingma & Ba, 2015) without momentum, reducing it to SignSGD (Bernstein et al., 2018), which gives element-wise sign updates:

$$g_A^t := \text{sign}\left(\frac{\partial \mathcal{L}_t}{\partial A_{t-1}}\right), \qquad g_B^t := \text{sign}\left(\frac{\partial \mathcal{L}_t}{\partial B_{t-1}}\right),$$

and parameter updates

$$A_t = A_{t-1} - \eta\, g_A^t, \qquad B_t = B_{t-1} - \eta\, g_B^t,$$

where $\text{sign}(\cdot)$ is applied element-wise (with $\text{sign}(0) = 0$) and $\eta$ denotes the learning rate.

At a specific training step $t$, we would have:

$$\frac{\partial \mathcal{L}_t}{\partial A_{t-1}} = \alpha(B_{t-1}^\top d\bar{Z}^{t-1}) \otimes \underline{Z} \tag{3}$$

$$\frac{\partial \mathcal{L}_t}{\partial B_{t-1}} = \alpha d\bar{Z}^{t-1} \otimes Z_A^{t-1} \tag{4}$$

where $\otimes$ denotes outer product.

Following (Hayou et al., 2024a), we also consider the single-sample loss setting. This simplifies the theoretical analysis and leads to closed-form expressions.

A.2.2. DISCUSSION OF APPROXIMATIONS

To keep the analysis tractable and to obtain closed-form expressions, our derivations rely on several approximation assumptions. We discuss the motivation behind each below, together with the extent to which the resulting scaling rules are expected to remain valid in more general settings.

**SignSGD abstraction of Adam.** We adopt SignSGD as an abstraction of Adam because it preserves the property of Adam most relevant to our scaling analysis—*element-wise gradient normalization*—while admitting closed-form expressions (Bernstein et al., 2018; Hayou et al., 2024b). In Adam, the adaptive second-moment denominator rescales each gradient coordinate so that the effective update magnitude is $\Theta(1)$ with respect to width and rank, regardless of the raw gradient scale; SignSGD captures exactly this property and can be viewed as Adam without momentum. Adam's momentum and second-moment estimates additionally smooth the per-element updates, but we do not expect them to alter the $\Theta(1)$ scaling of each update coordinate. We thus expect the scaling rules derived under SignSGD to remain valid under Adam.

The SignSGD abstraction is specifically tailored to Adam-like optimizers; other optimizer families impose different gradient-normalization structures and would require separate abstractions. For instance, SGD uses raw unnormalized gradients, LARS (You et al., 2017) enforces a unit $\ell_2$-norm per layer, and Muon (Jordan et al., 2024) enforces a unit spectral norm. These distinct normalization structures will alter the asymptotic scaling of feature updates. Importantly, the $\mu$A framework is modular: Assumption A.10 can be replaced with an appropriate counterpart for other optimizer families, and the same analysis pipeline can be reused to derive the corresponding rules. We view this as a promising direction for future work.

**Single-sample loss.** We compute the loss on a single sample, following Hayou et al. (2024a). Because gradients of matrix-valued parameters are approximately low-rank in practice (Hu et al., 2022), a single-sample gradient is expected to capture the leading direction relevant to our scaling analysis while keeping the derivations closed-form.

**Single trainable layer.** We restrict attention to a single trainable LoRA layer to isolate the per-layer feature-learning dynamics, following standard practice in the literature (Hayou et al., 2024b;a; Li et al., 2025). Applying LoRA to multiple layers does introduce non-trivial cross-layer dynamics; nonetheless, for a network of fixed depth, our forward-stability argument extends layer by layer. Because the first adapted layer receives a fixed input, our scaling rules ensure that its LoRA update remains bounded over a fixed training horizon. As a result, the next layer continues to receive a $\Theta(1)$ input, and the same asymptotic analysis applies. This logic cascades, keeping forward signals controlled across the full network. Our multi-layer LoRA experiments are consistent with this intuition; a fully rigorous proof additionally requires controlling backward gradient signals across layers, which we leave for future work.

**Fixed training horizon.** We consider a fixed number of training steps $T$ that does not scale with $(n, r)$. This regime is natural for finetuning, which typically runs for a limited number of steps; analyzing the case where $T$ grows with $(n, r)$ would require tracking cumulative drift in the LoRA factors and is beyond the scope of this work.

A.2.3. AUXILIARY LEMMAS

This section contains auxiliary lemmas that are used as intermediate steps in the following analysis.

**Lemma A.11** (Recursive Formulas)**.** *For $t$ fixed, the asymptotic dynamics of $Z_A^t$ and $B_t$ follow the recursive formula:*

$$Z_A^t = \max(\Theta(Z_A^{t-1}), \Theta(\eta n)) \tag{5}$$
$$B_t = \max(\Theta(B_{t-1}), \Theta(\eta)) \tag{6}$$

*Proof.* Let $S^t = \alpha B_{t-1}^\top d\bar{Z}^{t-1}$. We, from Eq. 3, have:

$$g_A^t = \text{sign}(S^t \otimes \underline{Z})$$
$$= \text{sign}(S^t) \otimes \text{sign}(\underline{Z})$$

where in the second equation we use the fact that both $S^t$ and $\underline{Z}$ are both vectors. Hence, we obtain:

$$g_A^t \underline{Z} = (\text{sign}(S^t) \otimes \text{sign}(\underline{Z}))\underline{Z}$$
$$= (\text{sign}(\underline{Z})^\top \underline{Z}) \text{sign}(S^t)$$
$$= \Theta(n)$$

where we used the fact that $\text{sign}(\underline{Z})^\top \underline{Z} = \|\underline{Z}\|_1 = \Theta(n)$ and $\text{sign}(\cdot) = \Theta(1)$ due to the $\text{sign}$ operator. Combining, we obtain Eq. 5:

$$\begin{aligned}
Z_A^t &= \Theta((A_{t-1} - \eta g_A^t)\underline{Z}) \\
&= \Theta(Z_A^{t-1} - \eta g_A^t \underline{Z}) \\
&= \max(\Theta(Z_A^{t-1}), \Theta(\eta n))
\end{aligned}$$

Similarly, we have the following result for $B_t$:

$$\begin{aligned}
B_t &= \Theta(B_{t-1} - \eta g_B^t) \\
&= \Theta(B_{t-1} - \eta \, \text{sign}(\frac{\partial \mathcal{L}_t}{\partial B_{t-1}})) \\
&= \max(\Theta(B_{t-1}), \Theta(\eta))
\end{aligned}$$

$\square$

We take the next lemma from (Hayou & Liu, 2025), and we restate it and show its proof to make this proof self-contained.

**Lemma A.12** (Stein's lemma for a sign–Gaussian product). *Let $(Z, G)$ be a bi-variate centered Gaussian vector with* $\text{Var}(Z) = \text{Var}(G) = 1$ *and correlation* $\rho = \mathbb{E}[ZG]$. *Then*

$$\mathbb{E}[\text{sign}(Z)G] = \rho\sqrt{\frac{2}{\pi}}$$

*Proof.* For a pair of standard Gaussians with correlation $\rho$, we have:

$$G = \rho Z + \sqrt{1 - \rho^2} G'$$

where $G' \sim \mathcal{N}(0, 1)$ is another standard Gaussian independent of $Z$. Putting it into the expectation:

$$\begin{aligned}
\mathbb{E}[\text{sign}(Z)G] &= \mathbb{E}[\text{sign}(Z)(\rho Z + \sqrt{1 - \rho^2} G')] \\
&= \mathbb{E}[\rho \, \text{sign}(Z)Z] + \sqrt{1 - \rho^2} \cdot \mathbb{E}[\text{sign}(Z)G'] \\
&= \rho \mathbb{E}[\text{sign}(Z)Z] \\
&= \rho \mathbb{E}\,|Z|
\end{aligned}$$

Since $Z \sim \mathcal{N}(0, 1)$, $\mathbb{E}\,|Z| = \sqrt{\frac{2}{\pi}}$. Plugging this in, we obtain:

$$\mathbb{E}[\text{sign}(Z)G] = \rho\sqrt{\frac{2}{\pi}}$$

$\square$

**Corollary A.13** (General sign–Gaussian product). *Let $(Z, G)$ be a bi-variate centered Gaussian vector with* $\text{Var}(Z) = \sigma_Z^2$ *and* $\text{Var}(G) = \sigma_G^2$ *and correlation* $\rho = \frac{\mathbb{E}[ZG]}{\sigma_Z \sigma_G}$. *Then*

$$\mathbb{E}[\text{sign}(Z)G] = \rho \sigma_G \sqrt{\frac{2}{\pi}}$$

*Proof.* Define the standardized variables as:

$$U := \frac{Z}{\sigma_Z}, \quad V := \frac{G}{\sigma_G}$$

Then $(U, V)$ is a centered bi-variate Gaussian with

$$\text{Var}(U) = \text{Var}(V) = 1, \quad \rho_{UV} = Corr(U, V) = \mathbb{E}[UV] = \frac{\mathbb{E}[ZG]}{\sigma_Z \sigma_G} = \rho$$

Thus, $(U, V)$ satisfies the assumptions of Lemma A.12. Applying Lemma A.12 gives:

$$\mathbb{E}[\text{sign}(U)V] = \rho \sqrt{\frac{2}{\pi}}$$

Note that $\text{sign}(U) = \text{sign}(Z)$ since $\sigma_Z > 0$ and $V = \frac{G}{\sigma_G}$. Thus,

$$\begin{aligned}
\mathbb{E}[\text{sign}(Z)G] &= \mathbb{E}[\text{sign}(U)\sigma_G V] \\
&= \sigma_G \mathbb{E}[\text{sign}(U)V] \\
&= \rho \sigma_G \sqrt{\frac{2}{\pi}}
\end{aligned}$$

$\square$

With these tools in place, we can now analyze each contribution $\delta_t^i$, $i \in \{1, 2, 3\}$ to the feature increment $\Delta Z_B^t$ in turn.

### A.3. Scaling of Per-Step Update Terms

Recall from Eq. (2) that the per-step LoRA feature increment decomposes into three contributions. In this part we derive the asymptotic scaling of each term for a fixed step $t$, expressed in terms of the learning rate $\eta$, LoRA multiplier $\alpha$, network width $n$, rank $r$, and the current scales of $A_{t-1}$ and $B_{t-1}$.

**Lemma A.14** (Asymptotic Dynamics of $\delta_t^1$). *For step $t$ fixed, under Assumption A.9 and Assumption A.10, we have:*

$$\delta_t^1 = (-\eta\alpha) \cdot (\text{sign}(\underline{Z})^\top \underline{Z}) \cdot B_{t-1} \cdot \text{sign}(\alpha B_{t-1}^\top d\bar{Z}^{t-1}). \tag{7}$$

*Thus, the asymptotic dynamics of $\delta_t^1$ is determined by the term*

$$M_t := B_{t-1} \cdot \text{sign}(\alpha B_{t-1}^\top d\bar{Z}^{t-1}). \tag{8}$$

*For $r \leq n$, the proof uses the following two approximation regimes:*

(I) **Init[B] / random-initialization-dominated regime.** *If $B_{t-1}$ is approximated by its Gaussian initialization $B_0$ over the fixed training horizon, and if $B_0$ is treated as independent of $h := \alpha d\bar{Z}^{t-1}$, then*

$$M_t = \Theta\left(\sqrt{r + \frac{2}{\pi}\frac{r(r-1)}{n}}\right)\Theta(B_{t-1}) = \Theta(\sqrt{r})\Theta(B_{t-1}). \tag{9}$$

(II) **Init[A] / correlated low-rank-gradient regime.** *If the accumulated update in $B_{t-1}$ is dominated by correlated rank-one task-gradient directions, so that it may be modeled at leading order by a non-cancelling low-rank matrix (for example $B_{t-1} = \beta uv^\top$ with $u$ and $v$ having $\Theta(1)$ entries), then*

$$M_t = \Theta(r)\Theta(B_{t-1}). \tag{10}$$

*Consequently,*

$$\delta_t^1 = \begin{cases} \Theta(\eta\alpha nr)\,\Theta(B_{t-1}), & \text{Init[A], using the correlated low-rank-gradient branch,} \\ \Theta(\eta\alpha n\sqrt{r})\,\Theta(B_{t-1}), & \text{Init[B], using the random-initialization-dominated branch.} \end{cases} \tag{11}$$

*Proof.* Substituting $\Delta Z_A^t = -\eta g_A^t \underline{Z}$ into $\delta_t^1$, we have:

$$\delta_t^1 = \alpha B_{t-1} \Delta Z_A^t$$

$$= \alpha B_{t-1}(-\eta \cdot \text{sign}(\frac{\partial \mathcal{L}_t}{\partial A_{t-1}})\underline{Z})$$

$$= (-\eta\alpha) \cdot (\text{sign}(\underline{Z})^\top \underline{Z}) \cdot B_{t-1} \cdot \text{sign}(\alpha B_{t-1}^\top d\bar{Z}^{t-1})$$

where we used

$$\frac{\partial \mathcal{L}_t}{\partial A_{t-1}} = \alpha(B_{t-1}^\top d\bar{Z}^{t-1}) \otimes \underline{Z}$$

Since $\text{sign}(\underline{Z})^\top \underline{Z} = \|\underline{Z}\|_1 = \Theta(n)$, it remains to estimate

$$M_t = B_{t-1} \cdot \text{sign}(\alpha B_{t-1}^\top d\bar{Z}^{t-1})$$

The reduction to $M_t$ in Equation (8) is algebraic; the approximations enter in the two estimates of $M_t$ below.

**Regime (I): Init[B] / random-initialization-dominated approximation.** For Init[B], we have

$$B_{t-1} = B_0 - \eta \times \sum_{s=0}^{t-2} \text{sign}(d\bar{Z}^s) \otimes \text{sign}(Z_A^s)$$

up to the sign of the positive scalar $\alpha$. We know from (Hayou et al., 2024a) that with Init[B], the optimal learning rate $\eta$ scales as $1/n$ with width, as also confirmed by the stable Init[B] regime derived below. Consequently, over a fixed training horizon $T$, the entries of $B_t$ are expected to remain close to those of $B_0$. This motivates approximation regime (I) from the lemma statement: the update to $B_t$ is lower-order relative to the Gaussian initialization, so we approximate:

$$B_{t-1} \text{sign}(\alpha B_{t-1}^\top d\bar{Z}^{t-1}) \approx B_0 \text{sign}(\alpha B_0^\top d\bar{Z}^{t-1}).$$

We further model $B_0$ as an i.i.d. Gaussian matrix independent of $h := \alpha d\bar{Z}^{t-1}$. The next second-moment computation derives Equation (9).

For the ease of analysis, we drop the subscript in $B_{t-1}$ and instead use $B$ with $B = \Theta(\beta)$. In addition, we use $h = \alpha d\bar{Z}^{t-1} \in \mathbb{R}^n$ with $\Theta(h) = \Theta(\alpha)$. Let $X = B \cdot \text{sign}(B^T h)$, we aim to analyze the asymptotic dynamic of $X$ by taking an arbitrary element $X_k$ and analyze $\mathbb{E}[X_k^2]$. Based on our approximation, we treat $B \in \mathbb{R}^{n \times r}$ as an i.i.d. Gaussian matrix with entries $B_{ij} \overset{\text{i.i.d.}}{\sim} \mathcal{N}(0, \beta^2)$, and we assume $h \in \mathbb{R}^n$ is independent of $B$ with entries $h_i \overset{\text{i.i.d.}}{\sim} \mathcal{N}(0, \alpha^2)$.

Fixing an index $k$, we have:

$$X_k = \sum_{a=1}^r B_{ka} \cdot \text{sign}(u_a)$$

where $u_a = \sum_{j=1}^n B_{ja} h_j$. Then we have:

$$\mathbb{E}[X_k^2] = \mathbb{E}[\sum_{a=1}^r B_{ka}^2 \text{sign}(u_a)^2 + 2\sum_{a<a'} B_{ka} \text{sign}(u_a) B_{ka'} \text{sign}(u_{a'})]$$

$$= r\beta^2 + 2\sum_{a<a'} \mathbb{E}[B_{ka} \text{sign}(u_a) B_{ka'} \text{sign}(u_{a'})]$$

Given $h$, the pair $(u_a, B_{ka})$ is Gaussian with $\mathbb{E}[u_a \mid h] = 0$ and $\mathbb{E}[B_{ka} \mid h] = 0$, and the correlation is

$$\rho_k = Corr(B_{ka}, u_a \mid h)$$

$$= \frac{\mathbb{E}[B_{ka} u_a \mid h]}{\sqrt{\text{Var}[B_{ka} \mid h]\text{Var}[u_a \mid h]}}$$

$$= \frac{h_k \beta^2}{\sqrt{\beta^2 \cdot (\sum_{j=1}^n h_j^2 \beta^2)}}$$

$$= \frac{h_k}{\sqrt{\sum_{j=1}^n h_j^2}}$$

Using Corollary A.13, we have

$$\mu := \mathbb{E}[B_{ka} \operatorname{sign}(u_a) \mid h]$$

$$= \rho_k \beta \sqrt{\frac{2}{\pi}}$$

By the same argument, $\mathbb{E}[B_{ka'} \operatorname{sign}(u_{a'}) \mid h] = \rho_k \beta \sqrt{\frac{2}{\pi}}$. Moreover, conditionally on $h$, the random variables $B_{ka} \operatorname{sign}(u_a)$ and $B_{ka'} \operatorname{sign}(u_{a'})$ are independent for $a \neq a'$, since they depend on distinct columns of $B$. This yields:

$$\mathbb{E}[B_{ka} \operatorname{sign}(u_a) B_{ka'} \operatorname{sign}(u_{a'}) | h] = \rho_k^2 \beta^2 \frac{2}{\pi}$$

Therefore,

$$\mathbb{E}[B_{ka} \operatorname{sign}(u_a) B_{ka'} \operatorname{sign}(u_{a'})] = \beta^2 \frac{2}{\pi} \mathbb{E}[\rho_k^2]$$

Denote $h_k = \alpha q_k$ for all $k$ where $q_k \overset{\text{iid}}{\sim} \mathcal{N}(0, 1)$, then we have:

$$\mathbb{E}[\rho_k^2] = \mathbb{E}[\frac{q_k^2}{\sum_{j=1}^n q_j^2}]$$

Let $S = \sum_{j=1}^n q_j^2$. Given that $q_j$ are independent, standard normal random variables, we get $S \sim \chi^2(n)$. Furthermore, by symmetry, we have:

$$\sum_{k=1}^n \mathbb{E}[\frac{q_k^2}{S}] = \mathbb{E}[\frac{\sum_{k=1}^n q_k^2}{S}] = 1$$

Therefore, $\mathbb{E}[\rho_k^2] = \frac{1}{n}$, and we obtain:

$$\mathbb{E}[B_{ka} \operatorname{sign}(u_a) B_{ka'} \operatorname{sign}(u_{a'})] = \beta^2 \cdot \frac{2}{\pi} \cdot \frac{1}{n}$$

Plugging this term:

$$\mathbb{E}[X_k^2] = r\beta^2 + 2 \sum_{a < a'} \mathbb{E}[B_{ka} \operatorname{sign}(u_a) B_{ka'} \operatorname{sign}(u_{a'})]$$

$$= r\beta^2 + 2 \cdot \frac{r(r-1)}{2} \cdot \beta^2 \cdot \frac{2}{\pi} \cdot \frac{1}{n}$$

$$= \beta^2 (r + \frac{2}{\pi} \cdot \frac{r(r-1)}{n})$$

Hence,

$$X_k = \sqrt{\mathbb{E}[X_k^2]} = \Theta(\beta) \cdot \Theta(\sqrt{r + \frac{2}{\pi} \cdot \frac{r(r-1)}{n}})$$

This is the random-Gaussian second-moment estimate advertised in regime (I).

Since $r \leq n$,

$$X_k = \Theta(\beta \sqrt{r}).$$

Therefore,

$$M_t = \Theta(\sqrt{r}) \, \Theta(B_{t-1}) \qquad \text{under Init[B].}$$

**Regime (II): Init[A] / correlated low-rank-gradient approximation.** For Init[A], we have:

$$B_{t-1} = -\eta \times \sum_{s=0}^{t-2} \text{sign}(d\bar{Z}^s) \otimes \text{sign}(Z_A^s)$$

$$= -\eta \times \sum_{s=0}^{t-2} \text{sign}(d\bar{Z}^s) \, \text{sign}(Z_A^s)^\top$$

up to the sign of the positive scalar $\alpha$. $B_{t-1}$ is a finite sum of rank-one task gradients $\text{sign}(d\bar{Z}^s) \otimes \text{sign}(Z_A^s)$. For finetuning, we can assume that these rank-one task gradients are sufficiently correlated across a fixed training horizon, and thus this sum retains a low-rank structure even as $r$ grows. This motivates the approximation regime (II) from the lemma statement.

To see the resulting scaling, consider for instance the extreme case where $B_{t-1}$ is actually of rank 1, i.e. the task gradients are perfectly correlated. Under this case, we have the following rank-one approximation

$$B_{t-1} = \beta u v^\top,$$

where $u \in \mathbb{R}^n$ and $v \in \mathbb{R}^r$ have $\Theta(1)$ entries, and $\beta = \Theta(B_{t-1})$ is the entrywise scale. Therefore,

$$\text{sign}\big(\alpha B_{t-1}^\top d\bar{Z}^{t-1}\big) = \text{sign}\big(\alpha\beta v u^\top d\bar{Z}^{t-1}\big)$$

$$= \text{sign}\big(\alpha\beta u^\top d\bar{Z}^{t-1}\big) \cdot \text{sign}(v).$$

where $\text{sign}\big(\alpha\beta u^\top d\bar{Z}^{t-1}\big)$ is a scalar. Hence,

$$B_{t-1} \, \text{sign}\big(\alpha B_{t-1}^\top d\bar{Z}^{t-1}\big) = \beta u v^\top \cdot \text{sign}\big(\alpha\beta u^\top d\bar{Z}^{t-1}\big) \cdot \text{sign}(v)$$

$$= \beta \cdot \text{sign}\big(\alpha\beta u^\top d\bar{Z}^{t-1}\big) \cdot u \cdot \|v\|_1.$$

Since $\|v\|_1 = \Theta(r)$, we obtain

$$M_t = \Theta(\beta r) = \Theta(r)\,\Theta(B_{t-1}) \qquad \text{under Init[A]}.$$

**Putting things together.** We get

$$\delta_t^1 = \begin{cases} \Theta(\eta\alpha n r)\,\Theta(B_{t-1}), & \text{Init[A]}, \\ \Theta(\eta\alpha n\sqrt{r})\,\Theta(B_{t-1}), & \text{Init[B]}. \end{cases}$$

$\square$

**Lemma A.15** (Asymptotic Dynamics of $\delta_t^2$). *For $t$ fixed, the asymptotic dynamics of $\delta_t^2$ follows the following formula:*

$$\delta_t^2 = \Theta(\eta\alpha r) \cdot \Theta(Z_A^{t-1}) \tag{12}$$

*Proof.* Substituting $\Delta B_t = -\eta g_B^t$ and $g_B^t = \text{sign}(\frac{\partial \mathcal{L}_t}{\partial B_{t-1}})$ into $\delta_t^2$, we have:

$$\delta_t^2 = \alpha \Delta B_t Z_A^{t-1}$$

$$= \alpha(-\eta \cdot \text{sign}(\frac{\partial \mathcal{L}_t}{\partial B_{t-1}})) Z_A^{t-1}$$

$$= (-\eta\alpha)\,\text{sign}(\alpha d\bar{Z}^{t-1} \otimes Z_A^{t-1}) Z_A^{t-1}$$

$$= (-\eta\alpha)(\text{sign}(\alpha d\bar{Z}^{t-1}) \otimes \text{sign}(Z_A^{t-1})) Z_A^{t-1}$$

$$= (-\eta\alpha)(\text{sign}(Z_A^{t-1})^\top Z_A^{t-1}) \, \text{sign}(\alpha d\bar{Z}^{t-1})$$

Therefore, we can get the asymptotic behavior of $\delta_t^2$ as:

$$\delta_t^2 = \Theta(-\eta\alpha) \cdot \Theta(r) \cdot \Theta(Z_A^{t-1}) \cdot \Theta(1)$$

$$= \Theta(\eta\alpha r) \cdot \Theta(Z_A^{t-1})$$

where the extra $r$ comes from the fact that we are summing over $r$ positive terms in $\text{sign}(Z_A^{t-1})^\top Z_A^{t-1}$. $\square$

**Lemma A.16** (Asymptotic Dynamics of $\delta_t^3$). *For step $t$ fixed, under Assumption A.9 and Assumption A.10, we have:*

$$\delta_t^3 = (\alpha\eta^2) \cdot (\text{sign}(\underline{Z})^\top \underline{Z}) \cdot \left[ \text{sign}(Z_A^{t-1})^\top \text{sign}(\alpha B_{t-1}^\top d\bar{Z}^{t-1}) \right] \cdot \text{sign}(\alpha d\bar{Z}^{t-1}) \tag{13}$$

*Thus, the asymptotic dynamics of $\delta_t^3$ is determined by the term*

$$\Gamma_t := \text{sign}(Z_A^{t-1})^\top \text{sign}(\alpha B_{t-1}^\top d\bar{Z}^{t-1}). \tag{14}$$

*The proof uses the following two approximation regimes.*

(I) ***Init[B] / weak rank-space sign-alignment regime.*** *If $B_{t-1}$ is approximated by its Gaussian initialization $B_0$ over the fixed training horizon, and if*

$$Z_A^{t-1} \approx -\eta(\text{sign}(\underline{Z})^\top \underline{Z}) \sum_{s=1}^{t-1} \text{sign}(\alpha B_0^\top d\bar{Z}^{s-1}), \tag{15}$$

*then $\Gamma_t$ is approximated by the rank-space sign overlap*

$$\Gamma_t \approx \text{sign}\left( -\sum_{s=1}^{t-1} \text{sign}(\alpha B_0^\top d\bar{Z}^{s-1}) \right)^\top \text{sign}(\alpha B_0^\top d\bar{Z}^{t-1}). \tag{16}$$

*Under the weak-alignment approximation that these rank-space sign vectors behave like weakly aligned random signs, this inner product has random-sign-sum scaling*

$$\Gamma_t = \Theta(r^{1/2}). \tag{17}$$

(II) ***Init[A] / non-cancelling sign-overlap regime.*** *Under Init[A], we have*

$$\Gamma_t = \text{sign}(Z_A^{t-1})^\top \text{sign}\left( -\alpha\eta \sum_{s=1}^{t-1} c_{s,t} \, \text{sign}(Z_A^{s-1}) \right), \quad c_{s,t} := \text{sign}(\alpha d\bar{Z}^{s-1})^\top d\bar{Z}^{t-1}. \tag{18}$$

*If this empirical rank-space sign overlap does not cancel, namely*

$$\frac{1}{r} \text{sign}(Z_A^{t-1})^\top \text{sign}\left( -\alpha\eta \sum_{s=1}^{t-1} c_{s,t} \, \text{sign}(Z_A^{s-1}) \right) = \Theta(1), \tag{19}$$

*then*

$$\Gamma_t = \Theta(r). \tag{20}$$

*Consequently,*

$$\delta_t^3 = \begin{cases} \Theta(\alpha\eta^2 nr), & \textit{Init[A], using the non-cancelling sign-overlap branch,} \\ \Theta(\alpha\eta^2 nr^{1/2}), & \textit{Init[B], using the weak rank-space sign-alignment branch.} \end{cases} \tag{21}$$

*Proof.* Substituting $\Delta B_t = -\eta g_B^t$ and $\Delta Z_A^t = -\eta g_A^t \underline{Z}$ into $\delta_t^3$, we have:

$$\begin{aligned} \delta_t^3 &= \alpha \Delta B_t \Delta Z_A^t \\ &= \alpha \cdot (-\eta g_B^t) \cdot (-\eta g_A^t \underline{Z}) \\ &= (\alpha\eta^2) \cdot \text{sign}(\alpha d\bar{Z}^{t-1} \otimes Z_A^{t-1}) \cdot \text{sign}(\alpha(B_{t-1}^\top d\bar{Z}^{t-1}) \otimes \underline{Z}) \cdot \underline{Z} \\ &= (\alpha\eta^2) \cdot (\text{sign}(\underline{Z})^\top \underline{Z}) \cdot \text{sign}(\alpha d\bar{Z}^{t-1} \otimes Z_A^{t-1}) \cdot \text{sign}(\alpha B_{t-1}^\top d\bar{Z}^{t-1}) \\ &= (\alpha\eta^2) \cdot (\text{sign}(\underline{Z})^\top \underline{Z}) \cdot \left[ \text{sign}(Z_A^{t-1})^\top \text{sign}(\alpha B_{t-1}^\top d\bar{Z}^{t-1}) \right] \cdot \text{sign}(\alpha d\bar{Z}^{t-1}) \end{aligned}$$

where $\text{sign}(\underline{Z})^\top \underline{Z} = \Theta(n)$ and $\text{sign}(\alpha d\bar{Z}^{t-1}) = \Theta(1)$ entrywise. The only additional rank-dependent term is:

$$\Gamma_t := \text{sign}(Z_A^{t-1})^\top \text{sign}(\alpha B_{t-1}^\top d\bar{Z}^{t-1}).$$

The reduction to $\Gamma_t$ is algebraic. The approximation choices enter through the estimates of $\Gamma_t$ below.

**Regime (II): Init[A] / non-cancelling sign-overlap approximation** Here $B_0 = 0$ and $Z_A^0 = A_0 \underline{Z}$ is random. Unrolling the SignSGD update for $B$ gives the exact identity

$$B_{t-1} = -\eta \sum_{s=1}^{t-1} \text{sign}(\alpha d\bar{Z}^{s-1}) \, \text{sign}(Z_A^{s-1})^\top.$$

Therefore,

$$B_{t-1}^\top d\bar{Z}^{t-1} = -\eta \sum_{s=1}^{t-1} c_{s,t} \cdot \text{sign}(Z_A^{s-1}), \qquad c_{s,t} := \text{sign}(\alpha d\bar{Z}^{s-1})^\top d\bar{Z}^{t-1}.$$

Equivalently, the exact expression for the rank-dependent term is

$$\Gamma_t = \text{sign}(Z_A^{t-1})^\top \, \text{sign}\left(-\alpha\eta \sum_{s=1}^{t-1} c_{s,t} \, \text{sign}(Z_A^{s-1})\right).$$

The approximation enters when estimating the empirical sign overlap above.

In particular, at the first non-trivial step after $B$ has been updated,

$$B_1 = -\eta \, \text{sign}(\alpha d\bar{Z}^0) \, \text{sign}(Z_A^0)^\top,$$

So,

$$B_1^\top d\bar{Z}^1 = -\eta c_{1,2} \cdot \text{sign}(Z_A^0).$$

Since $B_0 = 0$, the first update to $A$ vanishes and $Z_A^1 = Z_A^0$. Therefore,

$$\begin{aligned}
\Gamma_2 &= \text{sign}(Z_A^1)^\top \, \text{sign}\left(\alpha B_1^\top d\bar{Z}^1\right) \\
&= \text{sign}(-\alpha\eta c_{1,2}) \cdot \text{sign}(Z_A^1)^\top \, \text{sign}(Z_A^0) \\
&= \text{sign}(-\alpha\eta c_{1,2}) \cdot \|\text{sign}(Z_A^0)\|_2^2
\end{aligned}$$

Since $Z_A^0$ has no zero coordinates almost surely and $c_{1,2} \neq 0$ generically, we obtain

$$\Gamma_2 = \Theta(r)$$

For a fixed later step $t$, the vector $B_{t-1}^\top d\bar{Z}^{t-1}$ is a finite linear combination of previous rank-space signs $\text{sign}(Z_A^{s-1})$. We use the correlated-gradient, non-cancellation approximation that these rank-space signs maintain non-vanishing average signed alignment with the current sign vector $\text{sign}(Z_A^{t-1})$, namely

$$\frac{1}{r}\left[\text{sign}(Z_A^{t-1})^\top \, \text{sign}\left(-\alpha\eta \sum_{s=1}^{t-1} c_{s,t} \, \text{sign}(Z_A^{s-1})\right)\right] = \Theta(1).$$

This is the approximation in the Init[A] branch: the empirical rank-space sign overlap is assumed not to cancel. Equivalently,

$$\Gamma_t = \Theta(r).$$

Consequently,

$$\delta_t^3 = \Theta(\alpha\eta^2 nr) \qquad \text{under Init[A].}$$

**Regime (I): Init[B] / weak rank-space sign-alignment approximation.** Here $A_0 = 0$ and $B_0$ is randomly initialized. Unrolling the SignSGD update for $B$ gives the exact identity

$$B_{t-1} = B_0 - \eta \times \sum_{s=1}^{t-1} \text{sign}(\alpha d\bar{Z}^{s-1}) \otimes \text{sign}(Z_A^{s-1})$$

In the random-initialization-dominated branch in Lemma A.14, we approximate $B_{s-1}$ by $B_0$ over the fixed training horizon. This approximation is self-consistent with the stable Init[B] scaling derived below: for the main Init[B], $\alpha = 1$ regime,

$\eta = \Theta(n^{-1})$, so the finite sum of the updates is lower order entrywise compared to the Gaussian initialization scale of $B_0$. Thus,

$$\text{sign}(\alpha B_{t-1}^\top d\bar{Z}^{t-1}) \approx \text{sign}(\alpha B_0^\top d\bar{Z}^{t-1}).$$

Moreover, since $A_0 = 0$, unrolling the update for $A$ gives

$$Z_A^{t-1} = -\eta(\text{sign}(\underline{Z})^\top \underline{Z}) \sum_{s=1}^{t-1} \text{sign}(\alpha B_{s-1}^\top d\bar{Z}^{s-1}).$$

Using the same random-initialization-dominated approximation $B_{s-1} \approx B_0$, we obtain

$$Z_A^{t-1} \approx -\eta(\text{sign}(\underline{Z})^\top \underline{Z}) \sum_{s=1}^{t-1} \text{sign}(\alpha B_0^\top d\bar{Z}^{s-1})$$

Therefore,

$$\Gamma_t = \text{sign}(Z_A^{t-1})^\top \text{sign}(\alpha B_{t-1}^\top d\bar{Z}^{t-1})$$

$$\approx \text{sign}\left(\sum_{s=1}^{t-1} \text{sign}(\alpha B_0^\top d\bar{Z}^{s-1})\right)^\top \text{sign}(\alpha B_0^\top d\bar{Z}^{t-1})$$

where the omitted global sign $\text{sign}(-\eta(\text{sign}(\underline{Z})^\top \underline{Z}))$ has no effect on the magnitude.

The additional approximation here is weak rank-space sign-alignment: the rank-space sign vectors induced by the Gaussian columns of $B_0$ do not remain coherently aligned across time. Hence, their inner product has random-sign-sum scaling:

$$\text{sign}\left(\sum_{s=1}^{t-1} \text{sign}(\alpha B_0^\top d\bar{Z}^{s-1})\right)^\top \text{sign}(\alpha B_0^\top d\bar{Z}^{t-1}) = \Theta(r^{1/2}).$$

Under this approximation,

$$\Gamma_t = \Theta(r^{1/2}).$$

Consequently,

$$\delta_t^3 = \Theta(\alpha\eta^2 nr^{1/2}) \qquad \text{under Init[B]}.$$

*Remark (strong-alignment alternative).* The Init[B] theorem and corollaries below build on the preceding Init[B] estimate, which in turn relies on the weak rank-space sign-alignment approximation. If the output-gradient directions are instead strongly aligned across time steps, then the rank-space sign overlap may scale as

$$\Gamma_t = \Theta(r),$$

which gives

$$\delta_t^3 = \Theta(\alpha\eta^2 nr).$$

Nevertheless, this stronger-alignment alternative does not change the main conclusions on the learning-rate scaling of Init[B], since the resulting $\delta_t^3$ term is of the same order as the existing $\delta_t^2 = \Theta(\eta^2 \alpha nr)$ contribution and therefore does not introduce a larger term in $\Delta Z_B^t$.

**Putting things together.** We get

$$\delta_t^3 = \begin{cases} \Theta(\alpha\eta^2 nr), & \text{Init[A]}, \\ \Theta(\alpha\eta^2 nr^{1/2}), & \text{Init[B]}. \end{cases}$$

$\square$

With the foregoing lemmas established, the proof of the main theorems becomes straightforward: for each initialization scheme, we (i) identify the induced initial scales of $(Z_A^0, B_0)$, (ii) propagate these scales through Lemma A.11, (iii) plug into Lemma A.14, Lemma A.15, and Lemma A.16, and (iv) derive the learning rate scaling that yields stable, non-vanishing feature updates $\Delta Z_B^t$, i.e., $\Delta Z_B^t = \Theta(1)$.

## A.4. Init[A]: Theorems and Corollaries

This section provides the derivations for the Init[A] regime in which $A_0$ is randomly initialized and $B_0 = 0$. We follow the same structure as the main text: first characterize the scale of $\Delta Z_B^t$, then impose the stable feature learning condition to obtain the learning rate scaling.

**Theorem A.17** (Init[A]). *Under Assumptions A.9 and A.10, for any fixed $t \leq T$ with Init[A],*

$$\Delta Z_B^t \;=\; \max\{\Theta(\eta \alpha r), \; \Theta(\eta^2 \alpha r n)\}. \tag{22}$$

*Proof.* With Init[A], we have $B_0 = 0$ and $Z_A^0 = A_0 \underline{Z} = \Theta(1)$. Thus, for all $t$, we have:

$$Z_A^t = \max(\Theta(1), \Theta(\eta n))$$
$$B_t = \Theta(\eta)$$

Thus, we have:

$$\delta_t^1 = \Theta(\eta^2 \alpha n r)$$
$$\delta_t^2 = \Theta(\eta \alpha r) \cdot \max(\Theta(1), \Theta(\eta n))$$
$$= \max(\Theta(\eta \alpha r), \Theta(\eta^2 \alpha r n))$$
$$\delta_t^3 = \Theta(\alpha \eta^2 n r)$$

Therefore,

$$\Delta Z_B^t = \max(\Theta(\delta_t^1), \Theta(\delta_t^2), \Theta(\delta_t^3))$$
$$= \max(\Theta(\eta^2 \alpha r n), \Theta(\eta \alpha r), \Theta(\eta^2 \alpha r n))$$
$$= \max(\Theta(\eta \alpha r), \Theta(\eta^2 \alpha r n))$$

$\square$

Imposing $\Delta Z_B^t = \Theta(1)$ and solving for the learning rate scaling in terms of $(n, r)$ and $\alpha = r^{-\gamma}$ with $\gamma \in [0, 1]$ yields the following unified scaling laws for Init[A]:

**Corollary A.18** (Unified scaling rules for Init[A]). *Let $\alpha = r^{-\gamma}$ for $\gamma \in [0, 1]$. Under Init[A], achieving stable feature learning (i.e., $\Delta Z_B^t = \Theta(1)$ for fixed $t \leq T$) requires*

$$\eta = \Theta(n^{-1/2} r^{-(1-\gamma)/2}).$$

*With this choice,*

$$\delta_t^1 = \Theta(1),$$
$$\delta_t^2 = \Theta(1),$$
$$\delta_t^3 = \Theta(1),$$

*and the intermediate feature scales as*

$$\Theta(Z_A^t) = \Theta(n^{1/2} r^{-(1-\gamma)/2}).$$

*In particular:*

| $\gamma$ | $\alpha$ | $\Theta(\eta)$ | $\Theta(Z_A^t)$ |
|---|---|---|---|
| $0$ | $1$ | $n^{-1/2} r^{-1/2}$ | $n^{1/2} r^{-1/2}$ |
| $\frac{1}{2}$ | $r^{-1/2}$ | $n^{-1/2} r^{-1/4}$ | $n^{1/2} r^{-1/4}$ |
| $1$ | $r^{-1}$ | $n^{-1/2}$ | $n^{1/2}$ |

*Proof.* From Equation (22), under Init[A] we have

$$\Delta Z_B^t = \max\left(\Theta(\eta\alpha r), \Theta(\eta^2\alpha rn)\right).$$

Substituting $\alpha = r^{-\gamma}$ yields

$$\Delta Z_B^t = \max\left(\Theta(\eta r^{1-\gamma}), \Theta(\eta^2 n r^{1-\gamma})\right).$$

To achieve $\Delta Z_B^t = \Theta(1)$, we require

$$\eta = \min\left(\Theta(r^{-(1-\gamma)}), \Theta(n^{-1/2}r^{-(1-\gamma)/2})\right).$$

Since $r \le n$, we have $r^{-(1-\gamma)} \ge n^{-1/2}r^{-(1-\gamma)/2}$ for all $\gamma \in [0,1]$. Thus,

$$\eta = \Theta(n^{-1/2}r^{-(1-\gamma)/2}).$$

*Feature update terms.* Substituting into the expressions from Lemmas A.14, A.15, A.16:

- From Lemma A.14 with $B_{t-1} = \Theta(\eta) = \Theta(n^{-1/2}r^{-(1-\gamma)/2})$:

$$\begin{aligned}
\delta_t^1 &= \Theta(\eta\alpha nr) \cdot \Theta(B_{t-1}) \\
&= \Theta(n^{-1/2}r^{-(1-\gamma)/2} \cdot r^{-\gamma} \cdot n \cdot r \cdot n^{-1/2}r^{-(1-\gamma)/2}) \\
&= \Theta(1).
\end{aligned}$$

- From Lemma A.15, using $Z_A^{t-1} = \max(\Theta(1), \Theta(\eta n)) = \Theta(\eta n)$:

$$\begin{aligned}
\delta_t^2 &= \Theta(\eta\alpha r) \cdot \Theta(Z_A^{t-1}) \\
&= \Theta(n^{-1/2}r^{-(1-\gamma)/2} \cdot r^{-\gamma} \cdot r \cdot n^{-1/2}r^{-(1-\gamma)/2} \cdot n) \\
&= \Theta(1).
\end{aligned}$$

- From Lemma A.16, we have:

$$\begin{aligned}
\delta_t^3 &= \Theta(\alpha\eta^2 nr) \\
&= \Theta(r^{-\gamma} \cdot n^{-1}r^{-(1-\gamma)} \cdot n \cdot r) \\
&= \Theta(1).
\end{aligned}$$

Thus, under Init[A], all three terms $\delta_t^1$, $\delta_t^2$, and $\delta_t^3$ remain order-one under the stable learning-rate scaling.

*Intermediate features.* From the recursive formula (Lemma A.11):

$$Z_A^t = \max\left(\Theta(1), \Theta(\eta n)\right) = \Theta(\eta n) = \Theta(n^{1/2}r^{-(1-\gamma)/2}).$$

$\square$

### A.5. Init[B]: Theorems and Corollaries

This part deals with Init[B], in which $B_0$ is randomly initialized and $A_0 = 0$. As in the main text, we first establish the scale of $\Delta Z_B^t$ and then specialize to common multiplier configurations, namely $\alpha = 1$ and $\alpha = r^{-1}$

**Theorem A.19** (Init[B])**.** *Under Assumptions A.9 and A.10, for any fixed $t \le T$ with Init[B],*

$$\Delta Z_B^t = \max\{\Theta(\eta\alpha n), \Theta(\eta^2\alpha nr)\}. \tag{23}$$

*Proof.* With Init[B], we have $B_0 = \Theta(r^{-1/2})$ and $Z_A^0 = 0$. As a result, we have for all $t \leq T$:

$$Z_A^t = \Theta(\eta n)$$
$$B_t = \max\left(\Theta(r^{-1/2}), \Theta(\eta)\right)$$

Thus, we have:

$$\delta_t^1 = \Theta(\eta\alpha n r^{1/2}) \cdot \Theta(B_{t-1})$$
$$= \max\left(\Theta(\eta\alpha n), \Theta(\eta^2\alpha n r^{1/2})\right)$$
$$\delta_t^2 = \Theta(\eta^2\alpha n r)$$
$$\delta_t^3 = \Theta(\alpha\eta^2 n r^{1/2})$$

Therefore,

$$\Delta Z_B^t = \max\left(\Theta(\delta_t^1), \Theta(\delta_t^2), \Theta(\delta_t^3)\right)$$
$$= \max\left(\Theta(\eta\alpha n), \Theta(\eta^2\alpha n r)\right)$$

$\square$

We proceed by considering the LoRA multiplier $\alpha$ case by case. We first consider the choice $\alpha = 1$ (Corollary A.20), and then the rank-dependent choice $\alpha = r^{-1}$ (Corollary A.21).

**Corollary A.20** (Init[B] with $\alpha = 1$)**.** *Let $\alpha = 1$. Under Init[B], achieving stable feature learning (i.e., $\Delta Z_B^t = \Theta(1)$ for fixed $t \leq T$) requires*

$$\eta = \Theta(n^{-1}).$$

*With this choice,*

$$\delta_t^1 = \Theta(1),$$
$$\delta_t^2 = \Theta(r n^{-1}),$$
$$\delta_t^3 = \Theta(r^{1/2} n^{-1}),$$

*and the intermediate feature scales as*

$$\Theta(Z_A^t) = \Theta(1).$$

*Proof.* From Equation (23) with constant $\alpha$:

$$\Delta Z_B^t = \max\left(\Theta(\eta n), \Theta(\eta^2 n r)\right).$$

Obtaining $\Delta Z_B^t = \Theta(1)$ requires:

$$\eta = \min\left(\Theta(n^{-1}), \Theta(n^{-1/2} r^{-1/2})\right).$$

Since $r \leq n$ implies $n^{-1} \leq n^{-1/2} r^{-1/2}$, we have $\eta = \Theta(n^{-1})$.

*Feature update terms.* Under Init[B], we have $B_{t-1} = \max\left(\Theta(r^{-1/2}), \Theta(\eta)\right) = \Theta(r^{-1/2})$ and $Z_A^{t-1} = \Theta(\eta n) = \Theta(1)$.

Substituting into Lemmas A.14, A.15, A.16:

$$\begin{aligned}
\delta_t^1 &= \Theta(\eta \alpha n \sqrt{r}) \cdot \Theta(B_{t-1}) \\
&= \Theta(n^{-1} \cdot 1 \cdot n \cdot r^{1/2} \cdot r^{-1/2}) \\
&= \Theta(1),
\end{aligned}$$

$$\begin{aligned}
\delta_t^2 &= \Theta(\eta \alpha r) \cdot \Theta(Z_A^{t-1}) \\
&= \Theta(n^{-1} \cdot r \cdot 1), \\
&= \Theta(r n^{-1})
\end{aligned}$$

$$\begin{aligned}
\delta_t^3 &= \Theta(\alpha \eta^2 n r^{1/2}) \\
&= \Theta(1 \cdot n^{-2} \cdot n \cdot r^{1/2}) \\
&= \Theta(r^{1/2} n^{-1}).
\end{aligned}$$

*Intermediate features.* From the recursive formula (Lemma A.11):

$$\begin{aligned}
Z_A^t &= \Theta(\eta n) \\
&= \Theta(1)
\end{aligned}$$

$\square$

**Corollary A.21** (Init[B] with $\alpha = r^{-1}$). *Let $\alpha = r^{-1}$. Under Init[B], achieving stable feature learning (i.e., $\Delta Z_B^t = \Theta(1)$ for fixed $t \leq T$) requires*

$$\eta = \min\left(\Theta(r n^{-1}), \Theta(n^{-1/2})\right). \tag{24}$$

*With this choice, the feature update terms satisfy:*

$$\begin{aligned}
\delta_t^1 &= \Theta(\eta n r^{-1}) \\
\delta_t^2 &= \Theta(\eta^2 n) \\
\delta_t^3 &= \Theta(\eta^2 n r^{-1/2})
\end{aligned}$$

*and the intermediate feature scales as*

$$Z_A^t = \Theta(\eta n)$$

*In particular:*

- *If $r \leq \sqrt{n}$, we have $\eta = \Theta(r n^{-1})$ and thus obtain:*

$$\begin{aligned}
\delta_t^1 &= \Theta(1) \\
\delta_t^2 &= \Theta(r^2 n^{-1}) \\
\delta_t^3 &= \Theta(r^{3/2} n^{-1}) \\
Z_A^t &= \Theta(r)
\end{aligned}$$

- *If $r > \sqrt{n}$, we have $\eta = \Theta(n^{-1/2})$ and thus obtain:*

$$\begin{aligned}
\delta_t^1 &= \Theta(n^{1/2} r^{-1}) \\
\delta_t^2 &= \Theta(1) \\
\delta_t^3 &= \Theta(r^{-1/2}) \\
Z_A^t &= \Theta(n^{1/2}).
\end{aligned}$$

*Proof.* Substituting $\alpha = r^{-1}$ into Theorem A.19 gives

$$\Delta Z_B^t = \max\left(\Theta(\eta\alpha n),\ \Theta(\eta^2\alpha nr)\right)$$
$$= \max\left(\Theta(\eta nr^{-1}),\ \Theta(\eta^2 n)\right).$$

Imposing $\Delta Z_B^t = \Theta(1)$ yields

$$\eta = \min\left(\Theta(rn^{-1}),\ \Theta(n^{-1/2})\right).$$

*Feature update terms.* Under Init[B], we have $B_{t-1} = \max(\Theta(r^{-1/2}), \Theta(\eta)) = \Theta(r^{-1/2})$ for either choice of $\eta$ and $Z_A^{t-1} = \Theta(\eta n)$. Substituting into Lemmas A.14, A.15, A.16:

$$\delta_t^1 = \Theta(\eta\alpha n\sqrt{r}) \cdot \Theta(B_{t-1})$$
$$= \Theta(\eta nr^{-1}),$$
$$\delta_t^2 = \Theta(\eta\alpha r) \cdot \Theta(Z_A^{t-1})$$
$$= \Theta(\eta^2 n),$$
$$\delta_t^3 = \Theta(\alpha\eta^2 nr^{1/2})$$
$$= \Theta(\eta^2 nr^{-1/2}).$$

Next, we need to consider the relationship between the network width $n$ and the LoRA rank $r$ to determine scaling rule of $\eta$.

When $r \le \sqrt{n}$, we have $rn^{-1} \le n^{-1/2}$ and thus $\eta = \Theta(rn^{-1})$, which incurs:

$$\delta_t^1 = \Theta(1)$$
$$\delta_t^2 = \Theta(r^2 n^{-1})$$
$$\delta_t^3 = \Theta(r^{3/2} n^{-1})$$

When $r > \sqrt{n}$, we have $rn^{-1} > n^{-1/2}$ and thus $\eta = \Theta(n^{-1/2})$, which incurs:

$$\delta_t^1 = \Theta(n^{1/2} r^{-1})$$
$$\delta_t^2 = \Theta(1)$$
$$\delta_t^3 = \Theta(r^{-1/2})$$

*Intermediate features.* From the recursive formula (Lemma A.11):

$$Z_A^t = \Theta(\eta n)$$

Similarly, when $r \le \sqrt{n}$, we have:

$$Z_A^t = \Theta(r)$$

When $r > \sqrt{n}$, we obtain:

$$Z_A^t = \Theta(n^{1/2})$$

$\square$

## A.6. Learning Rate Scaling for Full Finetuning

In this section, we analyze how the learning rate should scale for full finetuning with regard to the network width $n$. We use a framework similar to our previous derivations.

Consider a single linear layer of width $n$ and pretrained weights $W^\star \in \mathbb{R}^{n \times n}$. Under full finetuning, we update $W$ directly. Here we denote the per-step weight update by

$$\Delta W_t := W_t - W_{t-1}, \qquad W_0 := W^\star.$$

Then, for any fixed $t \leq T$, the layer output can be written as

$$\bar{Z}^t = W_t \underline{Z} = W^\star \underline{Z} + \sum_{s=1}^{t} \Delta W_s \underline{Z}. \tag{25}$$

Following Assumption A.9, we assume that pretraining yields bounded features $\underline{Z} = \mathcal{O}(1)$ and $W^\star \underline{Z} = \mathcal{O}(1)$. By Lemma A.6, it suffices to control each per-step increment $\Delta W_s \underline{Z}$ for feature stability: requiring $\Delta W_s \underline{Z} = \mathcal{O}(1)$ for all $s \leq t$ implies $\sum_{s=1}^{t} \Delta W_s \underline{Z} = \mathcal{O}(1)$. To avoid vanishing updates, we also impose $\Delta W_s \underline{Z} = \Theta(1)$. Assuming that we are using SignSGD (Assumption A.10) and single-sample loss, we can obtain the following theorem:

**Theorem A.22** (Scaling rules for full finetuning). *Under Assumptions A.9 and A.10, for any fixed $t \leq T$, with full finetuning, the learning rate should scale as*

$$\eta = \Theta(n^{-1}). \tag{26}$$

*Proof.* Under SignSGD, the per-step update is $\Delta W_t = -\eta g_t$, where

$$g_t := \mathrm{sign}\Big(\frac{\partial \mathcal{L}_t}{\partial W_{t-1}}\Big).$$

For the linear map $\bar{Z} = W\underline{Z}$, the gradient factorizes as:

$$\frac{\partial \mathcal{L}_t}{\partial W_{t-1}} = d\bar{Z}^{t-1} \otimes \underline{Z}, \tag{27}$$

where $d\bar{Z}^{t-1} := \partial \mathcal{L}_t / \partial \bar{Z}^{t-1}$ and $\otimes$ denotes the outer product. Therefore,

$$\Delta W_t = -\eta \, \mathrm{sign}(d\bar{Z}^{t-1} \otimes \underline{Z})$$
$$= -\eta \big(\mathrm{sign}(d\bar{Z}^{t-1}) \otimes \mathrm{sign}(\underline{Z})\big).$$

Multiplying by $\underline{Z}$ yields

$$\Delta W_t \underline{Z} = \Theta(\eta \cdot (\mathrm{sign}(\underline{Z})^\top \underline{Z}) \cdot \mathrm{sign}(d\bar{Z}^{t-1})) \tag{28}$$
$$= \Theta(\eta n), \tag{29}$$

since $\mathrm{sign}(\underline{Z})^\top \underline{Z} = \|\underline{Z}\|_1 = \Theta(n)$ and $\mathrm{sign}(\cdot) = \Theta(1)$. Imposing $\Delta W_t \underline{Z} = \Theta(1)$ yields:

$$\eta = \Theta(n^{-1}).$$

$\square$

From above, we show that under the same assumptions, full finetuning admits the *same* width-dependent learning rate scaling $\eta = \Theta(n^{-1})$. At the same time, our analysis for LoRA with Init[B] and constant $\alpha = 1$ shows that stable (non-exploding) and non-vanishing feature updates require a learning rate $\eta = \Theta(n^{-1})$ (Corollary A.20). This matching of learning rate scaling might indicate that the hyperparameter landscapes of full finetuning and LoRA with Init[B] and constant $\alpha = 1$ are very close, which opens up an opportunity to do learning rate transfer between these two regimes.

*Table 2.* Shared optimization and training settings.

| Setting | Value |
|---------|-------|
| Optimizer | AdamW (Loshchilov & Hutter, 2019) |
| AdamW momentum | $\beta_1 = 0.9$, $\beta_2 = 0.999$ |
| AdamW epsilon | $\epsilon = 10^{-8}$ |
| Weight decay | 0.01 |
| Gradient clipping | global norm clipped to 1.0 |
| LR schedule | linear warmup for first 5% of steps, then cosine decay to $0.1\times$ peak LR |
| Precision | bfloat16 mixed precision; TF32 `matmul` enabled |
| LoRA dropout | 0.0 |
| LoRA Bias | `None` |

## B. Implementation and Experimental Details

In this section, we provide more details on our experimental settings.

### B.1. Model Configuration

All pretrained models are obtained from public model hubs and used without architectural modifications beyond (i) adding LoRA adapters and (ii) adding a linear classification head for discriminative tasks. Concretely, we use the following models:

- **Llama-3.2-1B** (Grattafiori et al., 2024) (`meta-llama/Llama-3.2-1B`)[3].

- **Qwen2.5-3B-Instruct** (Qwen et al., 2025) (`Qwen/Qwen2.5-3B-Instruct`)[4].

- **RoBERTa-large** (Liu et al., 2019) (`FacebookAI/roberta-large`)[5].

- **ViT-Huge/14** (Dosovitskiy et al., 2021) (`google/vit-huge-patch14-224-in21k`)[6].

- **Qwen3-VL-2B-Instruct** (Bai et al., 2025) (`Qwen/Qwen3-VL-2B-Instruct`)[7].

- **Stable Diffusion v1.5** (Rombach et al., 2022). We use the standard *v1.5* weights in diffusers format; a widely used public mirror is `stable-diffusion-v1-5/stable-diffusion-v1-5`[8].

- **Llama-3.1-8B** (Grattafiori et al., 2024) (`meta-llama/Llama-3.1-8B`)[9].

For discriminative tasks, we attach a task-specific classification head to the final pooled representation: the pooled text representation for RoBERTa on ANLI, and the `[CLS]` token embedding for ViT on ImageNet-1K. During training, this classification head is trained jointly with either the full backbone (full finetuning) or the adapters (LoRA). Across all hyperparameter sweeps, we fix the classifier learning rate at $10^{-3}$ and only vary the backbone (or adapter) learning rate.

Unless otherwise mentioned, we use the same optimizer and learning rate schedule across all architectures. The random seed is set to 42 across all experiments. Table 2 summarizes these settings.

### B.2. LoRA Configurations and PEFT Implementation

Throughout the paper, we analyze the LoRA-augmented weight matrix formulated as follows:

$$\mathbf{W} \leftarrow \mathbf{W} + \alpha \mathbf{BA},$$

---

[3] https://huggingface.co/meta-llama/Llama-3.2-1B
[4] https://huggingface.co/Qwen/Qwen2.5-3B-Instruct
[5] https://huggingface.co/FacebookAI/roberta-large
[6] https://huggingface.co/google/vit-huge-patch14-224-in21k
[7] https://huggingface.co/Qwen/Qwen3-VL-2B-Instruct
[8] https://huggingface.co/stable-diffusion-v1-5/stable-diffusion-v1-5
[9] https://huggingface.co/meta-llama/Llama-3.1-8B

*Table 3.* Implementation details of LoRA placement across different models, based on the PEFT package. For vision-language models, we use regular expressions to identify target modules. For the diffusion model, we apply LoRA only to the U-Net denoiser.

| Models | Target Modules |
|---|---|
| Decoder-only LMs (Llama, Qwen2.5) | `gate_proj`, `up_proj`, `down_proj` |
| Encoder-only models (RoBERTa, ViT) | `query`, `key`, `value` 
 `attention.output.dense`, 
 `intermediate.dense`, `output.dense` |
| Vision–language model (Qwen3-VL) | `model.visual.blocks.\d+.attn.qkv` 
 `model.visual.blocks.\d+.attn.proj` 
 `model.visual.blocks.\d+.mlp.linear_fc[12]` 
 `model.visual.merger.linear_fc[12]` 
 `model.visual.deepstack_merger_list.\d+.linear_fc[12]` 
 `model.language_model.layers.\d+.mlp.gate_proj` 
 `model.language_model.layers.\d+.mlp.up_proj` 
 `model.language_model.layers.\d+.mlp.down_proj` |
| Diffusion (Stable Diffusion v1.5) | `to_k`, `to_q`, `to_v`, `to_out.0` |

where $\mathbf{A} \in \mathbb{R}^{r \times n}$ and $\mathbf{B} \in \mathbb{R}^{n \times r}$ are rank-$r$ factors, and $\alpha$ is the LoRA multiplier.

Our implementation uses the PEFT library (Mangrulkar et al., 2022), which internally parameterizes the update as

$$\mathbf{W} \leftarrow \mathbf{W} + \frac{s}{r} \mathbf{BA},$$

where $s$ corresponds to `lora_alpha` in `LoraConfig`. To achieve an effective multiplier $\alpha$ in our notation, we set $s = \alpha r$. For example, constant multiplier $\alpha = 1$ requires `lora_alpha = r`, whereas rank-dependent multiplier $\alpha = r^{-1}$ requires `lora_alpha = 1`. In both cases, we set `use_rslora=False`.

We study the three LoRA configurations described in Section 5.1:

- **Init[A] with $\alpha = 1$:** initialize $\mathbf{A}$ randomly and set $\mathbf{B} = \mathbf{0}$. Set `lora_alpha` to the rank used.

- **Init[A] with $\alpha = r^{-1}$:** same initialization as above, but use a multiplier $\alpha = r^{-1}$. Set `lora_alpha` to 1.

- **Init[B] with $\alpha = 1$:** initialize $\mathbf{B}$ randomly and set $\mathbf{A} = \mathbf{0}$. Set `lora_alpha` to the rank used.

In all cases, the *effective* update at initialization is exactly zero (because one factor is initialized to zero), so the LoRA-augmented model initially matches the pretrained backbone. For the nonzero factor, we use Kaiming normal initialization (He et al., 2015). In terms of the per-entry variance, this corresponds to $\mathrm{Var}(A_{ij}) = 1/n$ for Init[A] and $\mathrm{Var}(B_{ij}) = 1/r$ for Init[B], matching the theoretical definition in Definition 3.2.

**Where LoRA is applied.** As listed in Table 3, our adapter placement strategy is:

- **Decoder-only LMs (Llama, Qwen2.5):** LoRA is inserted *only* into the feed-forward (MLP) block of each transformer layer (i.e., the expansion, gating, and contraction projections of the MLP).

- **Encoder-only models (RoBERTa, ViT):** LoRA is inserted into all linear projections within each attention block (query/key/value/output) and within the feed-forward block (the two MLP linear layers).

- **Vision–language model (Qwen3-VL):** LoRA is inserted into (i) the vision encoder linear projections, (ii) the vision–language merger module (the "merger" MLP), and (iii) the text decoder *MLP blocks*.

- **Diffusion (Stable Diffusion v1.5):** LoRA is inserted into all self-attention and cross-attention linear projections (query/key/value/output) in the denoising U-Net. The variational autoencoder (VAE) (Kingma & Welling, 2014) and the CLIP text encoder (Radford et al., 2021) remain frozen.

*Table 4.* Task-specific settings used in our experiments. Effective batch size refers to the number of examples (or fixed-length packed blocks) processed per optimizer step, accounting for gradient accumulation. Input size refers to maximum sequence length for language tasks, image resolution for vision tasks, and maximum generation length for RLVR task; this field is omitted for vision–language tasks. NLL stands for negative log-likelihood, while FID stands for Fréchet Inception Distance.

| Model | Dataset | Train / Val size | Input size | Effective batch size | Validation metric |
|---|---|---|---|---|---|
| Llama-3.2-1B | Tulu-3 SFT mixture | 320k / 32k | 1024 | 32 | Token-level NLL |
| Qwen2.5-3B-Instruct | OpenThoughts-114k | 71k / 6k | 8192 | 8 | Token-level NLL |
| RoBERTa-large | ANLI | 163k / 3k | 128 | 32 | Accuracy |
| ViT-Huge/14 | ImageNet-1K | 1.28M / 50k | $224 \times 224$ | 512 | Top-1 accuracy |
| Qwen3-VL-2B-Instruct | LLaVA-Instruct-Mix | 198k / 17k | - | 32 | Token-level NLL |
| Stable Diffusion v1.5 | Naruto-BLIP-Captions | 1.1k / 0.1k | $512 \times 512$ | 8 | FID scores |
| Llama-3.1-8B (RLVR) | GSM8k | 500 / 50 | 1024 | 8 | Test Reward |

## B.3. Supervised Finetuning Setting

**Summary of task-specific settings.** Table 4 summarizes the task-level choices that are not fully shared (data sources, input sizing, and the primary validation metric). For datasets without an official validation split, we create a held-out split by reserving 8% of the samples for validation and using the remaining 92% for training.

### B.3.1. LANGUAGE MODELING TASKS

**Tulu-3 SFT mixture (Llama-3.2-1B).** For instruction-following task, we use Llama-3.2-1B (Grattafiori et al., 2024) trained on the Tulu-3 supervised finetuning mixture (Lambert et al., 2024), available as `allenai/tulu-3-sft-mixture`[10]. The dataset comprises instruction-following conversations formatted as (user prompt, assistant response) pairs. To keep training time per round manageable, we subsample 320k and 32k examples from the original dataset for training and validation, respectively. We truncate sequences to a maximum length of 1024 tokens, and use dynamic padding within each mini-batch. The effective batch size is 32 samples per optimizer step.

**OpenThoughts-114k (Qwen2.5-3B-Instruct).** To evaluate long-context reasoning, we train Qwen2.5-3B-Instruct (Qwen et al., 2025) on OpenThoughts-114k (Guha et al., 2025), available as `open-thoughts/OpenThoughts-114k`[11]. Each example contains a system instruction and a multi-turn conversation in which assistant messages include both reasoning traces and final answers. For computational tractability, we draw 71k and 6k examples for training and validation, respectively. Since sequence lengths vary widely, we employ fixed-length packing (Raffel et al., 2020): tokenized conversations are concatenated with end-of-sequence delimiters into blocks of exactly 8192 tokens. To minimize padding overhead, we adopt a padding-free batching strategy that concatenates packed blocks into a single sequence per optimizer step, using attention masking and position indexing to prevent cross-block attention. The effective batch size is 8 packed blocks per step.

### B.3.2. ENCODER-BASED CLASSIFICATION TASKS

**ANLI (RoBERTa-large).** Natural language inference experiments use RoBERTa-large (Liu et al., 2019) on the Adversarial NLI benchmark (Nie et al., 2020), available as `facebook/anli`[12]. We merge all three rounds by concatenating `train_r1`, `train_r2`, and `train_r3` into a single training set, and similarly concatenating `dev_r1`, `dev_r2`, and `dev_r3` into a single validation set. Each example consists of a premise–hypothesis pair, which we encode using the standard RoBERTa sentence-pair format with separator tokens and truncate to a maximum of 128 tokens. We apply dynamic padding within each mini-batch. The model predicts a 3-way classification label (entailment, neutral, or contradiction).

**ImageNet-1K (ViT-Huge/14).** For large-scale image classification, we adapt ViT-Huge/14 (Dosovitskiy et al., 2021) to ImageNet-1K (Deng et al., 2009), available as `ILSVRC/imagenet-1k`[13], using the standard ILSVRC-2012 training and

---

[10] https://huggingface.co/datasets/allenai/tulu-3-sft-mixture
[11] https://huggingface.co/datasets/open-thoughts/OpenThoughts-114k
[12] https://huggingface.co/datasets/facebook/anli
[13] https://huggingface.co/datasets/ILSVRC/imagenet-1k

*Table 5.* Prompt template used in the RLVR experiments, where {QUESTION} is replaced by the problem statement from each example.

| **Prompt template** |
| --- |
| *Solve the following grade-school math problem. Show reasoning in* `<think>...</think>`, *then give the final numerical answer in* `<answer>...</answer>`. 
 *Question:* {QUESTION} |
| *Solution:* |

validation splits (1.28M and 50k images, respectively). Images are converted to RGB and resized to $224 \times 224$ pixels. During training, we apply standard ImageNet augmentation (random resized crop and horizontal flip with a flipping probability of $0.5$); for evaluation, we use deterministic resizing with center cropping. We normalize inputs per channel with mean and standard deviation of $0.5$, as specified by the pretrained checkpoint.

### B.3.3. VISION–LANGUAGE TASK

**LLaVA-Instruct-Mix (Qwen3-VL-2B-Instruct).** Vision–language experiments employ Qwen3-VL-2B-Instruct (Bai et al., 2025) trained on LLaVA-Instruct-Mix, available as `trl-lib/llava-instruct-mix`[14]. Each example pairs a single image with a user prompt and assistant response. To limit compute, we subsample 198k training and 17k validation samples in our experiment. We use the official Qwen3-VL multimodal processor, which encodes images at native resolution with a SigLIP-2 vision encoder (Tschannen et al., 2025), compresses $2 \times 2$ visual features into a single visual token via an MLP merger, and applies the chat template to interleave visual and text tokens into the final input sequence.

### B.4. Reinforcement Learning with Verifiable Rewards (RLVR) Setting

**GSM8k (Llama-3.1-8B).** We finetune Llama-3.1-8B (Grattafiori et al., 2024) with reinforcement learning using verifiable, rule-based rewards via Group Relative Policy Optimization (GRPO) (Shao et al., 2024). Our experiments use the GSM8k (Cobbe et al., 2021) dataset, available as `openai/gsm8k`[15]. This dataset contains 8.5k grade-school math problems paired with their solutions. We randomly sample the 500 training problems and 50 test problems for our training and validation splits, respectively. Each problem is converted into a prompt using a template we design to elicit the model's reasoning; the prompt template is summarized in Table 5. We compute a scalar reward for each sampled completion and sum over two rule-based components:

- *Accuracy reward*: assigns $+1$ when the extracted final answer matches the reference, and $0$ otherwise.

- *Format reward*: assigns $+0.1$ when the completion places reasoning within `<think>...</think>` tags and the final answer within `<answer>...</answer>` tags, and $0$ otherwise.

For each prompt, we sample $8$ completions from the current policy using temperature $1.0$, with no nucleus truncation (top-$p = 1.0$), no top-$k$ truncation (top-$k = 0$), and repetition penalty $1.0$. We set the maximum completion length to $1024$ tokens and standardize rewards within each prompt's sample group. We use the DAPO variant (Yu et al., 2025) of the GRPO loss, which uses a token-level loss to accommodate longer sequences. We set weight decay to $0$, as it strongly degraded performance in our experiments. We omit the KL penalty term since it provides minimal benefit while increasing memory overhead by requiring a reference model to be loaded.

### B.5. Text-to-image Diffusion Model Finetuning Setting

**Naruto-BLIP-Captions (Stable Diffusion v1.5).** For text-to-image generation, we adapt Stable Diffusion v1.5 (Rombach et al., 2022) using Naruto-BLIP-Captions (Cervenka, 2022), available as `lambdalabs/naruto-blip-captions`[16]. The dataset contains 1.2k image–caption pairs; we reserve $8\%$ for validation. Images are resized and cropped to $512 \times 512$, then encoded into the latent space of the pretrained VAE (Kingma & Welling, 2014). Captions are tokenized using the

---

[14]`https://huggingface.co/datasets/trl-lib/llava-instruct-mix`
[15]`https://huggingface.co/datasets/openai/gsm8k`
[16]`https://huggingface.co/datasets/lambdalabs/naruto-blip-captions`

pretrained CLIP text encoder (Radford et al., 2021). During finetuning, we freeze both the VAE and text encoder, tuning only the denoising U-Net. We use the DDPM noise scheduler (Ho et al., 2020) with an epsilon (noise) prediction objective. We train for $10,000$ optimizer steps with a batch size of $8$. Unlike the supervised finetuning experiments, the denoising loss is not reliably monotonic during LoRA finetuning, so we select the best learning rate based on validation FID.

### B.6. Evaluation Protocol and Hyperparameter Sweeps

**Metrics.**  In addition to the training loss, we also report the evaluation metric for each task, summarized in Table 4. These metrics are defined as follows:

- **Token-level negative log-likelihood (NLL)** is computed as the average negative log-probability assigned by the model to each ground-truth token in the sequence. Lower NLL indicates better predictive performance.

- **Accuracy** for ANLI and ImageNet-1K: top-1 classification accuracy on the validation split. Higher is better.

- **Fréchet Inception Distance (FID)** (Heusel et al., 2017) measures the distributional similarity between generated and real images. We extract features from the 2048-dimensional final pooling layer of Inception v3 (Szegedy et al., 2016), with images resized to $299 \times 299$. Lower FID indicates higher fidelity to the training data.

- **Reward** for RLVR task: Mean per-completion reward on the held-out GSM8k test subset, averaged over $8$ sampled completions per prompt. Each completion receives $+1.0$ if the extracted final answer matches the reference, and an additional $+0.1$ if it follows the expected output format.

*Table 6.* Learning rate sweep ranges ($\log_2 \eta$) for each model and configuration.

| Model | Init[A], $\alpha{=}1$ | Init[A], $\alpha{=}r^{-1}$ | Init[B], $\alpha{=}1$ |
|---|---|---|---|
| Llama-3.2-1B | $[-17, -9]$ | $[-13, -7]$ | $[-20, -11]$ |
| Qwen2.5-3B-Instruct | $[-15, -8]$ | $[-11, -7]$ | $[-16, -11]$ |
| Qwen3-VL-2B-Instruct | $[-17, -9]$ | $[-13, -7]$ | $[-18, -11]$ |
| RoBERTa-large | $[-18, -8]$ | $[-13, -7]$ | $[-19, -12]$ |
| ViT-Huge/14 | $[-15, -7]$ | $[-11, -5]$ | $[-15, -9]$ |
| Stable Diffusion v1.5 | $[-14, -7]$ | $[-12, -5]$ | $[-14, -7]$ |
| Llama-3.1-8B | $[-18, -11]$ | $[-16, -11]$ | $[-21, -15]$ |

**Learning rate sweep design.**  We sweep the learning rate on a $log_2$ grid (i.e., adjacent points differ by a factor of two). The grid range varies across LoRA configurations, models, and tasks, chosen to span from peak training performance to the onset of divergence. For full finetuning, we choose the same learning rate grid as Init[B] with $\alpha = 1$ for comparison. For language modeling tasks, we report final training loss (EMA-smoothed with a smoothing factor of $0.1$) and validation NLL. For classification, we report final training loss (EMA-smoothed) and validation accuracy. For diffusion, we report validation FID. For RLVR tasks, we report final training reward (EMA-smoothed) and test reward. We select the optimal learning rate separately for each metric.

**Rank grids.**  To manage computational cost, we evaluate each configuration over a small grid of LoRA ranks, using 5 ranks in total. For Init[A], we examine how the optimal learning rate scales with rank under different multipliers $\alpha$ and compare the observed trends against our theoretical predictions. To capture behavior across a broad range, we use ranks $r \in \{4, 16, 64, 256, 1024\}$ (increasing by a factor of $4\times$). For Init[B] with $\alpha = 1$, we additionally test our hyperparameter transfer conjecture by comparing its optimal learning rate with that of full finetuning. To verify whether the transfer holds across most ranks within a range, we use a denser grid with $2\times$ steps: $r \in \{16, 32, 64, 128, 256\}$. For RLVR tasks, practitioners typically favor small ranks, with $r = 1$ being commonly used in practice. Therefore, we use $r \in \{1, 4, 16, 64, 256\}$ for both Init[A] and Init[B]. For diffusion model finetuning task, we use $r \in \{1, 4, 16, 64, 256\}$ for Init[A] and keep $r \in \{16, 32, 64, 128, 256\}$ for Init[B].

**Learning rate grids.**  Table 6 lists the learning rate exponent ranges used for each setting. We evaluate every integer exponent in the stated range (i.e., step size of $1$). Full finetuning uses the same range as Init[B], $\alpha = 1$.

*Table 7.* Software packages and versions.

| Component | Version |
| --- | --- |
| torch (Paszke et al., 2019) | 2.7.0+cu128 |
| CUDA toolkit | 12.8 |
| transformers (Wolf et al., 2020) | 4.57.1 |
| peft (Mangrulkar et al., 2022) | 0.17.0 |
| trl (von Werra et al., 2020) | 0.26.2 |
| datasets (Lhoest et al., 2021) | 4.3.0 |
| diffusers (von Platen et al., 2022) | 0.36.0 |
| evaluate | 0.4.6 |
| torchvision (TorchVision, 2016) | 0.22.0+cu128 |
| torchmetrics (Detlefsen et al., 2022) | 1.8.2 |
| kernels | 0.11.7 |
| flash_attn (Dao, 2024) | 2.8.3 |

### B.7. Hardware Configuration

All experiments were conducted on a single compute node running Ubuntu 24.04.3 LTS, equipped with two Intel Xeon Gold 6448Y CPUs (64 physical cores total), 1.0 TiB of system memory, and four NVIDIA H200 GPUs (141 GiB HBM each). We used NVIDIA driver version 590.44.01 with CUDA 13.1.

### B.8. Software Configuration

Our implementation is based on PyTorch and the Hugging Face ecosystem: transformers for model APIs, peft for LoRA injection, trl for supervised finetuning and reinforcement learning, diffusers for Stable Diffusion, datasets for data loading, and evaluate/torchmetrics for evaluation. We enable mixed-precision training (bf16/TF32 matmul), fused AdamW kernels, and FlashAttention-2 (Dao, 2024) implementation. Version details are listed in Table 7.

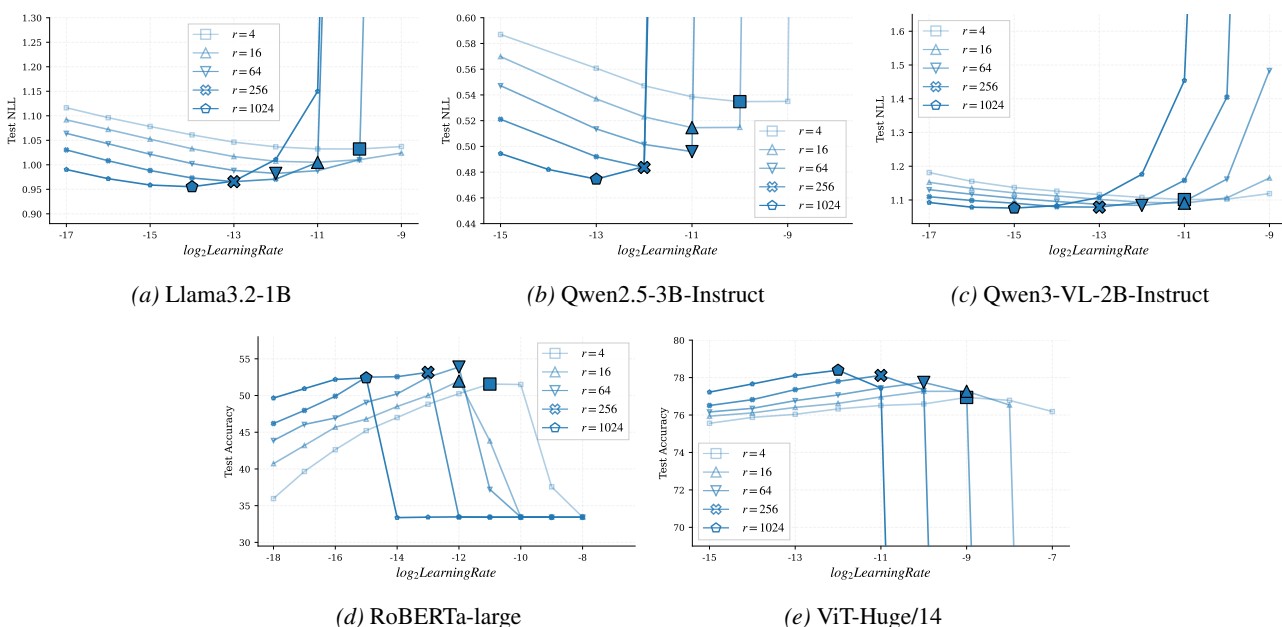

*Figure 10.* Learning rate sweeps for Init[A] with $\alpha = 1$ across five models. Curves show evaluation metrics versus log-scale learning rate $\log_2(\eta)$. Large markers denote per-rank optima. Y-axes are clipped for readability. Top: models finetuned on language modeling tasks, evaluated by test NLL (lower is better). Bottom: models finetuned on classification tasks, evaluated by top-1 accuracy (higher is better).

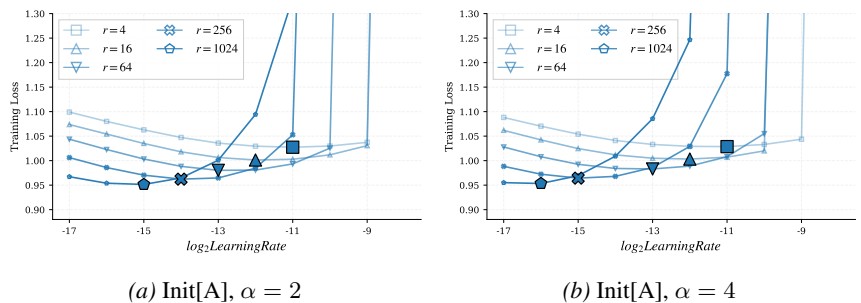

*Figure 11.* Learning rate sweeps on Llama-3.2-1B (tulu3) for Init[A] with (a) $\alpha = 2$ and (b) $\alpha = 4$. Curves show training loss versus log-scale learning rate $\log_2(\eta)$. Large markers indicate the optimal learning rate for each rank.

## C. Additional Results for Supervised Finetuning (SFT)

### C.1. Init[A] with Constant $\alpha = 1$

Figure 10 shows learning rate sweeps for Init[A] with $\alpha = 1$, where the optimal learning rate is selected using task-specific validation metrics. We observe a consistent pattern with Figure 3.

Our theory applies when the LoRA multiplier $\alpha$ remains constant as rank $r$ increases. Figure 11 and Figure 12 show that a consistent pattern holds for other constant values such as $\alpha = 2$ and $\alpha = 4$: as rank $r$ increases, the optimal learning rate decreases accordingly.

### C.2. Init[A] with Rank-Dependent $\alpha = r^{-1}$

Figure 13 shows learning rate sweeps for Init[A] with $\alpha = r^{-1}$, where the optimal learning rate is selected using task-specific validation metrics. We observe a consistent pattern with Figure 4.

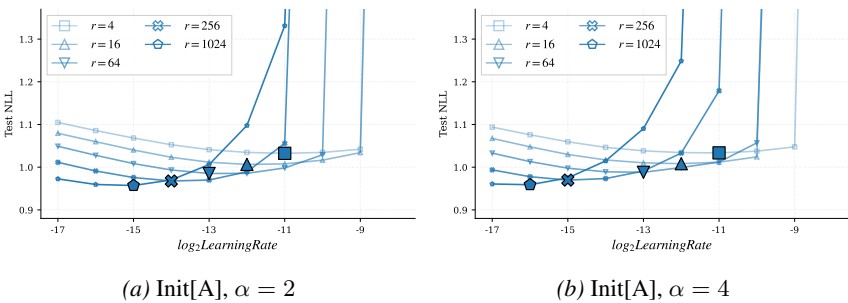

*(a)* Init[A], $\alpha = 2$        *(b)* Init[A], $\alpha = 4$

*Figure 12.* Learning rate sweeps on Llama-3.2-1B (tulu3) for Init[A] with (a) $\alpha = 2$ and (b) $\alpha = 4$. Curves show test loss versus log-scale learning rate $\log_2(\eta)$. Large markers indicate the optimal learning rate for each rank.

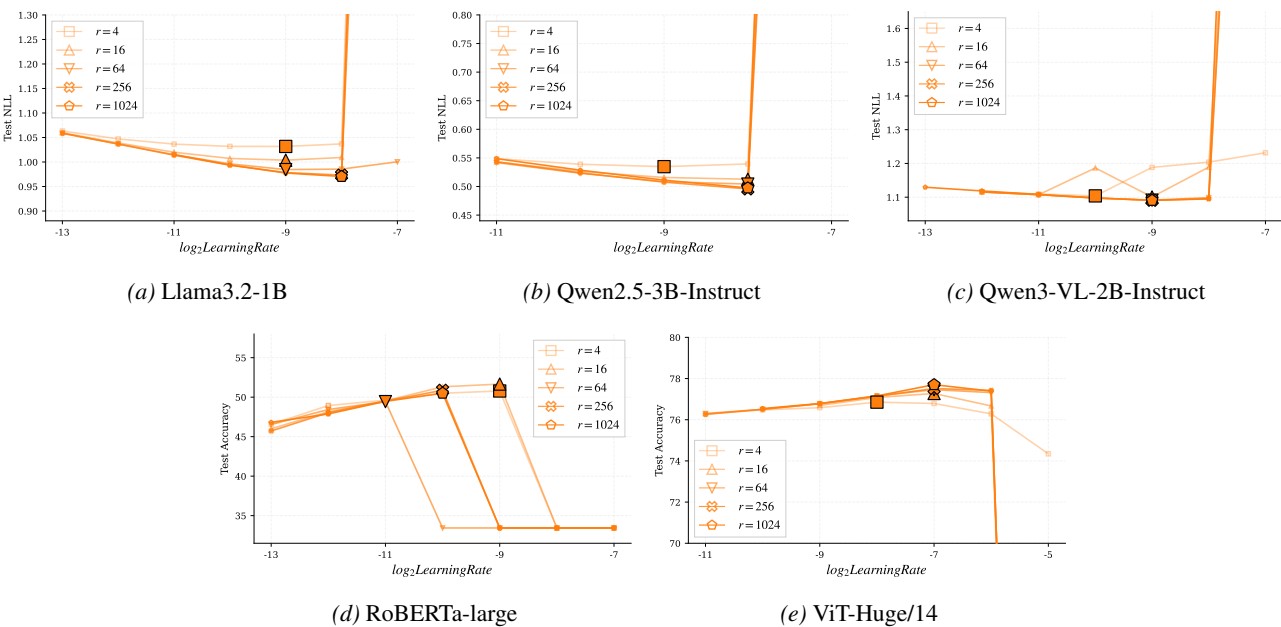

*(a)* Llama3.2-1B      *(b)* Qwen2.5-3B-Instruct      *(c)* Qwen3-VL-2B-Instruct

*(d)* RoBERTa-large      *(e)* ViT-Huge/14

*Figure 13.* Learning rate sweeps for Init[A] with $\alpha = r^{-1}$ across five models. Curves show evaluation metrics versus log-scale learning rate $\log_2(\eta)$. Large markers denote per-rank optima. Y-axes are clipped for readability. Top: models finetuned on language modeling tasks, evaluated by test NLL (lower is better). Bottom: models finetuned on classification tasks, evaluated by accuracy (higher is better).

### C.3. Init[B] with Constant $\alpha = 1$

Figure 14 shows learning rate sweeps for Init[B] with $\alpha = 1$, where the optimal learning rate is selected using task-specific validation metrics. We observe a consistent pattern with Figure 5.

To verify that the learning rate transfer extends beyond training loss to downstream-task performance, we evaluate Llama3.2-1B finetuned on Tulu3 with FFT at varying learning rates on the MMLU benchmark (Hendrycks et al., 2020). Table 8 reports both the final training loss (NLL) and the MMLU accuracy across learning rates. The learning rate that minimizes training loss ($\log_2 \eta = -14$) achieves near-optimal MMLU accuracy (30.21% versus the best 32.54%), confirming that training loss is a reliable proxy for downstream performance. Moreover, the best MMLU accuracy is attained at $\log_2 \eta = -15$, which coincides with the LoRA-to-FFT transferred rate obtained under Init[B] with $\alpha = 1$ at rank $r = 256$ (marked with ‡ in Table 8). This provides preliminary evidence that our learning rate transfer holds at the level of downstream tasks as well.

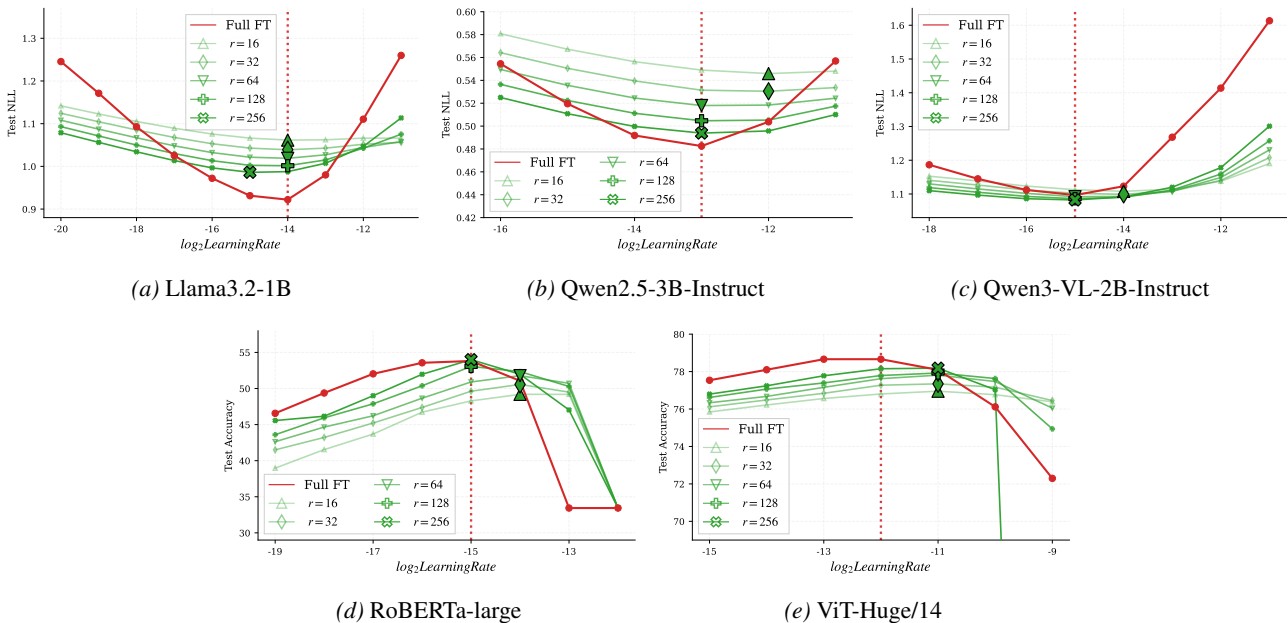

*Figure 14.* Learning rate sweeps for Init[B] with $\alpha = 1$ across five models. Curves show evaluation metrics versus log-scale learning rate $\log_2(\eta)$ for multiple ranks (green) and FFT (red). Large markers denote per-rank optima. Y-axes are clipped for readability. The red vertical dashed line marks the optimal FFT learning rate. Top: models finetuned on language modeling tasks, evaluated by test NLL (lower is better). Bottom: models finetuned on classification tasks, evaluated by top-1 accuracy (higher is better).

| $\log_2 \eta$ | $-18$ | $-17$ | $-16$ | $-15^{\ddagger}$ | $-14^{\ddagger}$ | $-13$ | $-12$ | $-11$ |
|---|---|---|---|---|---|---|---|---|
| Training Loss ($\downarrow$) | 1.086 | 1.019 | 0.966 | 0.925 | **0.915** | 0.972 | 1.103 | 1.253 |
| MMLU Acc. ($\uparrow$) | 24.42% | 27.52% | 29.70% | **32.54%** | 30.21% | 26.27% | 23.84% | 24.38% |

*Table 8.* Training loss (NLL) and MMLU accuracy for Llama3.2-1B finetuned on Tulu3 with FFT across different learning rates. Lower training loss and higher MMLU accuracy indicate better performance; bold denotes the best value for each metric. The ‡ symbol indicates learning rates where the optimal learning rates of Init[B] ($\alpha = 1$) cluster across all LoRA ranks in Figure 5.

## D. Additional Results for Reinforcement Learning with Verifiable Rewards (RLVR)

### D.1. Learning Rate Sweep Results

Figure 15 shows learning rate sweeps for Init[A] with $\alpha = r^{-1}$ and Init[B] with $\alpha = 1$, where the optimal learning rate is selected using the test reward. We observe a consistent pattern with Figure 6.

Figure 16 shows the learning rate sweeps for Init[A] with constant $\alpha = 1$ across multiple ranks. The learning rate sweeps reveal patterns consistent with our SFT findings. Under Init[A] with $\alpha = 1$, the optimal learning rate should scale inversely with the square root of the rank: $\eta \propto r^{-1/2}$. In the figure, we observe that as the rank increases, the optimal learning rate decreases accordingly, and this pattern is consistent across both training reward and test reward. Moreover, we observe that the peak performance is comparable across different ranks, consistent with findings in (Schulman, 2025).

### D.2. Case Study on the Relationship between Learning Rate and Training Time

In our experiment, we found that a suboptimal learning rate usually causes a longer training time under RLVR. We attribute this to a two-phase training pattern we find: in the first phase, the model explores the solution space by generating long sequences (often hitting the maximum generation limit); in the second phase, once the model learns to solve the task, it produces more concise solutions that are 4–5× shorter. At near-optimal learning rates, the model quickly exits the exploratory phase, whereas at suboptimal rates it remains stuck and continues generating long sequences, prolonging each training step. Here, we conduct a case study examining this phenomenon. We compare FFT and Init[B] (with constant $\alpha = 1$ and rank $r = 256$) under two learning rate settings: the optimal rate ($\eta = 2^{-18}$) and a suboptimal rate ($\eta = 2^{-16}$).

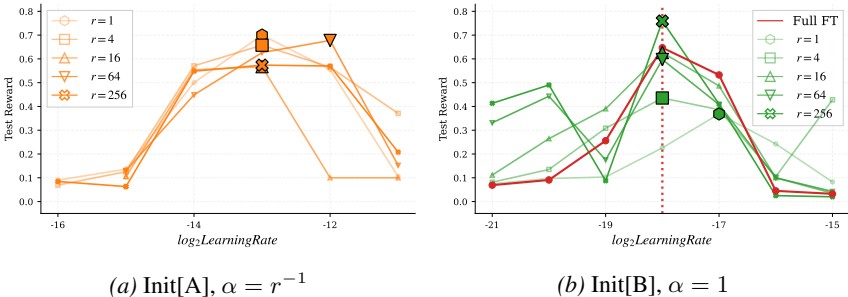

*(a)* Init[A], $\alpha = r^{-1}$           *(b)* Init[B], $\alpha = 1$

*Figure 15.* RLVR results on Llama-3.1-8B (GSM8k) for (a) Init[A] with $\alpha = r^{-1}$ and (b) Init[B] with $\alpha = 1$. Curves show test reward versus log-scale learning rate $\log_2(\eta)$. Large markers indicate the optimal learning rate for each rank.

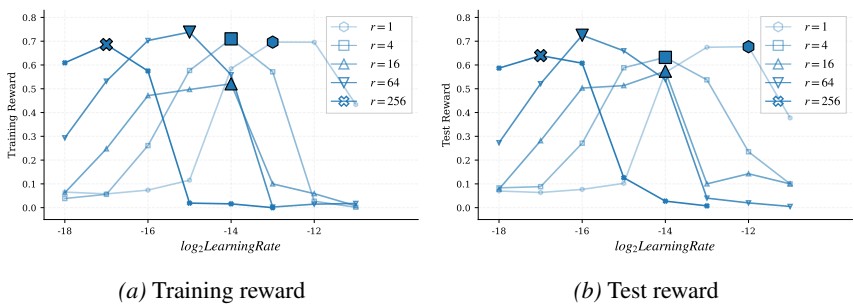

*(a)* Training reward           *(b)* Test reward

*Figure 16.* RLVR results on Llama-3.1-8B (GSM8k) for Init[A] with $\alpha = 1$. (a) Training reward and (b) test reward versus log-scale learning rate $\log_2(\eta)$. Large markers indicate the optimal learning rate for each rank.

As shown in Figures 17a and 17b, training and test rewards collapse to near zero under the suboptimal learning rate, whereas they steadily increase under the optimal rate. More importantly, with the optimal learning rate, the training exhibits the two-phase pattern discussed above: as shown in Figure 17c, completion lengths initially approach the maximum generation limit (1024 tokens) during early exploration (steps $< 50$), then sharply decrease to approximately 200 tokens (a 4–5× reduction) around step 100 as the model learns to produce concise solutions. In contrast, models trained with the suboptimal learning rate remain stuck in the exploratory phase, consistently generating maximum-length completions throughout training. This directly translates to longer per-step training times, as shown in Figure 17d: models with optimal learning rates require only 4–5 seconds per step, while those with suboptimal rates require 15–22 seconds—a 3–5× increase.

In Table 9, we sample responses from the model to a fixed question during full finetuning with optimal and suboptimal learning rates. With the optimal learning rate, the model first learns to follow the specified format, then attempts to solve the problem, and finally learns to simplify its solution. In contrast, with the suboptimal learning rate, the model fails to follow the specified response format at the beginning and eventually collapses into repetitive token generation.

We note one exception: Init[B] with $\alpha = 1$ and a higher learning rate ($\eta = 2^{-15}$) also exhibits shorter per-step training times. However, this represents the opposite failure mode—rather than remaining stuck at maximum-length generations, the model collapses and stops generation almost immediately, producing completions of $\sim 2$ tokens, as shown in Figure 18.

## E. Additional Results for Text-to-Image Diffusion Model Finetuning

### E.1. Learning Rate Sweep Results

Figure 19 shows learning rate sweeps for Init[A] with $\alpha = 1$. Consistent with our theoretical prediction that $\eta \propto r^{-1/2}$, the optimal learning rate $\eta$ shifts left by approximately 1 on the $\log_2$ scale for every 4× increase in rank $r$.

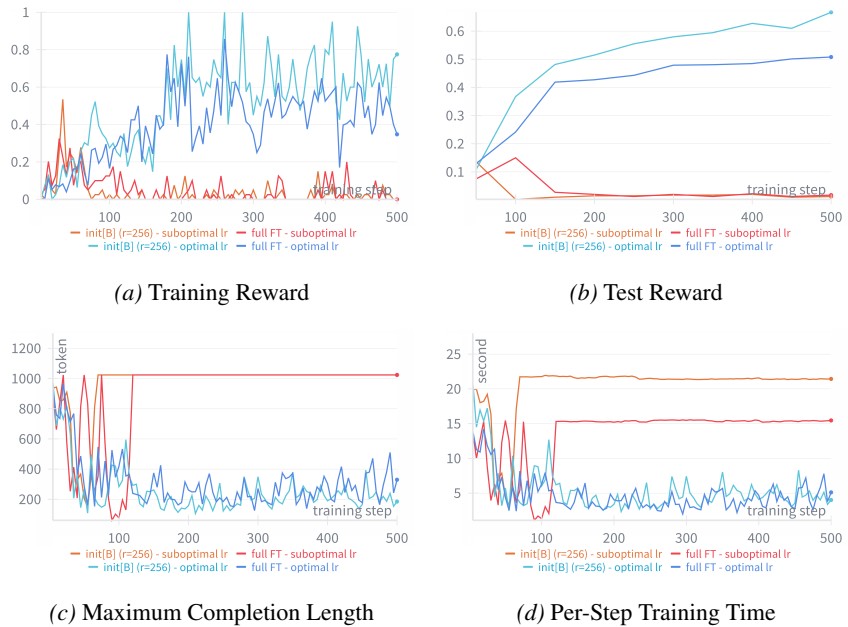

*(a)* Training Reward          *(b)* Test Reward

*(c)* Maximum Completion Length      *(d)* Per-Step Training Time

*Figure 17.* Case study comparing optimal ($\eta = 2^{-18}$) and suboptimal ($\eta = 2^{-16}$) learning rates for FFT and Init[B] ($\alpha = 1$ and $r = 256$). (a) Training reward and (b) test reward show that suboptimal learning rates cause reward collapse, while optimal rates yield steady improvement. (c) Maximum completion length reveals the two-phase training pattern: under optimal learning rates, models transition from generating near-maximum-length sequences ($\sim$1024 tokens) to concise solutions ($\sim$200 tokens), whereas suboptimal rates cause models stuck in the exploratory phase with persistently long outputs. (d) Per-step training time directly reflects the completion length, with optimal learning rates achieving 3–5$\times$ faster for each training step.

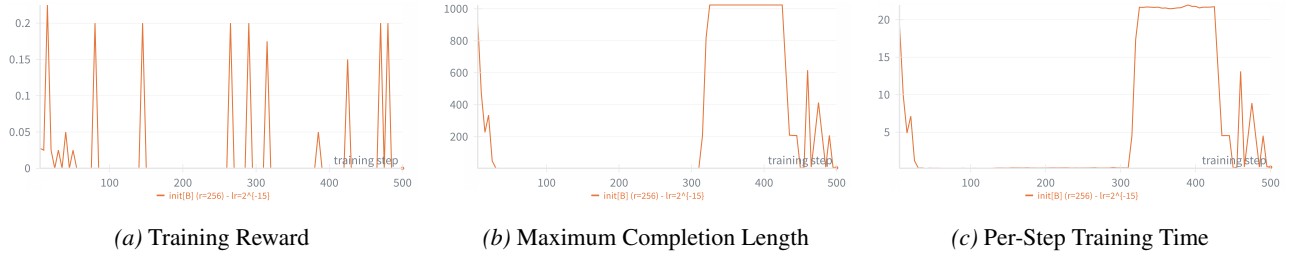

*(a)* Training Reward      *(b)* Maximum Completion Length      *(c)* Per-Step Training Time

*Figure 18.* RLVR finetuning dynamics for Init[B] with rank $r = 256$ and learning rate $\eta = 2^{-15}$.

# F. Additional Results for Ablation Study

## F.1. SignSGD versus AdamW

Figure 20 shows the learning rate sweeps under SignSGD for Init[A] with $\alpha = r^{-1}$ on Llama-3.2-1B (Tulu3). We observe a consistent pattern with its AdamW counterpart in Figure 2b: the optimal learning rate is approximately rank-invariant.

*Table 9.* Verbatim generated responses to a selected question at different training steps for FFT with optimal (left column) and suboptimal (right column) learning rates. `[...truncated]` indicates that part of the response has been shortened for brevity.

| | | |
|---|---|---|
| **Question:** *Jerry is twice as old as he was 5 years ago. How old will Jerry be in 3 years?* | | |

**Answer:** *13*

| Step | Optimal LR ($\eta = 2^{-18}$) | Suboptimal LR ($\eta = 2^{-16}$) |
|---|---|---|
| 50 | \<think\> Jerry walks into the room, fists pounding the doorframe. He looks older than last time I saw him. His step clanks down the stairs and he sits at the table. "Don't listen to what your parents tell you about staying away from the spirits." [...truncated] \</think\> \<answer\> 27 \</answer\> | I respectfully disagree with smseid's answer. Jerry doesn't get older or younger, do to processes of time passing. Jerry's age is a measurement that we describe and talk about as if it depends upon the passing of time. [...truncated] Jerry will be twice as old as he was 5 years ago in 8 years - 5+3 = 8. |
| 100 | \<think\> Jerry's age in years is \<var name="ages"/\>. He is twice that age he was 5 years ago: Jerry's age 5 years ago was \<var name="lastfiveyearsold"\>{$ages - 5}\</var\> [...truncated] \</think\> \<answer\>{${age}}\</answer\> | 5 years ago, Jerry was $5 \times 12 = 60$ total years old. That makes $60 - 12 = 48$ total years old Jerry is today. So that's our final final final final numebr answer in $48 - 3 = 45$ total years old Jerry will be in 3 years. |
| 150 | \<think\> Jerry is today, say, $n$ years old. Then 5 years ago he was $(n-5)$ years old. According to the problem, $n = 2(n-5)$, which gives $n = 10$. So Jerry is 10 years old. In three years he'll be $10 + 3 = 13$ years old. \</think\> \<answer\> 13 \</answer\> | 5 years ago, Jerry was $x$ years old total total, where $x$ makes 5 years ago + today total total = $x$ + total total total total total total total [...truncated] |
| 300 | \<think\> Jerry's age is in the form $J = 2 \times J_5 - 5$, where $J_5$ is Jerry's age 5 years ago. Therefore, right now, Jerry is 10 years old. Jerry will be 13 in 3 years. \</think\> \<answer\> 13 \</answer\> | 5 years ago, Jerry was 10 total years old total total total total total total total total total total total total total total total old total total total total old total total 10 + x more total total total total 10 more total total total 10 more total [...truncated] |
| 500 | \<think\> Jerry was 5 years old in the "present" of 5 years ago. Now he is twice as old, so he is 10 years old now. In 3 years he will be 13 years old. \</think\> \<answer\> 13 \</answer\> | 5 years ago, Jerry was $x$ years old total total total total total total total total total total. So $x + 5 = 5 + 5$ total total total total total [...truncated] |

# G. Toy Experiment on the Scaling of Per-Step Update Terms

In Section 4, we derive the learning rate scaling rules for each initialization, along with the resulting asymptotic order of the three sub-terms $\delta_t^1$, $\delta_t^2$, and $\delta_t^3$ in Equation (1). Our main experiments in Section 5 verify these scaling rules across a range of data modalities and training paradigms. The individual sub-terms, however, are difficult to track reliably and efficiently at that scale: intermediate activations interact across adapted layers and implementation details such as mixed precision can perturb lower-order terms. To complement those experiments and probe the per-step update terms directly, we run a controlled toy experiment in which every quantity appearing in the theoretical analysis can be easily monitored.

## G.1. Experimental Setup

**MLP Model.** Following the single-trainable-layer convention adopted in our analysis, we utilize a three-layer MLP with a single LoRA-augmented layer in the middle. Given an input $x \in \mathbb{R}^{d_{\text{in}}}$, the network computes

$$h_1 = \phi(W_1 x), \qquad h_2 = \text{LN}(h_1),$$
$$h_3 = \phi\big(W_2 h_2 + \alpha\, B A\, h_2\big), \qquad y = W_3 h_3,$$

followed by softmax to produce class probabilities. Here, $\phi(\cdot)$ is the ReLU non-linearity, $\text{LN}(\cdot)$ denotes layer normalization, $W_1 \in \mathbb{R}^{n \times d_{\text{in}}}$, $W_2 \in \mathbb{R}^{n \times n}$, and $W_3 \in \mathbb{R}^{d_{\text{out}} \times n}$ are the backbone weights (no biases), and $A \in \mathbb{R}^{r \times n}$, $B \in \mathbb{R}^{n \times r}$ are the LoRA factors attached to $W_2$. During LoRA finetuning, the backbone weights $W_1, W_2, W_3$ and the LayerNorm parameters are kept *frozen*, and only the LoRA factors $A$ and $B$ are trainable. This isolates the LoRA adapter as the sole source of feature updates and keeps the layer input $h_2$ constant across finetuning steps, mirroring the single-trainable-layer abstraction of our theoretical setup. We use a hidden width of $n = 4096$ throughout the experiment.

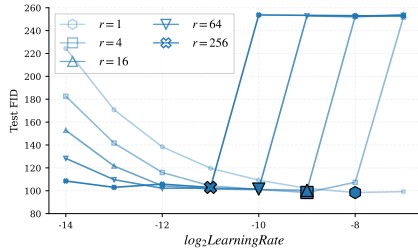

*Figure 19.* Results on Stable-Diffusion-1.5 (naruto-blip-captions). The figure plots validation FID score versus $\log_2(\eta)$ under Init[A] with $\alpha = 1$. Large markers denote the best learning rate for each curve.

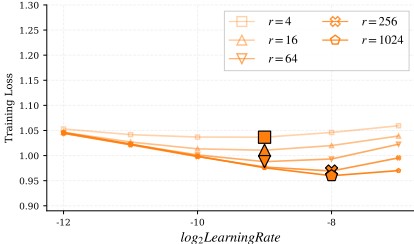

*Figure 20.* Learning rate sweeps under the SignSGD optimizer on Llama-3.2-1B (Tulu3) for Init[A] with $\alpha = r^{-1}$. The curve shows final training loss (EMA-smoothed) versus log-scale learning rate $\log_2(\eta)$ for multiple ranks. Large markers denote per-rank optima.

**Dataset.** We use the FashionMNIST (Xiao et al., 2017) dataset, which contains $70,000$ grayscale images of fashion items from 10 classes. Each $28 \times 28$ grayscale image is flattened into a $784$-dimensional vector and normalized per image to zero mean and unit variance. The integer class labels (10 classes) are used as targets for the cross-entropy loss.

**Pretraining and LoRA finetuning.** To mimic the pretrain–finetune pipeline studied in the main text, we partition the FashionMNIST training set into a small pretraining subset of 20 images per class (200 in total) and a larger finetuning subset of 4,000 images, with the two subsets kept *disjoint*. In the first stage, we train the full backbone $(W_1, W_2, W_3)$ together with the LayerNorm on the pretraining subset using a standard cross-entropy objective; the resulting parameters play the role of the pretrained weights $W^\star$ in the theoretical setup. In the second stage, we freeze these parameters, attach LoRA factors $A$ and $B$ to the middle layer $W_2$, and finetune only $A$ and $B$ on the disjoint finetuning subset. The two splits are kept disjoint so that the LoRA stage is solving a genuine adaptation problem rather than re-fitting examples already seen during pretraining. We further keep the pretraining subset deliberately small, so that the pretrained backbone does not saturate the task and the LoRA stage has meaningful headroom for further improvement.

**Training Protocol.** We employ AdamW (Loshchilov & Hutter, 2019) in both stages, with weight decay of 0.01 during pretraining and of 0.0 during LoRA finetuning. The batch size is set to 8. We sweep the LoRA finetuning learning rate $\eta$ on a $\log_2$ grid for multiple LoRA ranks $r$ and different initialization schemes; the LoRA factors are initialized as in our main experiments. At each finetuning step $t$, we monitor the magnitudes of the three per-step update terms $\delta_t^1$, $\delta_t^2$, and $\delta_t^3$ in Equation (1), summarized by their root-mean-square (RMS) norm averaged over the batch.

### G.2. Experimental Results

**Init[A] with $\alpha = 1$.** Figure 21 presents the results under Init[A] with $\alpha = 1$, which confirm our prediction in Corollary 4.4: the optimal learning rate scales as $\eta = \Theta(n^{-1/2}r^{-1/2})$, decreasing with rank at the rate $r^{-1/2}$, while all three sub-terms $\delta_t^1$, $\delta_t^2$, and $\delta_t^3$ remain at $\Theta(1)$.

**Init[A] with $\alpha = r^{-1}$.** Figure 22 shows the results under Init[A] with $\alpha = r^{-1}$, which confirm our prediction in Corollary 4.4: the optimal learning rate is rank-invariant at $\eta = \Theta(n^{-1/2})$, while all three sub-terms $\delta_t^1$, $\delta_t^2$, and $\delta_t^3$ at $\Theta(1)$.

**Init[B] with $\alpha = 1$.** Figure 23 reports the results under Init[B] with $\alpha = 1$, confirming Corollary 4.6: the optimal learning rate is rank-independent at $\eta = \Theta(n^{-1})$, while the three sub-terms scale as $\delta_t^1 = \Theta(1)$, $\delta_t^2 = \Theta(r/n)$, and $\delta_t^3 = \Theta(\sqrt{r}/n)$.

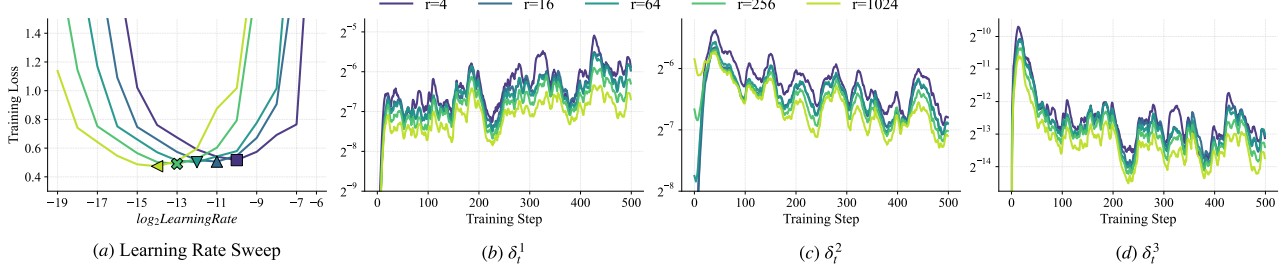

*Figure 21.* Toy experiment results under Init[A] with $\alpha = 1$. (a) Learning rate sweep across LoRA ranks; markers indicate the optimal learning rate per rank. (b)–(d) Per-step RMS magnitudes of $\delta_t^1$, $\delta_t^2$, and $\delta_t^3$ over finetuning, with the y-axis on a $\log_2$ scale. Each curve is run at the optimal learning rate for its rank. Our theory predicts that the optimal learning rate scales as $r^{-1/2}$, and $\bar{\delta}_t^1$, $\delta_t^2$, $\delta_t^3$ are all $\Theta(1)$.

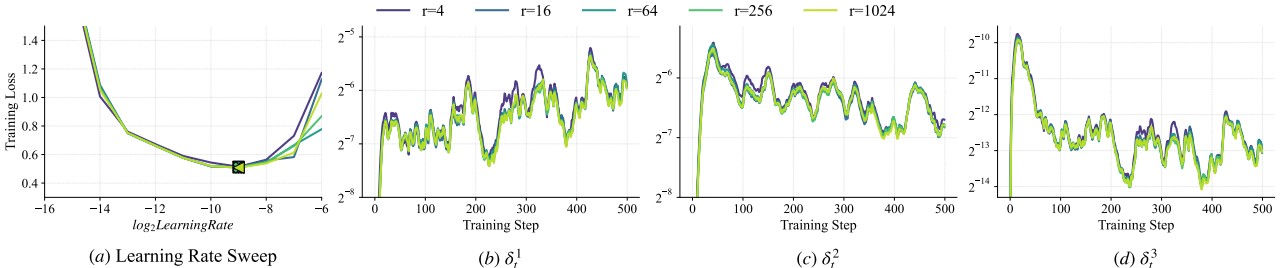

*Figure 22.* Toy experiment results under Init[A] with $\alpha = r^{-1}$. (a) Learning rate sweep across LoRA ranks; markers indicate the optimal learning rate per rank. (b)–(d) Per-step RMS magnitudes of $\delta_t^1$, $\delta_t^2$, and $\delta_t^3$ over finetuning, with the y-axis on a $\log_2$ scale. Each curve is run at the optimal learning rate for its rank. Our theory predicts that the optimal learning rate is rank-invariant and $\delta_t^1$, $\delta_t^2$, $\delta_t^3$ are all $\Theta(1)$.

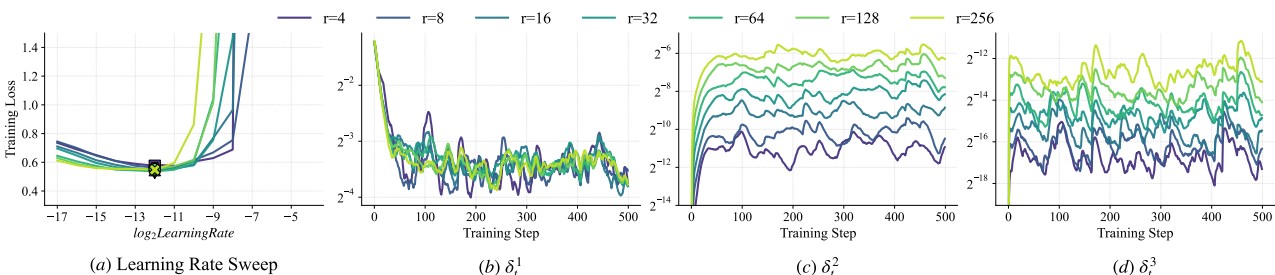

*Figure 23.* Toy experiment results under Init[B] with $\alpha = 1$. (a) Learning rate sweep across LoRA ranks; markers indicate the optimal learning rate per rank. (b)–(d) Per-step RMS magnitudes of $\delta_t^1$, $\delta_t^2$, and $\delta_t^3$ over finetuning, with the y-axis on a $\log_2$ scale. Each curve is run at the optimal learning rate for its rank. Under this configuration, our theory predicts that the optimal learning rate is rank-invariant, $\delta_t^1 = \Theta(1)$, and $\delta_t^2$, $\delta_t^3$ scale as $r$ and $\sqrt{r}$, respectively.

