# OpenReview forum: "Learning Rate Scaling across LoRA Ranks and Transfer to Full Finetuning"
_ICML.cc/2026/Conference — ICML 2026 regular_

### Official Review · Reviewer_DBsG · 2026-02-15

**Soundness:** 3
**Presentation:** 2
**Significance:** 3
**Originality:** 3
**Overall Recommendation:** 4
**Confidence:** 3

**Summary:**

This paper studies how the optimal learning rate of LoRA depends on rank and initialization, and whether it can transfer to full fine-tuning. The authors propose Maximal-Update Adaptation to analyze the feature update dynamics of LoRA by decomposing the output update into three terms and requiring stable feature learning. Under this framework, they derive scaling laws for the learning rate as a function of model width and LoRA rank, and identify two regimes: rank-dependent scaling and rank-invariant scaling. Extensive learning-rate sweeps across language, vision, multimodal, diffusion, and reinforcement learning tasks are used to validate these trends empirically.

**Compliance With Llm Reviewing Policy:**

Affirmed.

**Final Justification:**

The authors provided clarification, which properly addressed my concerns. I decide to keep a score of 4.

**Key Questions For Authors:**

1. How sensitive are the theoretical scaling predictions to the choice of optimizer in practice? Would the same rank-dependent vs. rank-invariant regimes appear under non-Adam optimizers?

2. The experiment part uses linear warmup with cosine decay. Do the derived scaling laws remain robust under different scheduling strategies?

3. The authors mention "Finite-size effects. When (n, r) are finite, constant factors and lower-order terms can shift the optimal learning rate. These shifts are more pronounced for smaller ranks, where the large-r asymptotics are less accurate." Does this apply for larger $n$ case? Do the authors observe a similar effect on $n$ during the experiment?

**Limitations:**

yes

**Strengths And Weaknesses:**

## Strengths

1. The paper starts from a practical pain point that learning rate must often be re-tuned when changing LoRA rank and confirm that optimal learning rates vary significantly across rank and design choices.

2. The framework offers an intuitive and structured way to reason about feature update stability, and analyze their scaling behavior. The resulting rank-dependent vs. rank-invariant is simple, interpretable, and explanatory.

3. The finding that Init[B] with $\alpha = 1$ yields the same width scaling as FT (i.e., $\eta \propto n^{-1}$) is practically valuable, as it suggests a configuration where learning rates may transfer from LoRA to FT without additional sweeping.

4. The scaling trends are examined across multiple domains and model families, including reinforcement learning, where the choice of $\eta$ also affects training duration. This breadth strengthens the claim that the observed phenomena are not task-specific.

## Weaknesses

1. The analysis relies on joint limits ($n, r \to \infty$), fixed training steps, and a SignSGD approximation of Adam. While standard in scaling-law analysis, these assumptions may limit how precisely the results translate to finite-width, long-horizon training with AdamW.

2. Although the theory abstracts Adam as SignSGD, experiments use AdamW exclusively. The paper does not study whether the same rank-dependent vs. rank-invariant behavior holds under other optimizers (e.g., SGD, Adafactor), leaving open questions about generality.

3. The scaling of the multiplicative term $\delta_3^t$ in Appendix A.3 (Lemma A.17) relies on a strong independence assumption between $\operatorname{sign}(Z_A^{t-1})$ and $\operatorname{sign}(\alpha B_{t-1}^\top d\bar Z^{t-1})$, which directly yields an inner product of order $\Theta(\sqrt r)$; however, since these quantities are generated by the same training dynamics, their independence is not obvious in practice, and deviations from this assumption could alter the dominant term and thus the derived learning-rate scaling law.

---

> ### Author Rebuttal · Authors · 2026-03-30
>
> Thank you for your valuable feedback. Due to space limitations, we refer to each question/weakness by number. Additional experimental results are provided in the figures via anonymous links below, discussed in detail throughout our response.
>
> - Fig. 1: https://sub-28479.b-cdn.net/SignSGD.png
> - Fig. 2: https://sub-28479.b-cdn.net/Muon.png
> - Fig. 3: https://sub-28479.b-cdn.net/mlp.png
> - Fig. 4: https://sub-28479.b-cdn.net/ConstantLR.png
>
> **W1**:
>
> We adopt these assumptions to ensure tractability and to obtain closed-form expressions. We thought the fixed training steps assumption is natural for LLM finetuning, which usually runs for a limited number of steps. We will discuss the SignSGD abstraction in our response to W2&Q1 and finite-size effects in our response to Q3 in the following.
>
> **W2 & Q1**:
>
> We adopt SignSGD as an abstraction of AdamW because it yields closed-form expressions (see also [1,2]). We hypothesize that the scaling rules derived under SignSGD remain valid under AdamW because SignSGD captures a property of AdamW most relevant to scaling analysis: **element-wise gradient normalization**. In AdamW, the adaptive second-moment denominator rescales each gradient coordinate so that the effective update magnitude is approximately $\Theta(1)$ with respect to width and rank, regardless of the raw gradient scale. SignSGD is basically AdamW without momentum. We acknowledge that AdamW's momentum and second-moment estimates additionally affect training dynamics, but argue that these components primarily smooth the per-element updates without altering the $\Theta(1)$ scaling of each update element. In an additional experiment replacing AdamW with SignSGD (see Fig. 1), we observe that the learning rate scaling patterns remain consistent, suggesting that momentum is not the dominant factor governing the scaling rules. SignSGD thus serves as a tractable, empirically justified proxy for AdamW in our derivations.
>
> Note that the SignSGD abstraction in our framework is specifically tailored to Adam-like optimizers. Other optimizer families impose different gradient normalization structures and would require separate abstractions. For instance, SGD uses raw unnormalized gradients, LARS [3] enforces a unit $\ell_2$-norm per layer, and Muon [4] enforces a unit spectral norm. For Adafactor [5], we also need to consider the relative step size used. These distinct normalization structures will alter the asymptotic scaling of feature updates. For example, in an additional experiment replacing AdamW with Muon (see Fig. 2), we observe a visibly different scaling pattern, confirming that a separate analysis would be needed. Importantly, our $\mu A$ framework is modular: Assumption 4.2 can be replaced with an appropriate counterpart for other optimizer families, and the same analysis pipeline can be reused to derive the corresponding scaling rules. We consider this a promising direction for future work, but beyond the scope of this paper.
>
> **W3**:
>
> For Lemma A.17, we only need $\langle\text{sign}(Z_A^{t-1}),\text{sign}(\alpha B_{t-1}^\top d\bar{Z}^{t-1})\rangle$ to scale as $r^{1/2}$, which is weaker than independence. We will revise the manuscript to reflect this point. Note that this approximation is used only to characterize the asymptotic scaling of $\delta^3$, without direct implications for the $\mu A$ scaling rules. Theoretically, even without the independence assumption, the worst case only changes the $r^{1/2}$ factor in the scaling of $\delta^3$ to $r$, and $\delta^3$ is still bounded by $\delta^2$; therefore, the results in Theorems A.18 and A.20 keep unchanged. To further validate this empirically, we train a 3-layer MLP with LoRA on FashionMNIST (Init[B], $\alpha=1$). As shown in Fig. 3 (d), $\delta^3$ remains orders of magnitude smaller than $\delta^1$ and $\delta^2$ across ranks. This was also discussed in [2].
>
> **Q2**:
>
> To measure the robustness of the derived scaling rules under different learning rate scheduling strategies, we carry out an additional experiment using a constant learning rate schedule (see Fig. 4). The results exhibit a pattern that aligns well with our $\mu A$ scaling rules.
>
> **Q3**:
>
> In practice, finite-size effects primarily impact smaller ranks. For width $n$, as noted in [2], infinite-width asymptotics hold well for practical models. Since all models in our experiments have large widths, we do not expect a similar effect for $n$, and our experiments show no indication of such effect.
>
> [1] Bernstein et al. (2018). signSGD: Compressed Optimisation for Non-Convex Problems.
>
> [2] Hayou et al. (2024). Lora+: Efficient low rank adaptation of large models.
>
> [3] You et al. (2017). Scaling sgd batch size to 32k for imagenet training.
>
> [4] Jordan et al. (2024). Muon: An optimizer for hidden layers in neural networks.
>
> [5] Shazeer et al. (2018). Adafactor: Adaptive Learning Rates with Sublinear Memory Cost.

---

> > ### Author Rebuttal · Reviewer_DBsG · 2026-04-01
> >
> > Thanks to the authors for the clarification and further experiments, which properly addressed my concerns. I decide to keep my positive score.

---

> > > ### Author Response · Authors · 2026-04-05
> > >
> > > Thank you for considering our rebuttal and for the positive evaluation of our work. We greatly appreciate your thoughtful feedback and are glad that the clarifications and additional experiments addressed your concerns. We will carefully incorporate your comments to further improve the future version.

---

### Official Review · Reviewer_Q1HH · 2026-03-07

**Soundness:** 3
**Presentation:** 3
**Significance:** 3
**Originality:** 3
**Overall Recommendation:** 5
**Confidence:** 4

**Summary:**

this paper seeks to outline a fundamental problem that practitioners face when changing LoRA rank: the optimal learning rate shifts in ways that are not well understood, forcing expensive re-tuning. The paper makes a genuine contribution by providing a rigorous theoretical framework (µA) that characterizes how the optimal learning rate scales with both model width and adapter rank, and validates these rules extensively across diverse tasks and modalities. The paper is well-motivated, theoretically grounded, and empirically thorough. However, I have several questions which is elaborated below.

**Compliance With Llm Reviewing Policy:**

Affirmed.

**Final Justification:**

the authors have addressed my concerns and promised to add more diffusion model analysis.

**Key Questions For Authors:**

see comments in weakness. i would increase the score if the authors address my concerns in weakness.

**Limitations:**

yes

**Strengths And Weaknesses:**

# Strengths

## 1. Theoretical contributions

The paper makes a clear theoretical contribution by analyzing LoRA in the *joint* limit n, r → ∞. The three-term decomposition of feature updates (δ¹, δ², δ³) and the term-wise asymptotic analysis are clean and provide mechanistic insight into why different configurations behave differently.

## 2. comprehensive experiments

The empirical coverage is impressive. The authors validate their theory across five model architectures (decoder-only LMs, encoder-only models, vision-language models), three training paradigms (SFT, RLVR, diffusion finetuning), and multiple modalities. This breadth significantly strengthens confidence in the generality of the µA scaling rules.

## 3. Practical Impact

The finding that a learning rate tuned cheaply with LoRA (Init[B], α=1) can transfer to full finetuning is highly valuable. Full finetuning requires more GPU memory compared to LoRA (as demonstrated in Figure 6d), so reducing the need for FFT-specific hyperparameter sweeps meaningfully lowers computational cost and carbon footprint. The paper makes this workflow concrete and actionable.


# Weakness

## 1. gap between theory and practice

The theoretical analysis relies on SignSGD as a proxy for Adam. While this is a common simplification in literature, it is a significant abstraction. Adam's adaptive per-parameter learning rates, momentum terms, and second-moment estimates all interact with the LoRA bilinear structure in ways that SignSGD does not capture. The paper acknowledges this but does not provide any theoretical justification for why the scaling rules derived under SignSGD should remain valid under Adam. This represents a potentially important gap.

## 2. single layer assumption

The theoretical analysis assumes only a single LoRA layer is trainable while all other components are frozen. In practice, LoRA is applied to many layers simultaneously across a deep network, and the interaction between multiple adapters can produce non-trivial dynamics. The paper does not discuss whether or how the single-layer analysis might be extended to multi-layer settings, or provide any theoretical justification for why single-layer results should propagate to the full network. The empirical success across multi-layer models is encouraging but does not close this gap.


## 3. scope of FFT Transfer Analysis

The paper focuses on learning rate transfer from LoRA to FFT, but does not examine *performance* transfer — i.e., whether the model trained with the transferred learning rate achieves comparable final task performance to one trained with an FFT-tuned learning rate. The experiments in Figure 5 demonstrate alignment of optimal learning rates, but do not directly report downstream task metrics (instead of training loss) comparing LoRA-to-FFT transfer against FFT with its own tuned rate. This would strengthen the practical claims considerably.


## 4. Width Scaling Not Empirically Validated

The µA framework predicts specific width-dependent scaling (η **∝** n⁻¹ for Init[B], η **∝** n⁻¹/² for Init[A] with α = r⁻¹). While rank-dependent scaling is extensively validated, the width-dependent component is not directly tested — for example, by comparing optimal learning rates across models of different widths (e.g., Llama-1B vs Llama-8B). Without this, the n-dependent component of the scaling rules remains empirically unverified.

## 5. Finite-Size Effects and Small Rank Behavior

The paper notes that finite-size effects are more pronounced at smaller ranks, and the theory is asymptotic in (n, r) → ∞. However, small ranks (r = 1, 4) are commonly used in practice, especially in RLVR settings. The deviations from theory at small ranks (e.g., r = 4 under Init[A] with α = r⁻¹ favoring a slightly smaller learning rate than predicted) are attributed to "finite-size effects" without quantifying their magnitude or providing guidance on how practitioners should correct for them.

## 6. Diffusion Model Analysis is Relatively Thin

While the diffusion model experiments are included to demonstrate breadth, the analysis for this setting is comparatively shallow. The UNet architecture has heterogeneous block widths, which the paper notes may amplify FFT sensitivity, but the theoretical implications of non-uniform width are not explored. The claim that µA scaling rules generalize to diffusion models would benefit from deeper treatment.

---

> ### Author Rebuttal · Authors · 2026-03-30
>
> Thank you for your valuable feedback. Due to space limitations, we refer to each question/weakness by number. Additional experimental results are provided in the figures via anonymous links below, discussed in detail throughout our response.
>
> - Fig. 1: https://sub-28479.b-cdn.net/SignSGD.png
> - Fig. 2: https://sub-28479.b-cdn.net/MMLU.png
>
> **W1**:
>
> We adopt SignSGD as an abstraction of AdamW because it yields closed-form expressions (see also [1,2]). We hypothesize that the scaling rules derived under SignSGD remain valid under AdamW because SignSGD captures a property of AdamW most relevant to scaling analysis: **element-wise gradient normalization**. In AdamW, the adaptive second-moment denominator rescales each gradient coordinate so that the effective update magnitude is approximately $\Theta(1)$ with respect to width and rank, regardless of the raw gradient scale. SignSGD is basically AdamW without momentum. We acknowledge that AdamW's momentum and second-moment estimates additionally affect training dynamics, but argue that these components primarily smooth the per-element updates without altering the $\Theta(1)$ scaling of each update element. In an additional experiment replacing AdamW with SignSGD (see Fig. 1), we observe that the learning rate scaling patterns remain consistent, suggesting that momentum is not the dominant factor governing the scaling rules. SignSGD thus serves as a tractable, empirically justified proxy for AdamW in our derivations.
>
>
> **W2**:
>
> We adopt the single-layer assumption to isolate the LoRA feature learning in a single layer and to make the theoretical analysis tractable, following standard practices (see e.g. [2,3]). We agree that applying LoRA to multiple layers can introduce non-trivial dynamics. However, for a network of fixed depth, our forward stability argument intuitively extends layer by layer. Because the first adapted layer receives a fixed input, our scaling rules ensure that its LoRA update remains bounded over a fixed training horizon. As a result, the next layer continues to receive an order-one input, allowing the same asymptotic analysis to apply. This logic cascades, keeping forward signals controlled across the full network. While our multi-layer LoRA experiments support this intuition, a rigorous theoretical proof also requires controlling backward gradient signals across layers, which we leave for future work.
>
>
> **W3**:
>
> We evaluate Llama-3.2-1B finetuned with FFT at various learning rates on the MMLU benchmark (see Fig. 2). The optimal learning rate identified by training loss achieves near-optimal MMLU accuracy, confirming that training loss is a reliable proxy for downstream task performance. Notably, the LoRA-to-FFT transferred learning rate ($log_2 \eta = −15$, obtained with Init[B] at $r=256$) achieves the best MMLU accuracy, indicating that transfer extends to downstream tasks.
>
>
> **W4**:
>
> The primary novelty of $\mu A$ lies in the rank-dependent scaling rules, which our experiments validate extensively. Width scaling appears in our prescriptions for theoretical completeness, but the width-dependent component itself is not a novel contribution of this work: when rank is held constant, our width-dependent prescriptions coincide with the results derived and empirically verified in [3]. This prior work includes a toy experiment with a controlled width sweep verifying the predicted width scaling (see Figure 2 of [3]).
>
> Regarding a same-family width sweep on pretrained models (e.g., Llama-1B vs. 8B): available pretrained models within a single family differ not only in width but also in depth, pretraining data, and other factors. This falls outside the scope of our theory, which holds only when all factors are fixed and only the scalable dimension (width in this case) grows. Nonetheless, we believe that the extensive rank-scaling experiments across diverse model architectures provide sufficient empirical supports for the $\mu A$ prescriptions, and we refer interested readers to [3] for direct width-scaling validation.
>
>
> **W5**:
>
> At smaller ranks, the observed deviations in the experiment remain within 2x (one $log_2$ grid step), i.e., at the sweep's resolution limit. We recommend practitioners sweep a small neighborhood around the predicted learning rate to correct for finite-size effects.
>
> **W6**:
>
> We agree that the diffusion model analysis is primarily empirical. As noted in Section 5.4, the UNet architecture features heterogeneous block widths. Built on the single-layer discussion in W2, a natural theoretical extension is to interpret the $\mu A$ scaling rules locally, utilizing each linear projection's specific width and rank. We will add this discussion in the future version.
>
> [1] Bernstein et al. (2018). signSGD: Compressed Optimisation for Non-Convex Problems.
>
> [2] Hayou et al. (2024). Lora+: Efficient low rank adaptation of large models.
>
> [3] Hayou et al. (2024). The Impact of Initialization on LoRA Finetuning Dynamics.

---

> > ### Author Rebuttal · Reviewer_Q1HH · 2026-04-02
> >
> > Thanks to the authors for the detailed rebuttal and for addressing my concerns. I appreciate the clarifications. I have no further questions. I am willing to increase my score.

---

> > > ### Author Response · Authors · 2026-04-05
> > >
> > > Thank you for your willingness to increase the score. We greatly appreciate your thoughtful and constructive feedback, which has helped strengthen our work. If convenient, we would also be grateful if you could update the score to reflect your revised assessment. We will revise our paper to expand the diffusion model analysis and carefully incorporate your other comments.

---

### Official Review · Reviewer_pgWT · 2026-03-12

**Soundness:** 1
**Presentation:** 3
**Significance:** 2
**Originality:** 2
**Overall Recommendation:** 3
**Confidence:** 4

**Summary:**

This paper proposes Maximal-Update Adaptation ($\mu$A), a theoretical framework which studies how the optimal learning rate for LoRA finetuning (FT) scales with adapter rank and model width in limit regime, inspired by $\mu$P. The core contribution is the chracterization of the asymptotic scaling of feature updates under different combinations of initialization (Init[A]: random A, B=0 vs. Init[B]: random B, A=0) and LoRA multiplier $\alpha$. The analysis reveals two distinct regimes with specific optimal learning rate scale. Crucially, Init[B] with $\alpha=1$ yields a learning rate scaling, matching that of full fine-tuning (FFT), which shows the potential of tuning the learning rate with LoRA and transferring it to FFT on the same model.

**Compliance With Llm Reviewing Policy:**

Affirmed.

**Final Justification:**

The authors confirm that they will change framing to informal.

**Key Questions For Authors:**

1. I have a concern about the validity of the theoretical results as currently stated. The main claims in Theorem 4.3 / Corollary 4.4 and Theorem 4.5 / Corollary 4.6 are presented under only Assumptions 4.1 and 4.2, but the appendix proofs rely on additional unstated assumptions:

   - In lines 935-940, proof of Lemma A.14, the paper states:
   > For the ease of analysis, we drop the subscript in $B_{t-1}$ and instead use $B$ with $B = \Theta(\beta)$. In addition, we use $h = \alpha d\bar{Z}^{t-1} \in \mathbb{R}^n$ with $\Theta(h) = \Theta(\alpha)$. Let $X = B \cdot \operatorname{sign}(B^\top h)$, we aim to analyze the asymptotic dynamic of $X$ by taking an arbitrary element $X_k$ and analyze $\mathbb{E}[X_k^2]$. To simplify the analysis, we treat $B \in \mathbb{R}^{n \times r}$ as an i.i.d. Gaussian matrix with entries $B_{ij} \overset{\text{i.i.d.}}{\sim} \mathcal{N}(0, \beta^2)$, and we assume $h \in \mathbb{R}^n$ is independent of $B$ with entries $h_i \overset{\text{i.i.d.}}{\sim} \mathcal{N}(0, \alpha^2)$.
   - In line 1045, proof of Lemma A.17, the paper states:
   > Here we assume that elements in $\operatorname{sign}(Z_A^{t-1})$ and $\operatorname{sign}(\alpha B^\top d\bar{Z}^{t-1})$ are independent.

   These lemmas are then used directly to derive the Init[A] and Init[B] theorems/corollaries. Neither $B_{t-1}$ as i.i.d. Gaussian matrix nor independence between $B_{t-1}$ and $\alpha d\bar{Z}^{t-1}$ holds in general.
   - For Init[A], $B_{t-1}$ is a sum of rank-1 matrix which has rank at most (t-1). The column/rows are highly correlated when t is small.
   - For both initializations at $t\geq 2$, $d\bar{Z}^{t-1}$ depends on $B_{t-1}$ so independence fails.
   - The only edge case is Init[B] at t=1, where $B_0$ is initialized from Gaussian.

   These assumptions are stronger than the assumptions stated in the main text. As a result, I do not think the current proofs establish the main theorem/corollary as stated. I would regard this as a **technical flaw** for now. If these are essential assumptions which cannot be proven, I think they should be surfaced explicitly in the main context and change frame to be "Result" instead of "Theorem/Corollary"; if they can be proven to hold in certain regime, the authors should provide rigorous proofs.

   (I will increase my score if a convincing clarification here can be provided.)

2. Strong simplifying assumptions limit theoretical generality. The theory relies on:
   - SignSGD as a proxy for Adam (Assumption 4.2), discards momentum and adaptive second-moment estimation, which might alter the training dynamics,
   - a single-sample loss, and a single trainable LoRA layer with all other parameters frozen,

   reduce the analysis to a tractable but impractical setting.

3. All experiments are only run for one fixed seed 42 (line 1416). Since exact optima can move on noisy tasks, it would be very helpful to know whether the observed theory-experiment alignment is stable across random seeds (mean+std+multi-run).

4. Can you add a controlled width-scaling experiment within a single model family? The theoretical part also covers width scaling but the current empirical support is indirect. A same-family width sweep (even toy transformer models) would make the paper’s core theoretical claim much stronger.

**Limitations:**

yes

**Strengths And Weaknesses:**

**Strength**

1. This paper's focused problem is well-motivated. LoRA has more parameters (rank, scale) to tune than FFT. The paper tackles this from a unified theoretical perspective.
2. The breadth of experiments is a strength. The paper covers five distinct task domains (instruction tuning, long-context reasoning, NLI, image classification, vision-language, diffusion finetuning, and RLVR), multiple model architectures (decoder-only, encoder-only, vision transformers, U-Net), and three LoRA configurations.

**Weakness**

1. The appendix proofs contain additional strong assumptions without justification, which might affect the rigor of main theoretical results.
2. Strong simplifying assumptions limit theoretical generality.
3. All experiments are only run for one fixed seed.
4. Lack of sufficient empirical evidence to support width scaling results.

---

> ### Author Rebuttal · Authors · 2026-03-30
>
> We thank the reviewer for the careful reading of our work. Due to space limitations, we refer to each question/weakness by number, and cross-reference our responses to other reviewers; we apologize for any inconvenience this may cause. Additional exps are provided via anonymous links below.
> - Fig. 1: https://sub-28479.b-cdn.net/mlp.png
> - Fig. 2: https://sub-28479.b-cdn.net/MSeed.png
>
> **W1 & Q1**:
>
> We agree that Theorem 4.3/4.5 and Corollary 4.4/4.6 should be stated as "informal". We made the following changes:
> - Present the main claims as informal (similar to [8]).
> - Emphasize that the results are derived under some approximations at the physics level of rigor for tractability.
>
> We added the following to justify the approximations:
>
> In Lemma A.14, we use Gaussian approx for $B_{t-1}$ and $d\bar{Z}^{t-1}$, with an independence approximation, to estimate the second-moment of $B_{t-1} \cdot \text{sign}(\alpha B_{t-1}^\top d\bar{Z}^{t-1})$. The independence approximation is widely used in the theory of signal propagation (see e.g. [2,3,4,5,9]) to obtain closed-form results, e.g. [4] used it to estimate the spectrum of the network Jacobian, and showed excellent match with real observations. [7,10] showed that activations and gradients become approximately i.i.d. as width grows, and that an assumption of independence between forward and backward propagation weights leads to exact computations in the infinite-width limit. They show that the independence approximation works well because the randomness from initialization persists during training. Such approximations have also been widely used in Approximate Message Passing [6] and statistical mechanics of NNs [11]. To further validate this empirically, we train a 3-layer MLP with LoRA on FashionMNIST (Init[B], $\alpha=1$). As shown in Fig. 1(b), the empirical $\delta^1$ scaling across ranks matches the theoretical prediction $\delta^1=\Theta(1)$, showing that the approximation can lead to the right $\delta^1$ scaling.
>
> For Lemma A.17, we only need $\langle\text{sign}(Z_A^{t-1}),\text{sign}(\alpha B_{t-1}^\top d\bar{Z}^{t-1})\rangle$ to scale as $r^{1/2}$, which is weaker than independence. Note that this approximation is used only to characterize the asymptotic scaling of $\delta^3$, without direct implications for the $\mu A$ scaling rules. Theoretically, even without the independence assumption, the worst case only changes the $r^{1/2}$ factor in the scaling of $\delta^3$ to $r$, and $\delta^3$ is still bounded by $\delta^2$; therefore, the results in Theorems A.18 and A.20 keep unchanged. Empirically, Fig. 1(d) shows that $\delta^3$ remains orders of magnitude smaller than $\delta^1$ and $\delta^2$ across ranks. This was also discussed in [1].
>
> **W2 & Q2**:
>
> We use these assumptions to obtain closed-form expressions, following standard practices (see e.g. [1,5,12]).
> - For the SignSGD abstraction and the single-sample loss, please kindly refer to our response to Reviewer eUPW's W1 & Q1.
> - For the single-layer assumption, please kindly refer to our response to Reviewer Q1HH's W2.
>
> **W3 & Q3**:
>
> A full multi-seed sweep is expensive, e.g. one Qwen2.5-3B-Instruct run takes about 6.5 H200 GPU-hours. We use a unified training recipe, including learning-rate warmup and decay, to make the runs stable. To assess whether the theory–experiment alignment is stable across random seeds, **we additionally run Llama-3.2-1B on Tulu3 (Init[A], $\alpha=1/r$) with three random seeds (see Fig. 2)**. The qualitative rank-scaling trend is stable across seeds and consistent with our derived $\mu A$ scaling rules.
>
> **W4 & Q4**:
>
> Please kindly refer to our response to Reviewer Q1HH's W4, where width scaling is discussed in detail.
>
> [1] Hayou et al. (2024). Lora+: Efficient low rank adaptation of large models.
>
> [2] Schoenholz et al. (2017). Deep Information Propagation.
>
> [3] Xiao et al. (2018). Dynamical isometry and a mean field theory of cnns: How to train 10,000-layer vanilla convolutional neural networks.
>
> [4] Pennington et al. (2017). Resurrecting the sigmoid in deep learning through dynamical isometry: theory and practice.
>
> [5] Hayou et al. (2019). On the Impact of the Activation function on Deep Neural Networks Training.
>
> [6] Berthier et al. (2020). State evolution for approximate message passing with non-separable functions.
>
> [7] Yang et al. (2021). Feature Learning in Infinite-Width Neural Networks.
>
> [8] Yang et al. (2024). Tensor Programs VI: Feature Learning in Infinite Depth Neural Networks.
>
> [9] Poole et al. (2016). Exponential expressivity in deep neural networks through transient chaos.
>
> [10] Yang. (2021). Tensor Programs III: Neural Matrix Laws.
>
> [11] Sompolinsky et al. (1988). Chaos in Random Neural Networks.
>
> [12] Hayou et al. (2024). The Impact of Initialization on LoRA Finetuning Dynamics.

---

> > ### Author Rebuttal · Reviewer_pgWT · 2026-04-04
> >
> > Thanks for the clarification on informal results. I decide to raise score as promise.

---

> > > ### Author Response · Authors · 2026-04-05
> > >
> > > Thank you for considering our rebuttal and for confirming that your concerns have been resolved. We greatly appreciate your careful reading and rigorous feedback, which has helped strengthen our work. We will revise the manuscript accordingly and incorporate the justifications for the approximations in the future version.

---

### Official Review · Reviewer_eUPW · 2026-03-12

**Soundness:** 4
**Presentation:** 4
**Significance:** 4
**Originality:** 4
**Overall Recommendation:** 5
**Confidence:** 4

**Summary:**

The authors of this paper investigate how Low-Rank Adaptation (LoRA) updates and hyperparameters interact with the learning rate. Through a rigorous theoretical analysis, the authors propose Maximal-Update Adaptation (μA), a theoretical framework that characterizes how the optimal learning rate should scale with model properties such as width (n) and LoRA rank (r). The analysis reveals that the optimal learning rate exhibits different scaling patterns depending on the initialization scheme, the LoRA scaling factors, and specifically whether the adapter starts with a random matrix A and zero matrix B (Init[A]), or a zero matrix A and random matrix B (Init[B]). The authors successfully identify two distinct scaling regimes and ultimately link LoRA updates (Init[B]) to Full Fine-Tuning. All theoretical findings, while rigorously grounded in math, are empirically validated through extensive experiments across a wide array of modalities and tasks. Specifically, the authors demonstrate the robustness of their scaling rules on real-world architectures through supervised fine-tuning for language and vision models, reinforcement learning with verifiable rewards, and diffusion models, confirming that the theoretically derived learning rates can be applied under practical, large-scale training conditions.

**Compliance With Llm Reviewing Policy:**

Affirmed.

**Final Justification:**

The rebuttal effectively addressed my main concerns about SignSGD abstraction. This reinforced my prior assessment, and I confirm my score.

**Key Questions For Authors:**

As a minor suggestion to enrich the final version: could the authors add a discussion in the paper on why the scaling rules derived under SignSGD transfer so effectively to AdamW in the experiments ?

**Limitations:**

yes

**Strengths And Weaknesses:**

**Strengths**

Rigorous theoretical characterization: the approach to studying the network in the joint limit of infinite width (n) and rank (r) to define the conditions for stable feature learning is both elegant and highly rigorous. By explicitly defining the mathematical objective of a LoRA update that is forcing the feature updates to be bounded at Θ(1), the authors avoid relying on empirical heuristics. Instead, they successfully map out the exact asymptotic conditions required for the learning rate to reach that stability, establishing the Maximal-Update Adaptation (μA) framework. This provides a deeply principled foundation for hyperparameter scaling.

Comprehensive analysis of LoRA initialization and scaling: the paper does an excellent job of covering all standard LoRA configurations by exhaustively analyzing the two primary initialization schemes: Init[A] (random A, zero B) and Init[B] (zero A, random B). By coupling this analysis with the various standard LoRA scaling factors (α), the authors thoroughly demystify the complex hyperparameters interaction, proving mathematically why different configurations yield fundamentally different learning rate dynamics.

Extensive and robust empirical validation: after mathematically deriving the conditions for optimal learning rates in the infinite n and r limits, the authors provide empirical validation showing that these theories hold true in finite real-world networks. Notably, the authors explicitly address the theory-experiment alignment by acknowledging and accounting for the gap between asymptotic limits and finite experimental settings. The experiments are highly comprehensive, spanning multiple modalities including supervised fine-tuning (SFT) for language and vision architectures, reinforcement learning with verifiable rewards (RLVR), and diffusion models. The continuous and convincing alignment of the empirical plots with the theoretical narratives gives the paper a substantial scientific contribution.

Connection to FFT: the finding that the optimal learning rate for Init[B] depends only on model width (n) and is independent of rank (r) is a valuable contribution. It can establish a practical mechanism to transfer hyperparameters directly from a low-rank adapter to full finetuning.


**Weaknesses**

Gap between theoretical and empirical optimizers: to maintain mathematical tractability, the μA framework relies on assumption 4.2, which abstracts the optimizer to SignSGD without momentum and assumes a single-sample loss. However, all empirical validations use AdamW with batched inputs and standard moment estimates. While the authors discuss finite-size effects in Section 5.1, they omit a discussion on how AdamW’s adaptive denominators and momentum buffers might theoretically alter the scaling rules. Briefly justifying why the SignSGD approximation safely transfers to AdamW in practice would strengthen the theoretical claims.

---

> ### Author Rebuttal · Authors · 2026-03-30
>
> Thank you for your valuable feedback. Due to space limitations, we refer to each question/weakness by number. Additional experimental results are provided in the figures via anonymous links below, discussed in detail throughout our response.
>
> - Fig. 1: https://sub-28479.b-cdn.net/SignSGD.png
> - Fig. 2: https://sub-28479.b-cdn.net/Muon.png
>
>
> **W1 & Q1:**
>
> Thank you for these suggestions. We adopt SignSGD as an abstraction of AdamW because it yields closed-form expressions (see also [1,2]). We hypothesize that the scaling rules derived under SignSGD remain valid under AdamW because SignSGD captures a property of AdamW most relevant to scaling analysis: **element-wise gradient normalization**. In AdamW, the adaptive second-moment denominator rescales each gradient coordinate so that the effective update magnitude is approximately $\Theta(1)$ with respect to width and rank, regardless of the raw gradient scale. SignSGD is basically AdamW without momentum. We acknowledge that AdamW's momentum and second-moment estimates additionally affect training dynamics, but argue that these components primarily smooth the per-element updates without altering the $\Theta(1)$ scaling of each update element. In an additional experiment replacing AdamW with SignSGD (see Fig. 1), we observe that the learning rate scaling patterns remain consistent, suggesting that momentum is not the dominant factor governing the scaling rules. SignSGD thus serves as a tractable, empirically justified proxy for AdamW in our derivations.
>
> Note that the SignSGD abstraction in our framework is specifically tailored to Adam-like optimizers. Other optimizer families impose different gradient normalization structures and would require separate abstractions. For instance, SGD uses raw unnormalized gradients, LARS [3] enforces a unit $\ell_2$-norm per layer, and Muon [4] enforces a unit spectral norm. These distinct normalization structures will alter the asymptotic scaling of feature updates. For example, in an additional experiment replacing AdamW with Muon (see Fig. 2), we observe a visibly different scaling pattern, confirming that a separate analysis would be needed. Importantly, our $\mu A$ framework is modular: Assumption 4.2 can be replaced with an appropriate counterpart for other optimizer families, and the same analysis pipeline can be reused to derive the corresponding scaling rules. We consider this a promising direction for future work, but beyond the scope of this paper.
>
> For the single-sample loss, since gradients of matrix-valued parameters are approximately low-rank in practice (see e.g. [5]), we expect a single-sample gradient to capture the leading direction relevant to our scaling analysis while keeping the derivations closed-form.
>
> We will incorporate this discussion into the revised manuscript.
>
> [1] Bernstein et al. (2018). signSGD: Compressed Optimisation for Non-Convex Problems.
>
> [2] Hayou et al. (2024). Lora+: Efficient low rank adaptation of large models.
>
> [3] You et al. (2017). Scaling sgd batch size to 32k for imagenet training.
>
> [4] Jordan et al. (2024). Muon: An optimizer for hidden layers in neural networks.
>
> [5] Zhao et al. (2024). GaLore: Memory-Efficient LLM Training by Gradient Low-Rank Projection.

---

> > ### Author Rebuttal · Reviewer_eUPW · 2026-04-01
> >
> > Thank you for the clarifications. My concerns are resolved, and I confirm my score.

---

> > > ### Author Response · Authors · 2026-04-05
> > >
> > > Thank you for considering our rebuttal and for confirming that your concerns have been resolved. We greatly appreciate your constructive feedback, which has helped strengthen our work. We will make sure to include the discussion on SignSGD abstraction in the future version.

---

### Decision · Program_Chairs · 2026-04-30

**Decision:**

Accept (regular)

**Comment:**

This paper identifies the regimes when learning rate transfers across ranks and towards full fine-tuning. The authors propose muA to identify transferrable combinations of initialization and alpha-scaling. Extensive experiments show very promising results. All reviewers have concerns and questions regarding the theoretical assumptions, which may not hold in practice. These issues are mostly resolved by further discussion and empirical results. Overall, there is some room for theoretical improvement but the practical applicability outweighs the discrepancy.